# Addressing Parameter Choice Issues in Unsupervised Domain Adaptation by Aggregation

**Marius-Constantin Dinu**[1,2]   **Markus Holzleitner**[1]   **Maximilian Beck**[1]

**Duc Hoan Nguyen**[5]   **Andrea Huber**[1]   **Hamid Eghbal-zadeh**[1]   **Bernhard A. Moser**[4]

**Sergei V. Pereverzyev**[5]   **Sepp Hochreiter**[1,3]   **Werner Zellinger**[5]

[1]ELLIS Unit Linz and LIT AI Lab, Institute for Machine Learning, Johannes Kepler University Linz
[2]Dynatrace Research
[3]Institute of Advanced Research in Artificial Intelligence
[4]Software Competence Center Hagenberg
[5]Johann Radon Institute for Computational and Applied Mathematics, Austrian Academy of Sciences

## Abstract

We study the problem of choosing algorithm hyper-parameters in unsupervised domain adaptation, i.e., with labeled data in a source domain and unlabeled data in a target domain, drawn from a different input distribution. We follow the strategy to compute several models using different hyper-parameters, and, to subsequently compute a linear aggregation of the models. While several heuristics exist that follow this strategy, methods are still missing that rely on thorough theories for bounding the target error. In this turn, we propose a method that extends weighted least squares to vector-valued functions, e.g., deep neural networks. We show that the target error of the proposed algorithm is asymptotically not worse than twice the error of the unknown optimal aggregation. We also perform a large scale empirical comparative study on several datasets, including text, images, electroencephalogram, body sensor signals and signals from mobile phones. Our method[1] outperforms deep embedded validation (DEV) and importance weighted validation (IWV) on all datasets, setting a new state-of-the-art performance for solving parameter choice issues in unsupervised domain adaptation with theoretical error guarantees. We further study several competitive heuristics, all outperforming IWV and DEV on at least five datasets. However, our method outperforms each heuristic on at least five of seven datasets.

## 1 Introduction

The goal of *unsupervised domain adaptation* is to learn a model on unlabeled data from a *target* input distribution using labeled data from a different *source* distribution (Pan & Yang, 2010; Ben-David et al., 2010). If this goal is achieved, medical diagnostic systems can successfully be trained on unlabeled images using labeled images with a different modality (Varsavsky et al., 2020; Zou et al., 2020); segmentation models for natural images can be learned using only labeled data from computer simulations Peng et al. (2018); natural language models can be learned from unlabeled biomedical abstracts by means of labeled data from financial journals (Blitzer et al., 2006); industrial quality inspection systems can be learned on unlabeled data from new products using data from related products (Jiao et al., 2019; Zellinger et al., 2020).

However, missing target labels combined with distribution shift makes parameter choice a hard problem (Sugiyama et al., 2007; You et al., 2019; Saito et al., 2021; Zellinger et al., 2021; Musgrave et al., 2021). Often, one ends up with a sequence of models, e.g., originating from different hyper-parameter configurations (Ben-David et al., 2007; Saenko et al., 2010; Ganin et al., 2016; Long et al.,

---

[1]Large scale benchmark experiments are available at `https://github.com/Xpitfire/iwa`; dinu@ml.jku.at, werner.zellinger@ricam.oeaw.ac.at

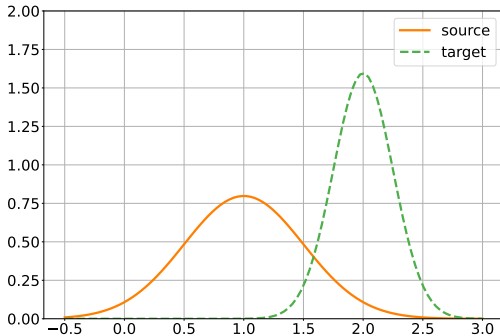 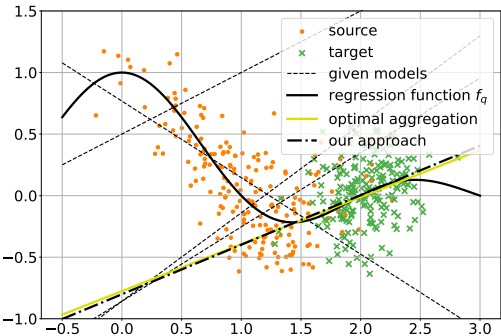

Figure 1: Unsupervised domain adaptation problem (Shimodaira, 2000; Sugiyama et al., 2007; You et al., 2019). Left: Source distribution (solid) and target distribution (dashed). Right: A sequence of different linear models (dashed) is used to find the optimal linear aggregation of the models (solid). Model selection methods (Sugiyama et al., 2007; Kouw et al., 2019; You et al., 2019; Zellinger et al., 2021) cannot outperform the best single model in the sequence, confidence values as used in Zou et al. (2018) are not available, and, approaches based on averages or tendencies of majorities of models (Saito et al., 2017) suffer from a high fraction of large-error-models in the sequence. In contrast, our approach (dotted-dashed) is nearly optimal. In addition, the model computed by our method provably approaches the optimal linear aggregation for increasing sample size. For further details on this example we refer to Section C in the Supplementary Material.

2015; Zellinger et al., 2017; Peng et al., 2019). In this work, we study the problem of constructing an optimal aggregation using all models in such a sequence. Our main motivation is that the error of such an optimal aggregation is clearly smaller than the error of the best single model in the sequence.

Although methods with mathematical error guarantees have been proposed to select the best model in the sequence (Sugiyama et al., 2007; Kouw et al., 2019; You et al., 2019; Zellinger et al., 2021), methods for learning aggregations of the models are either heuristics or their theory guarantees are limited by severe assumptions (cf. Wilson & Cook (2020)). Typical aggregation approaches are (a) to learn an aggregation on source data only (Nozza et al., 2016), (b) to learn an aggregation on a set of (unknown) labeled target examples (Xia et al., 2013; Dai et al., 2007; III & Marcu, 2006; Duan et al., 2012), (c) to learn an aggregation on target examples (pseudo-)labeled based on confidence measures of the given models (Zhou et al., 2021; Ahmed et al., 2022; Sun, 2012; Zou et al., 2018; Saito et al., 2017), (d) to aggregate the models based on data-structure specific transformations (Yang et al., 2012; Ha & Youn, 2021), and, (e) to use specific (possibly not available) knowledge about the given models, such as information obtained at different time-steps of its gradient-based optimization process (French et al., 2018; Laine & Aila, 2017; Tarvainen & Valpola, 2017; Athiwaratkun et al., 2019; Al-Stouhi & Reddy, 2011) or the information that the given models are trained on different (source) distributions (Hoffman et al., 2018; Rakshit et al., 2019; Xu et al., 2018; Kang et al., 2020; Zhang et al., 2015). One problem shared among all methods mentioned above is that they cannot guarantee a small error, even if the sample size grows to infinity. See Figure 1 for a simple illustrative example.

In this work, we propose (to the best of our knowledge) the first algorithm for computing aggregations of vector-valued models for unsupervised domain adaptation with target error guarantees. We extend the *importance weighted least squares algorithm* (Shimodaira, 2000) and corresponding recently proposed error bounds (Gizewski et al., 2022) to linear aggregations of vector-valued models. The importance weights are the values of an estimated ratio between target and source density evaluated at the examples. Every method for density-ratio estimation can be used as a basis for our approach, e.g. Sugiyama et al. (2012); Kanamori et al. (2012) and references therein. Our error bound proves that the target error of the computed aggregation is asymptotically at most twice the target error of the optimal aggregation.

In addition, we perform extensive empirical evaluations on several datasets with academic data (Transformed Moons), text data (Amazon Reviews (Blitzer et al., 2006)), images (MiniDomainNet (Peng et al., 2019; Zellinger et al., 2021)), electroencephalography signals (Sleep-EDF (Eldele et al., 2021; Goldberger et al., 2000)), body sensor signals (UCI-HAR (Anguita et al., 2013), WISDM (Kwapisz et al., 2011)), and, sensor signals from mobile phones and smart watches (HHAR (Stisen et al., 2015)).

We compute aggregations of models obtained from different hyper-parameter settings of 11 domain adaptation methods (e.g., DANN (Ganin et al., 2016) and Deep-Coral Sun & Saenko (2016)). Our method sets a new state of the art for methods with theoretical error guarantees, namely importance weighted validation (IWV) (Sugiyama et al., 2007) and deep embedded validation (DEV) (Kouw et al., 2019), on all datasets. We also study (1) classical least squares aggregation on source data only, (2) majority voting on target predictions, (3) averaging over model confidences, and (4) learning based on pseudo-labels. All of these heuristics outperform IWV and DEV on at least five of seven datasets, which is a result of independent interest. In contrast, our method outperforms each heuristic on at least five of seven datasets.

Our main contributions are summarized as follows:

- We propose the (to the best of our knowledge) first algorithm for ensemble learning of vector-valued models in (single-source) unsupervised domain adaptation that satisfies a non-trivial target error bound.

- We prove that the target error of our algorithm is asymptotically (for increasing sample sizes) at most twice the target error of the unknown optimal aggregation.

- We outperform IWV and DEV, and therefore set a new state-of-the-art performance for re-solving parameter choice issues under theoretical target error guarantees.

- We describe four heuristic baselines which all outperform IWV and DEV on at least five of seven datasets. Our method outperforms each heuristic on at least five of seven datasets.

- Our method tends to be more stable than others w.r.t. adding inaccurate models to the given sequence of models.

## 2 RELATED WORK

It is well known that aggregations of models in an ensemble often outperform individual models (Dong et al., 2020; Goodfellow et al., 2016). Traditional ensemble methods that have shown the advantage of aggregation are Boosting (Schapire, 1990; Breiman, 1998), Bootstrap Aggregating (bagging) (Breiman, 1994; 1996a) and Stacking (Wolpert, 1992; Breiman, 1996b). For example, averages of multiple models pre-trained on data from a distribution different from the target one have recently been shown to achieve state-of-the-art performance on ImageNet (Wortsman et al., 2022) and their good generalization properties can be related to flat minima (Hochreiter & Schmidhuber, 1994; 1997). However, most such methods don't take into account a present distribution shift.

Although some ensemble learning methods exist, which take into account a present distribution shift, in contrast to our work, they are either relying on labeled target data (Nozza et al., 2016; Xia et al., 2013; III & Marcu, 2006; Dai et al., 2007; Mayr et al., 2016), are restricted by fixing the aggregation weights to be the same (Razar & Samothrakis, 2019), make assumptions on the models in the sequence or the corresponding process for learning the models (Yang et al., 2012; Ha & Youn, 2021; French et al., 2018; Laine & Aila, 2017; Tarvainen & Valpola, 2017; Athiwaratkun et al., 2019; Al-Stouhi & Reddy, 2011; Hoffman et al., 2018; Rakshit et al., 2019; Xu et al., 2018; Kang et al., 2020; Zhang et al., 2015), or, learn an aggregation based on the heuristic approach of (pseudo-)labeling some target data based on confidence measures of models in the sequence (Zhou et al., 2021; Ahmed et al., 2022; Sun, 2012; Zou et al., 2018; Saito et al., 2017). Another crucial difference of all methods above is that none of these methods can guarantee a small target error in the general setting (distribution shift, vector valued models, different classes, single source domain) described above, even if the sample size grows to infinity.

Another branch of research are methods which aim at selecting the best model in the sequence. Although, such methods with error bounds have been proposed for the general setting above (Sugiyama et al., 2007; You et al., 2019; Zellinger et al., 2021), they cannot overcome a limited performance of the best model in the given sequence (cf. Figure 1 and Section 6 in the Supplementary Material of Zellinger et al. (2021)). In contrast, our method can outperform the best model in the sequence, and our empirical evaluations show that this is indeed the case in practical examples. A recent kernel-based algorithm for univariate regression, that is similar to ours, can be found in Gizewski et al. (2022). However, in contrast to Gizewski et al. (2022), our method allows a much more general form of vector-valued models which are not necessarily obtained from regularized kernel least squares, and, can therefore be applied to practical deep learning tasks.

Our work employs technical tools developed in Caponnetto & De Vito (2007; 2005). In fact, we extend Caponnetto & De Vito (2007; 2005) to deal with importance weighted least squares. Finally, it is important to note Huang et al. (2006), where a core Lemma of our proofs is proposed.

## 3 AGGREGATION BY IMPORTANCE WEIGHTED LEAST SQUARES

This section gives a summary of the main problem of this paper and our approach. For detailed assumptions and proofs, we refer to Section A of the Supplementary Material.

**Notation and Setup**  Let $\mathcal{X} \subset \mathbb{R}^{d_1}$ be a compact *input space* and $\mathcal{Y} \subset \mathbb{R}^{d_2}$ be a compact *label space* with inner product $\langle ., . \rangle_{\mathcal{Y}}$ such that for the associated norm $\|y\|_{\mathcal{Y}} \leq y_0$ holds for all $y \in \mathcal{Y}$ and some $y_0 > 0$. Following Ben-David et al. (2010), we consider two datasets: A *source dataset* $(\mathbf{x}, \mathbf{y}) = ((x_1, y_1), \ldots, (x_n, y_n)) \in (\mathcal{X} \times \mathcal{Y})^n$ independently drawn according to some source distribution (probability measure) $p$ on $\mathcal{X} \times \mathcal{Y}$ and an unlabeled *target* dataset $\mathbf{x}' = (x_1', \ldots, x_m') \in \mathcal{X}^m$ with elements independently drawn according to the marginal distribution[2] $q_{\mathcal{X}}$ of some target distribution $q$ on $\mathcal{X} \times \mathcal{Y}$. The marginal distribution of $p$ on $\mathcal{X}$ is analogously denoted as $p_{\mathcal{X}}$. We further denote by $\mathcal{R}_q(f) = \int_{\mathcal{X} \times \mathcal{Y}} \|f(x) - y\|_{\mathcal{Y}}^2 \, \mathrm{d}q(x, y)$ the *expected target risk* of a vector valued function $f : \mathcal{X} \to \mathcal{Y}$ w.r.t. the least squares loss.

**Problem**  Given a set $f_1, \ldots, f_l : \mathcal{X} \to \mathcal{Y}$ of models, the labeled source sample $(\mathbf{x}, \mathbf{y})$ and the unlabeled target sample $\mathbf{x}'$, the problem considered in this work is to find a model $f : \mathcal{X} \to \mathcal{Y}$ with a minimal target error $\mathcal{R}_q(f)$.

**Main Assumptions**  We rely (a) on the *covariate shift* assumption that the source conditional distribution $p(y|x)$ equals the target conditional distribution $q(y|x)$, and, (b) on the *bounded density ratio* assumption that there is a function $\beta : \mathcal{X} \to [0, B]$ with $B > 0$ such that $\mathrm{d}q_{\mathcal{X}}(x) = \beta(x) \, \mathrm{d}p_{\mathcal{X}}(x)$.

**Approach**  Our goal is to compute the linear aggregation $f = \sum_{i=1}^{l} c_i f_i$ for $c_1, \ldots, c_l \in \mathbb{R}$ with minimal squared target risk $\mathcal{R}_q\left(\sum_{i=1}^{l} c_i f_i\right)$. Our approach relies on the fact that

$$\underset{c_1,\ldots,c_l \in \mathbb{R}}{\arg\min} \ \mathcal{R}_q\left(\sum_{i=1}^{l} c_i f_i\right) = \underset{c_1,\ldots,c_l \in \mathbb{R}}{\arg\min} \int_{\mathcal{X}} \left\|\sum_{i=1}^{l} c_i f_i(x) - f_q(x)\right\|_{\mathcal{Y}}^2 \mathrm{d}q_{\mathcal{X}}(x) \qquad (1)$$

for the *regression functions* given by $f_q(x) = \int_{\mathcal{Y}} y \, \mathrm{d}q(y|x)$[3], see e.g. Cucker & Smale (2002, Proposition 1). Unfortunately, the right hand side of Eq. (1) contains information about labels $f_q(x)$ which are not given in our setting of unsupervised domain adaptation. However, borrowing an idea from *importance sampling*, it is possible to estimate Eq. (1). More precisely, from the covariate shift assumption we get $f_p(x) = \int_{\mathcal{Y}} y \, \mathrm{d}p(y|x) = f_q(x)$ and we can use the bounded density ratio $\beta$ to obtain

$$\underset{c_1,\ldots,c_l \in \mathbb{R}}{\arg\min} \ \mathcal{R}_q\left(\sum_{i=1}^{l} c_i f_i\right) = \underset{c_1,\ldots,c_l \in \mathbb{R}}{\arg\min} \int_{\mathcal{X}} \beta(x) \left\|\sum_{i=1}^{l} c_i f_i(x) - f_p(x)\right\|_{\mathcal{Y}}^2 \mathrm{d}p_{\mathcal{X}}(x) \qquad (2)$$

which extends importance weighted least squares (Shimodaira, 2000; Kanamori et al., 2009) to linear aggregations $\sum_{i=1}^{l} c_i f_i$ of vector-valued functions $f_1, \ldots, f_l$. The unique minimizer of Eq. (2) can be approximated based on available data analogously to classical least squares estimation as detailed in Algorithm 1. In the following, we call Algorithm 1 Importance Weighted Least Squares Linear Aggregation (IWA).

---

[2]The existence of the conditional probability density $q(y|x)$ with $q(x,y) = q(y|x)q_{\mathcal{X}}(x)$ is guaranteed by the fact that $\mathcal{X} \times \mathcal{Y}$ is Polish, i.e., a separable and complete metric space, c.f. Dudley (2002, Theorem 10.2.2.).

[3]$\mathcal{Y}$-valued integrals are defined in the sense of Lebesgue-Bochner.

**Relation to Model Selection** The optimal aggregation $f^* := \arg\min_{c_1,\ldots,c_l \in \mathbb{R}} \mathcal{R}_q \left( \sum_{i=1}^l c_i f_i \right)$ defined in Eq. (2) is clearly better than any single model selection since

$$\mathcal{R}_q(f^*) = \min_{c_1,\ldots,c_l \in \mathbb{R}} \mathcal{R}_q \left( \sum_{i=1}^l c_i f_i \right) \leq \min_{c_1,\ldots,c_l \in \{0,1\}} \mathcal{R}_q \left( \sum_{i=1}^l c_i f_i \right) \leq \min_{f_1,\ldots,f_l} \mathcal{R}_q(f_i). \quad (3)$$

However, the optimal aggregation $f^*$ cannot be computed based on finite datasets and the next logical questions are about the accuracy of the approximation $\widetilde{f}$ in Algorithm 1.

---

**Algorithm 1:** Importance Weighted Least Squares Linear Aggregation (IWA).

| | |
|---|---|
| **Input** | :Set $f_1,\ldots,f_l : \mathcal{X} \to \mathcal{Y}$ of models, labeled source sample $(\mathbf{x}, \mathbf{y})$ and unlabeled target sample $\mathbf{x}'$. |
| **Output** | :Linear aggregation $\widetilde{f} = \sum_{i=1}^l \widetilde{c}_i f_i$ with weights $\widetilde{c} = (\widetilde{c}_1,\ldots,\widetilde{c}_l) \in \mathbb{R}^l$. |

**Step 1** *Use unlabeled samples $\mathbf{x}$ and $\mathbf{x}'$ to approximate density ratio $\frac{\mathrm{d}q_{\mathcal{X}}}{\mathrm{d}p_{\mathcal{X}}}$ by some function $\beta : \mathcal{X} \to [0, B]$ using a classical algorithm, e.g. Sugiyama et al. (2012).*

**Step 2** *Compute weight vector $\widetilde{c} = \widetilde{G}^{-1}\widetilde{g}$ with empirical Gram matrix $\widetilde{G}$ and vector $\widetilde{g}$ defined by*

$$\widetilde{G} = \left( \frac{1}{m} \sum_{k=1}^m \langle f_i(x'_k), f_j(x'_k) \rangle_{\mathcal{Y}} \right)^l_{i,j=1} \qquad \widetilde{g} = \left( \frac{1}{n} \sum_{k=1}^n \beta(x_k) \langle y_k, f_i(x_k) \rangle_{\mathcal{Y}} \right)^l_{i=1}.$$

| | |
|---|---|
| **Return** | :Linear aggregation $\widetilde{f} = \sum_{i=1}^l \widetilde{c}_i f_i$. |

---

## 4 Target Error Bound for Algorithm 1

Let us start by introducing some further notation: $L^2(p)$ refers to the Lebesgue-Bochner space of functions from $\mathcal{X}$ to $\mathcal{Y}$, associated to a measure $p$ on $\mathcal{X}$ with corresponding inner product $\langle \cdot, \cdot \rangle_{L^2(p)}$ (this space basically consists of all $\mathcal{Y}$-valued functions whose $\mathcal{Y}$-norms are square integrable with respect to the given measure $p$). Moreover, let us introduce the (positive semi-definite) Gram matrix $G = \left( \langle f_i, f_j \rangle_{L^2(q_{\mathcal{X}})} \right)^l_{i,j=1}$ and the vector $\overline{g} = \left( \langle \beta f_p, f_i \rangle_{L^2(p_{\mathcal{X}})} \right)^l_{i=1}$. We can assume that $G$ is invertible (and thus positive definite), since otherwise some models are too similar to others and can be withdrawn from consideration (see Section D). Next, we recall that the minimizer of Eq. (2) is $c^* = (c_1^*,\ldots,c_l^*) = G^{-1}\overline{g}$, see Lemma 4.

However, neither $G$ nor the vector $\overline{g}$ is accessible in practice, because there is no access to the target measure $q_{\mathcal{X}}$. Driven by the law of large numbers we try to approximate them by averages over our given data and therefore arrive at the formulas for $\widetilde{G}$ and $\widetilde{g}$ given in Algorithm 1. This leads to the approximation $\widetilde{f}$. Up to this point, we were only considering an intuitive perspective on the problem setting, therefore, we will now formally discuss statements on the distance between the model $\widetilde{f}$ and the optimal linear model $f^* = \sum_{i=1}^l c_i^* f_i$, measured in terms of target risks, and how this distance behaves with increasing sample sizes. This is what we attempt with our main result:

**Theorem 1.** *With probability $1 - \delta$ it holds that*

$$\mathcal{R}_q(\tilde{f}) - \mathcal{R}_q(f_q) \leq 2 \left( \mathcal{R}_q(f^*) - \mathcal{R}_q(f_q) \right) + C \left( \log \frac{1}{\delta} \right) (n^{-1} + m^{-1}) \quad (4)$$

*for some coefficient $C > 0$ not depending on $m, n$ and $\delta$, and sufficiently large $m$ and $n$.*

Before we give an outline of the proof (see Section A), let us briefly comment on the main message of Algorithm 1. Observe, that (Cucker & Smale, 2002, Proposition 1) $\mathcal{R}_q(f) - \mathcal{R}_q(f_q) = \|f - f_q\|^2_{L^2(q_{\mathcal{X}})}$ can be interpreted as the total target error made by Algorithm 1, sometimes called *excess risk*. Indeed, in the deterministic setting of labeling functions, $f_q$ equals the target labeling function and the excess risk equals the target error of Ben-David et al. (2010). Eq. (4) compares this error for the aggregation $\widetilde{f}$, computed by Algorithm 1, to the error for the optimal aggregation $f^*$. Note that the error of the optimal aggregation $f^*$ is unavoidable in the sense that it is determined by

the decision of searching for linear aggregations of $f_1, \ldots, f_l$ only. However, if the models $f_1, \ldots, f_l$ are sufficiently different, then this error can be expected to be small. Theorem 1 tells us that the error of $\widetilde{f}$ approaches the one of $f^*$ with increasing target and source sample size. The rate of convergence is at least linear. Finally, we emphasize that Theorem 1 does not take into account the error of the density-ratio estimation. We refer to the recent work Gizewski et al. (2022), who, for the first time, included such error in the analysis of importance weighted least squares.

Let us now give a brief outline for the proof of Theorem 1. One key part concerns the existence of a Hilbert space $\mathcal{H}$ with associated inner product $\langle ., . \rangle_{\mathcal{H}}$ (a reproducing kernel space of functions from $\mathcal{X} \to \mathcal{Y}$) which contains all given models $f_1, \ldots, f_l$ and the regression function $f_q = f_p$. The space $\mathcal{H}$ can be constructed from any given models that are bounded and continuous functions. Furthermore, Algorithm 1 does not need any knowledge of $\mathcal{H}$, which is a modeling assumption only needed for the proofs, so that we can apply many arguments developed in Caponnetto & De Vito (2007; 2005). $\mathcal{H}$ is also not necessarily generated by a prescribed kernel such as Gaussian or linear kernel, and, no further smoothness assumption is required, see Sections A and B in the Supplementary Material.

Moreover, in this setting one can express the excess risk as follows: $\mathcal{R}_q(f) - \mathcal{R}_q(f_q) = \|A(f - f_q)\|_{\mathcal{H}}^2$ for some bounded linear operator $A : \mathcal{H} \to \mathcal{H}$. This also allows us to formulate the entries of $G$ and $\bar{g}$ in terms of the inner product $\langle ., . \rangle_{\mathcal{H}}$ instead. Using properties related to the operators that appear in the construction of $\mathcal{H}$, in combination with Hoeffding-like concentration bounds in Hilbert spaces and bounds that measure, e.g., the deviation between empirical averages in source and target domain (as done in Gretton et al. (2006, Lemma 4)), we can quantify differences between the entries of $G$ and $\widetilde{G}$ (and $\bar{g}$ and $\widetilde{g}$ respectively) in terms of $n$, $m$ and $\delta$. This leads to Eq. (4).

## 5 EMPIRICAL EVALUATIONS

We now empirically evaluate the performance of our approach compared to classical ensemble learning baselines and state-of-the-art model selection methods. Therefore, we structure our empirical evaluation as follows. First, we outline our experimental setup for unsupervised domain adaptation and introduce all domain adaptation methods for our analysis. Second, we describe the ensemble learning and model selection baselines, and third, we present the datasets used for our experiments. We then conclude with our results and a detailed discussion thereof.

### 5.1 EXPERIMENTAL SETUP

To assess the performance of our ensemble learning Algorithm 1 IWA, we perform numerous experiments with different domain adaptation algorithms on different datasets. By changing the hyper-parameters of each algorithm, we obtain, as results of applying these algorithms, sequences of models. The goal of our method is to find optimal models based on combinations of candidates from each sequence. As domain adaptation algorithms, we consider the AdaTime benchmark suite, and run our experiments on language, image, text and time-series data. This suite comprises a collection of 11 domain adaptation algorithms. We follow their evaluation setup and apply the following algorithms: Adversarial

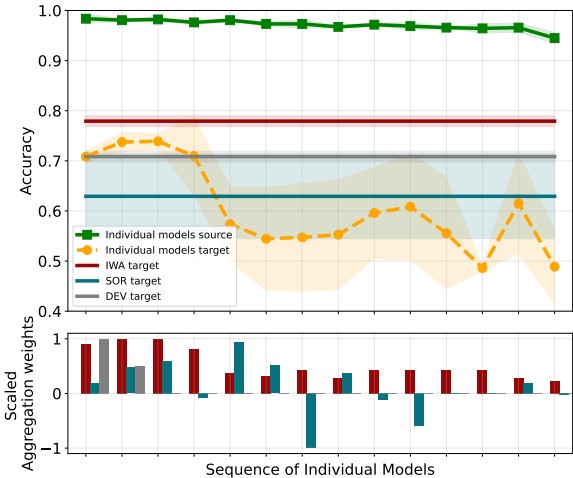

Figure 2: Top: Mean classification accuracy (y-axis) of our method (IWA), source-only regression (SOR), deep embedded validation (DEV) and individual models (green: source accuracy, orange: target accuracy) used in the aggregation for the HHAR dataset (Stisen et al., 2015) over 3 seeds. The individual models (x-axis) are trained with DIRT (Shu et al., 2018) for different hyper-parameter choices. Bottom: Scaled Aggregation weights (y-axis) for individual models (x-axis) computed by IWA, SOR and DEV (average over 3 seeds). Instead of searching for the best model in the sequence, IWA effectively uses all models in the sequence and obtains a performance not reachable by any procedure selecting only one model.

Spectral Kernel Matching (AdvSKM) (Liu & Xue, 2021), Deep Domain Confusion (DDC) (Tzeng et al., 2014), Correlation Alignment via Deep Neural Networks (Deep-Coral) (Sun et al., 2017), Central Moment Discrepancy (CMD) (Zellinger et al., 2017), Higher-order Moment Matching (HoMM) (Chen et al., 2020), Minimum Discrepancy Estimation for Deep Domain Adaptation (MMDA) (Rahman et al., 2020), Deep Subdomain Adaptation (DSAN) (Zhu et al., 2021), Domain-Adversarial Neural Networks (DANN) (Ganin et al., 2016), Conditional Adversarial Domain Adaptation (CDAN) (Long et al., 2018), A DIRT-T Approach to Unsupervised Domain Adaptation (DIRT) (Shu et al., 2018) and Convolutional deep Domain Adaptation model for Time-Series data (CoDATS) (Wilson et al., 2020). In addition to the sequence of models, IWA requires an estimate of the density ratio between source and target domain. To compute this quantity we follow (Bickel et al., 2007) and (You et al., 2019, Section 4.3), and, train a classifier discriminating between source and target data. The output of this classifier is then used to approximate the density ratio denoted as $\beta$ in Algorithm 1. Overall, to compute the results in our tables we trained 16680 models over approximately a timeframe of 1500 GPU/hours using computation resources of NVIDIA P100 16GB GPUs.

For example, consider the top plot of Figure 2, where we compare the performance of Algorithm 1 to deep embedded validation (DEV) (You et al., 2019), a heuristics baseline source-only regression (SOR, see Section 5.2) and each individual model in the sequence. The bottom plot shows the the scaled aggregation weights, i.e. how much each individual model contributes to the aggregated prediction of IWA, DEV, and SOR. In this example, the given sequence of models is obtained from applying the algorithm proposed in Shu et al. (2018) with different hyper-parameter choices to the Heterogeneity Human Activity Recognition dataset (Stisen et al., 2015). See Section D.3 in the Supplementary Material for the exact hyper-parameter values.

## 5.2 BASELINES

As representatives for the most prominent methods discussed in Section 1, we compare our method, IWA, to ensemble learning methods that use linear regression and majority voting as *heuristic* for model aggregation, and, model selection methods with *theoretical error guarantees*.

**Heuristic Baselines** The first baseline is majority voting on target data (TMV). It aggregates the predictions of all models by counting the overall class predictions and selects the class with the maximum prediction count as ensemble output. In addition, we implement three heuristic baselines which aggregate the vector-valued output, i.e. probabilities, of all classifiers using weights learned via linear regression. The final ensemble prediction is then made by selecting the class with the highest probability. The three heuristic regression baselines differ in the input used for the performed regression. Source-only regression (SOR) trains a regression model on classifier predictions (of the given models) and labels from the source domain only. Target majority voting regression (TMR) uses the same voting procedure as explained above to generate pseudo-labels on the target domain, which are then further used to train a linear regression model. In contrast, target confidence average regression (TCR) selects the highest average class probability over all classifiers to pseudo-label the target samples, which is then used for training the linear regression model.

**Baselines with Theoretical Error Guarantees** We compare IWA to the model selection methods importance weighted validation (IWV) (Sugiyama et al., 2007) and deep embedded validation (DEV) (You et al., 2019), which select models according to their (importance weighted) target risk. Both methods assume the knowledge of an estimated density ratio between target and source domains. In our experiments we follow Bickel et al. (2007); You et al. (2019) and estimate this ratio, by using a classifier that discriminates between source and target domain (see Supplementary Material Section D for more details).

## 5.3 DATASETS

We evaluate the previously mentioned methods according to a diverse set of datasets, including language, image and time-series data. All datasets have a train, evaluation and test split, with results only presented on the held-out test sets. For additional details we refer to Appendix C and D.

**TransformedMoons** This specific form of twinning moons is based on Zellinger et al. (2021). The source domain consists of two-dimensional input data points and their transformations to two opposing moon-shaped forms.

**MiniDomainNet** is a reduced version of DomainNet-2019 (Peng et al., 2019) consisting of six different image domains (Quickdraw, Real, Clipart, Sketch, Infograph, and Painting). In particular,

MiniDomainNet (Zellinger et al., 2021) reduces the number of classes of DomainNet-2019 to the top-five largest representatives in the training set of each class across all six domains.

**AmazonReviews** is based on Blitzer et al. (2006) and consists of text reviews from four domains: books, DVDs, electronics, and kitchen appliances. Reviews are encoded in feature vectors of bag-of-words unigrams and bigrams with binary labels indicating the rankings. From the four categories we obtain twelve domain adaptation tasks where each category serves once as source domain and once as target domain.

**UCI-HAR** The *Human Activity Recognition* (Anguita et al., 2013) dataset from the UC Irvine Repository contains data from three motion sensors (accelerometer, gyroscope and body-worn sensors) gathered using smartphones from 30 different subjects. It classifies their activities in several categories, namely, walking, walking upstairs, downstairs, standing, sitting, and lying down.

**WISDM** (Kwapisz et al., 2011) is a class-imbalanced dataset variant from collected accelerometer sensors, including GPS data, from 29 different subjects which are performing similar activities as in the UCI-HAR dataset.

**HHAR** The *Heterogeneity Human Activity Recognition* (Stisen et al., 2015) dataset investigate sensor-, device- and workload-specific heterogeneities using 36 smartphones and smartwatches, consisting of 13 different device models from four manufacturers.

**Sleep-EDF** The *Sleep Stage Classification* time-series setting aims to classify the electroencephalography (EEG) signals into five stages i.e., Wake (W), Non-Rapid Eye Movement stages (N1, N2, N3), and Rapid Eye Movement (REM). Analogous to Ragab et al. (2022); Eldele et al. (2021), we adopt the Sleep-EDF-20 dataset obtained from PhysioBank (Goldberger et al., 2000), which contains EEG readings from 20 healthy subjects.

We rely on the AdaTime benchmark suite (Ragab et al., 2022) in most evaluations. The four time-series datasets above are originally included there. We extend AdaTime to support the other discussed datasets as well, and extend its domain adaptation methods.

## 5.4 RESULTS

We separate the applied methods into two groups, namely *heuristic* and methods with *theoretical error guarantees*. All tables show accuracies of source-only (SO) and target-best (TB) models, where source-only denotes training without domain adaptation and target-best the best performing model obtained among all parameter settings. We highlight in bold the performance of the best performing method with theoretical error guarantees, and in italic the best performing heuristic. See Table 1 for results. Please find the full tables in the Supplementary Material Section D.

**Outperformance of theoretically justified methods:** On all datasets, our method outperforms IWV and DEV, setting a new state of the art for solving parameter choice issues under theoretical guarantees.

**Outperformance of heuristics:** It is interesting to note that each heuristic outperforms IWV and DEV on at least five of seven datasets. Moreover, every heuristic outperforms the (average) target best model (TB) in at least two cases, making it impossible for *any* model selection method to win in these cases. These facts highlight the quality of the predictions of our chosen heuristics. However, each heuristic is outperformed by our method on at least five of seven datasets.

**Information in aggregation weights and robustness w.r.t. inaccurate models:** It is interesting to observe that, in contrast to the other heuristic aggregation baselines, the aggregation weights $c_1, \ldots, c_l$ of our method tend to be larger for accurate models, see Section D.5. Another result is that our method tends to be less sensitive to a high number of inaccurate models than the baselines, see Section D.6. This serves as another reason for its high empirical performance.

## 6 CONCLUSION AND FUTURE WORK

We present a constructive theory-based method for approaching parameter choice issues in the setting of unsupervised domain adaptation. Its theoretical approach relies on the extension of weighted least squares to vector-valued functions. The resulting aggregation method distinguishes itself by a wide scope of admissible model classes without strong assumptions, e.g. support vector machines, decision trees and neural networks. A broad empirical comparative study on benchmark datasets for language,

images, body sensor signals and handy signals, underpins the theory-based optimality claim. It is left for future research to further refine the theory and its estimates, e.g., by exploiting concentration bounds from Gretton et al. (2006) or advanced density ratio estimators from Sugiyama et al. (2012).

Table 1: Mean and standard deviation (after ±) of target classification accuracy on Amazon Reviews, Sleep-EDF, UCI-HAR, HHAR and WISDM datasets over three different random initialization of model weights and several domain adaptation tasks.

| Amazon Reviews | | | | | | | | | |
|---|---|---|---|---|---|---|---|---|---|
| | | Heuristic | | | | Theoretical error guarantees | | | |
| Method | SO | TMV | TMR | TCR | SOR | IWV | DEV | IWA (ours) | TB |
| HoMM | 0.769(±0.009) | 0.777(±0.010) | 0.778(±0.010) | 0.778(±0.011) | 0.777(±0.010) | 0.765(±0.011) | 0.766(±0.011) | **0.778(±0.010)** | 0.769(±0.012) |
| AdvSKM | 0.766(±0.012) | 0.780(±0.009) | 0.779(±0.010) | 0.779(±0.008) | 0.778(±0.011) | 0.769(±0.012) | 0.766(±0.012) | **0.780(±0.009)** | 0.770(±0.012) |
| DIRT | 0.764(±0.009) | 0.786(±0.008) | 0.786(±0.010) | 0.786(±0.008) | 0.800(±0.008) | 0.778(±0.022) | 0.773(±0.056) | **0.787(±0.008)** | 0.786(±0.009) |
| DDC | 0.766(±0.012) | 0.779(±0.010) | 0.780(±0.009) | 0.779(±0.010) | 0.778(±0.010) | 0.767(±0.017) | 0.768(±0.011) | **0.780(±0.010)** | 0.770(±0.013) |
| CMD | 0.767(±0.012) | 0.791(±0.009) | 0.792(±0.010) | 0.789(±0.010) | 0.792(±0.010) | 0.765(±0.015) | 0.710(±0.015) | **0.794(±0.009)** | 0.785(±0.009) |
| MMDA | 0.767(±0.011) | 0.787(±0.011) | 0.785(±0.010) | 0.785(±0.010) | 0.787(±0.012) | 0.769(±0.011) | 0.766(±0.010) | **0.787(±0.011)** | 0.782(±0.011) |
| CoDATS | 0.766(±0.013) | 0.795(±0.009) | 0.793(±0.010) | 0.794(±0.012) | 0.799(±0.010) | 0.779(±0.016) | 0.773(±0.020) | **0.796(±0.009)** | 0.791(±0.015) |
| Deep-Coral | 0.766(±0.012) | 0.784(±0.009) | 0.783(±0.009) | 0.783(±0.009) | 0.782(±0.009) | 0.769(±0.016) | 0.769(±0.037) | **0.785(±0.009)** | 0.776(±0.013) |
| CDAN | 0.767(±0.012) | 0.788(±0.010) | 0.787(±0.009) | 0.787(±0.010) | 0.787(±0.011) | 0.775(±0.011) | 0.776(±0.014) | **0.788(±0.010)** | 0.777(±0.011) |
| DANN | 0.767(±0.012) | 0.796(±0.010) | 0.792(±0.010) | 0.793(±0.010) | 0.800(±0.011) | 0.776(±0.011) | 0.778(±0.012) | **0.797(±0.009)** | 0.798(±0.012) |
| DSAN | 0.769(±0.009) | 0.796(±0.009) | 0.792(±0.009) | 0.791(±0.010) | 0.800(±0.010) | 0.779(±0.012) | 0.763(±0.017) | **0.795(±0.009)** | 0.789(±0.012) |
| Avg. | 0.767(±0.011) | 0.787(±0.009) | 0.786(±0.010) | 0.786(±0.010) | 0.789(±0.010) | 0.772(±0.014) | 0.764(±0.019) | **0.788(±0.009)** | 0.781(±0.012) |

| Sleep-EDF | | | | | | | | | |
|---|---|---|---|---|---|---|---|---|---|
| | | Heuristic | | | | Theoretical error guarantees | | | |
| Method | SO | TMV | TMR | TCR | SOR | IWV | DEV | IWA (ours) | TB |
| HoMM | 0.676(±0.036) | 0.722(±0.017) | 0.719(±0.023) | 0.718(±0.021) | 0.724(±0.032) | 0.726(±0.046) | 0.678(±0.035) | **0.747(±0.025)** | 0.715(±0.047) |
| AdvSKM | 0.665(±0.058) | 0.708(±0.023) | 0.712(±0.027) | 0.712(±0.032) | 0.718(±0.030) | 0.703(±0.069) | 0.692(±0.038) | **0.722(±0.025)** | 0.706(±0.054) |
| DIRT | 0.656(±0.058) | 0.743(±0.009) | 0.745(±0.012) | 0.748(±0.019) | 0.742(±0.031) | 0.679(±0.038) | 0.686(±0.066) | **0.749(±0.010)** | 0.728(±0.037) |
| DDC | 0.646(±0.035) | 0.717(±0.029) | 0.721(±0.037) | 0.712(±0.031) | 0.695(±0.020) | 0.694(±0.066) | 0.666(±0.031) | **0.724(±0.017)** | 0.704(±0.031) |
| CMD | 0.653(±0.057) | 0.740(±0.022) | 0.736(±0.016) | 0.723(±0.020) | 0.709(±0.015) | 0.716(±0.052) | 0.640(±0.068) | **0.729(±0.018)** | 0.725(±0.053) |
| MMDA | 0.650(±0.051) | 0.736(±0.014) | 0.727(±0.021) | 0.723(±0.018) | 0.714(±0.028) | 0.704(±0.033) | 0.660(±0.034) | **0.745(±0.031)** | 0.715(±0.042) |
| CoDATS | 0.672(±0.084) | 0.738(±0.029) | 0.739(±0.036) | 0.736(±0.030) | 0.723(±0.039) | 0.683(±0.090) | 0.690(±0.107) | **0.744(±0.021)** | 0.715(±0.045) |
| Deep-Coral | 0.643(±0.049) | 0.716(±0.018) | 0.717(±0.028) | 0.712(±0.027) | 0.694(±0.032) | 0.700(±0.053) | 0.675(±0.077) | **0.713(±0.021)** | 0.702(±0.070) |
| CDAN | 0.652(±0.056) | 0.732(±0.016) | 0.739(±0.024) | 0.739(±0.018) | 0.728(±0.029) | 0.697(±0.031) | 0.642(±0.065) | **0.748(±0.019)** | 0.713(±0.045) |
| DANN | 0.641(±0.047) | 0.722(±0.017) | 0.723(±0.026) | 0.721(±0.025) | 0.714(±0.024) | 0.687(±0.034) | 0.644(±0.046) | **0.724(±0.018)** | 0.710(±0.035) |
| DSAN | 0.653(±0.060) | 0.748(±0.008) | 0.740(±0.016) | 0.732(±0.016) | 0.728(±0.026) | 0.712(±0.070) | 0.589(±0.063) | **0.757(±0.016)** | 0.700(±0.033) |
| Avg. | 0.655(±0.054) | 0.729(±0.018) | 0.729(±0.024) | 0.725(±0.023) | 0.717(±0.028) | 0.700(±0.052) | 0.660(±0.057) | **0.737(±0.020)** | 0.712(±0.045) |

| UCI-HAR | | | | | | | | | |
|---|---|---|---|---|---|---|---|---|---|
| | | Heuristic | | | | Theoretical error guarantees | | | |
| Method | SO | TMV | TMR | TCR | SOR | IWV | DEV | IWA (ours) | TB |
| HoMM | 0.782(±0.078) | 0.833(±0.020) | 0.818(±0.023) | 0.818(±0.022) | 0.783(±0.040) | 0.809(±0.095) | 0.800(±0.098) | **0.826(±0.010)** | 0.854(±0.039) |
| AdvSKM | 0.724(±0.059) | 0.791(±0.024) | 0.800(±0.022) | 0.810(±0.022) | 0.768(±0.042) | 0.707(±0.100) | 0.711(±0.167) | **0.800(±0.022)** | 0.811(±0.039) |
| DIRT | 0.783(±0.044) | 0.912(±0.013) | 0.907(±0.009) | 0.890(±0.016) | 0.756(±0.036) | 0.807(±0.107) | 0.808(±0.112) | **0.900(±0.015)** | 0.928(±0.034) |
| DDC | 0.790(±0.061) | 0.806(±0.019) | 0.807(±0.026) | 0.810(±0.017) | 0.756(±0.108) | 0.724(±0.066) | 0.734(±0.109) | **0.804(±0.028)** | 0.792(±0.013) |
| CMD | 0.788(±0.058) | 0.869(±0.012) | 0.849(±0.014) | 0.839(±0.023) | 0.731(±0.066) | 0.804(±0.064) | 0.812(±0.080) | **0.842(±0.025)** | 0.888(±0.037) |
| MMDA | 0.785(±0.018) | 0.819(±0.022) | 0.812(±0.028) | 0.800(±0.032) | 0.759(±0.085) | 0.773(±0.073) | 0.767(±0.107) | **0.807(±0.025)** | 0.840(±0.055) |
| CoDATS | 0.760(±0.037) | 0.854(±0.022) | 0.832(±0.027) | 0.832(±0.006) | 0.785(±0.057) | 0.801(±0.079) | 0.794(±0.078) | **0.846(±0.016)** | 0.867(±0.012) |
| Deep-Coral | 0.790(±0.035) | 0.810(±0.007) | 0.800(±0.022) | 0.808(±0.030) | 0.771(±0.025) | 0.768(±0.044) | 0.773(±0.087) | **0.808(±0.016)** | 0.806(±0.022) |
| CDAN | 0.756(±0.055) | 0.842(±0.009) | 0.843(±0.020) | 0.840(±0.034) | 0.802(±0.080) | 0.781(±0.072) | 0.687(±0.068) | **0.846(±0.018)** | 0.853(±0.026) |
| DANN | 0.756(±0.026) | 0.858(±0.016) | 0.856(±0.033) | 0.856(±0.033) | 0.800(±0.057) | 0.763(±0.043) | 0.780(±0.043) | **0.849(±0.023)** | 0.847(±0.007) |
| DSAN | 0.762(±0.032) | 0.849(±0.023) | 0.843(±0.033) | 0.854(±0.025) | 0.749(±0.065) | 0.775(±0.043) | 0.744(±0.035) | **0.858(±0.023)** | 0.865(±0.038) |
| Avg. | 0.770(±0.046) | 0.840(±0.017) | 0.833(±0.023) | 0.832(±0.024) | 0.769(±0.060) | 0.774(±0.070) | 0.765(±0.090) | **0.835(±0.020)** | 0.850(±0.029) |

| HHAR | | | | | | | | | |
|---|---|---|---|---|---|---|---|---|---|
| | | Heuristic | | | | Theoretical error guarantees | | | |
| Method | SO | TMV | TMR | TCR | SOR | IWV | DEV | IWA (ours) | TB |
| HoMM | 0.739(±0.044) | 0.757(±0.014) | 0.759(±0.013) | 0.759(±0.011) | 0.700(±0.058) | 0.720(±0.027) | 0.733(±0.031) | **0.759(±0.007)** | 0.764(±0.023) |
| AdvSKM | 0.718(±0.042) | 0.749(±0.027) | 0.742(±0.032) | 0.748(±0.034) | 0.676(±0.046) | 0.730(±0.051) | 0.728(±0.051) | **0.752(±0.031)** | 0.749(±0.025) |
| DIRT | 0.728(±0.026) | 0.803(±0.011) | 0.792(±0.016) | 0.803(±0.017) | 0.796(±0.066) | 0.743(±0.028) | 0.739(±0.075) | **0.816(±0.008)** | 0.820(±0.015) |
| DDC | 0.716(±0.063) | 0.748(±0.014) | 0.750(±0.009) | 0.748(±0.007) | 0.717(±0.075) | 0.711(±0.048) | 0.705(±0.066) | **0.748(±0.012)** | 0.729(±0.027) |
| CMD | 0.748(±0.027) | 0.760(±0.014) | 0.764(±0.006) | 0.767(±0.007) | 0.737(±0.100) | **0.775(±0.031)** | 0.643(±0.031) | 0.766(±0.016) | 0.794(±0.023) |
| MMDA | 0.738(±0.036) | 0.783(±0.017) | 0.781(±0.016) | 0.780(±0.015) | 0.698(±0.038) | 0.719(±0.036) | 0.731(±0.047) | **0.780(±0.017)** | 0.785(±0.035) |
| CoDATS | 0.710(±0.030) | 0.766(±0.023) | 0.772(±0.040) | 0.773(±0.050) | 0.722(±0.064) | 0.739(±0.028) | 0.739(±0.040) | **0.812(±0.009)** | 0.785(±0.039) |
| Deep-Coral | 0.745(±0.046) | 0.766(±0.012) | 0.762(±0.015) | 0.766(±0.027) | 0.681(±0.073) | 0.754(±0.054) | 0.758(±0.244) | **0.764(±0.006)** | 0.776(±0.023) |
| CDAN | 0.728(±0.039) | 0.762(±0.012) | 0.758(±0.017) | 0.764(±0.016) | 0.765(±0.063) | 0.774(±0.035) | 0.775(±0.036) | **0.816(±0.011)** | 0.790(±0.038) |
| DANN | 0.757(±0.057) | 0.779(±0.012) | 0.774(±0.009) | 0.773(±0.011) | 0.722(±0.103) | 0.798(±0.041) | 0.793(±0.045) | **0.818(±0.009)** | 0.807(±0.020) |
| DSAN | 0.721(±0.053) | 0.803(±0.010) | 0.797(±0.014) | 0.802(±0.007) | 0.724(±0.065) | 0.741(±0.033) | 0.596(±0.031) | **0.825(±0.008)** | 0.826(±0.046) |
| Avg. | 0.732(±0.042) | 0.771(±0.015) | 0.768(±0.017) | 0.771(±0.018) | 0.722(±0.068) | 0.746(±0.037) | 0.722(±0.063) | **0.787(±0.012)** | 0.784(±0.028) |

| WISDM | | | | | | | | | |
|---|---|---|---|---|---|---|---|---|---|
| | | Heuristic | | | | Theoretical error guarantees | | | |
| Method | SO | TMV | TMR | TCR | SOR | IWV | DEV | IWA (ours) | TB |
| HoMM | 0.753(±0.054) | 0.741(±0.026) | 0.738(±0.031) | 0.739(±0.047) | 0.775(±0.062) | 0.753(±0.054) | 0.740(±0.054) | 0.728(±0.021) | 0.774(±0.037) |
| AdvSKM | 0.747(±0.050) | 0.771(±0.043) | 0.781(±0.055) | 0.779(±0.035) | 0.742(±0.062) | 0.747(±0.050) | 0.747(±0.135) | **0.777(±0.031)** | 0.779(±0.041) |
| DIRT | 0.738(±0.038) | 0.792(±0.015) | 0.797(±0.024) | 0.797(±0.037) | 0.756(±0.071) | 0.738(±0.059) | 0.797(±0.059) | **0.798(±0.018)** | 0.816(±0.063) |
| DDC | 0.741(±0.071) | 0.780(±0.032) | 0.779(±0.052) | 0.787(±0.049) | 0.737(±0.071) | 0.741(±0.076) | 0.741(±0.063) | **0.782(±0.038)** | 0.770(±0.060) |
| CMD | 0.710(±0.088) | 0.772(±0.021) | 0.765(±0.032) | 0.767(±0.040) | 0.728(±0.092) | 0.713(±0.084) | 0.686(±0.113) | **0.773(±0.032)** | 0.742(±0.071) |
| MMDA | 0.759(±0.047) | 0.789(±0.017) | 0.772(±0.030) | 0.745(±0.035) | 0.754(±0.050) | 0.759(±0.047) | 0.750(±0.047) | **0.790(±0.018)** | 0.775(±0.030) |
| CoDATS | 0.711(±0.039) | 0.775(±0.018) | 0.757(±0.027) | 0.751(±0.020) | 0.682(±0.057) | 0.709(±0.039) | 0.735(±0.054) | **0.764(±0.015)** | 0.770(±0.019) |
| Deep-Coral | 0.694(±0.030) | 0.717(±0.041) | 0.723(±0.037) | 0.713(±0.035) | 0.664(±0.055) | 0.694(±0.030) | 0.670(±0.149) | **0.723(±0.026)** | 0.736(±0.044) |
| CDAN | 0.760(±0.057) | 0.762(±0.048) | 0.762(±0.046) | 0.781(±0.051) | 0.745(±0.074) | 0.760(±0.057) | 0.750(±0.091) | **0.765(±0.040)** | 0.779(±0.049) |
| DANN | 0.724(±0.042) | 0.789(±0.018) | 0.802(±0.028) | 0.796(±0.036) | 0.745(±0.044) | 0.720(±0.042) | 0.702(±0.053) | **0.778(±0.019)** | 0.765(±0.044) |
| DSAN | 0.759(±0.030) | 0.765(±0.024) | 0.769(±0.034) | 0.756(±0.025) | 0.757(±0.068) | **0.759(±0.030)** | 0.663(±0.025) | 0.722(±0.013) | 0.779(±0.044) |
| Avg. | 0.736(±0.050) | 0.768(±0.027) | 0.768(±0.036) | 0.765(±0.037) | 0.737(±0.062) | 0.736(±0.052) | 0.726(±0.077) | **0.764(±0.025)** | 0.771(±0.046) |

ACKNOWLEDGMENTS

The ELLIS Unit Linz, the LIT AI Lab, and the Institute for Machine Learning are supported by the Federal State Upper Austria. IARAI is supported by Here Technologies. We thank the projects AI-MOTION (LIT-2018-6-YOU-212), AI-SNN (LIT-2018-6-YOU-214), DeepFlood (LIT-2019-8-YOU-213), Medical Cognitive Computing Center (MC3), INCONTROL-RL (FFG-881064), PRIMAL (FFG-873979), S3AI (FFG-872172), DL for GranularFlow (FFG-871302), AIRI FG 9-N (FWF-36284, FWF-36235), and ELISE (H2020-ICT-2019-3 ID: 951847). We further thank Audi.JKU Deep Learning Center, TGW LOGISTICS GROUP GMBH, Silicon Austria Labs (SAL), FILL GmbH, Anyline GmbH, Google, ZF Friedrichshafen AG, Robert Bosch GmbH, UCB Biopharma SRL, Merck Healthcare KGaA, Verbund AG, TÜV Austria, Frauscher Sensonic, and the NVIDIA Corporation. The research reported in this paper has been funded by the Federal Ministry for Climate Action, Environment, Energy, Mobility, Innovation and Technology (BMK), the Federal Ministry for Digital and Economic Affairs (BMDW), and the Province of Upper Austria in the frame of the COMET–Competence Centers for Excellent Technologies Programme and the COMET Module S3AI managed by the Austrian Research Promotion Agency FFG.

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

# A  NOTATION AND PROOF OF MAIN RESULT

The aim of this section is to give a full proof of our main result, Theorem 1 in the main paper. We start by introducing and summarizing the notation and the required concepts from functional analysis and measure theory, so that we can state and prove the required lemmas.

## Summary of Notation

- *Spaces:* input space $\mathcal{X} \subset \mathbb{R}^{d_1}$ and label space $\mathcal{Y}$ with inner product $\langle .,. \rangle_{\mathcal{Y}}$. $\mathcal{Y}$ is assumed to be a separable Hilbert space such that for the associated norm $\|y\|_{\mathcal{Y}} \leq y_0$ holds for all $y \in \mathcal{Y}$ and some $y_0 > 0$. Note that this setting is more general than the one from the main text, where we assumed $\mathcal{Y} \subset \mathbb{R}^{d_2}$ (the simplification in the main text improves readability and respectation of space limits).

- *Datasets and Distributions:* Source data set: $(\mathbf{x}, \mathbf{y}) = ((x_1, y_1), \dots, (x_n, y_n)) \in (\mathcal{X} \times \mathcal{Y})^n$ independently drawn according to source distribution $p$ on $\mathcal{X} \times \mathcal{Y}$ and an unlabeled *target* dataset $\mathbf{x}' = (x_1', \dots, x_m') \in \mathcal{X}^m$ independently drawn according marginal distribution $q_{\mathcal{X}}$ of target distribution $q$ on $\mathcal{X} \times \mathcal{Y}$ (the corresponding marginal distribution of $p$ on $\mathcal{X}$ is similarly denoted as $p_{\mathcal{X}}$).

- *Source Risk:* $\mathcal{R}_p(f) = \int_{\mathcal{X} \times \mathcal{Y}} \|f(x) - y\|_{\mathcal{Y}}^2 \, \mathrm{d}p(x, y)$.

- *Source Regression function* $f_p(x) = \int_{\mathcal{Y}} y \, \mathrm{d}p(y|x)$. (Vector valued) integral in the sense of Lebesgue-Bochner.

- *Target Risk:* $\mathcal{R}_q(f) = \int_{\mathcal{X} \times \mathcal{Y}} \|f(x) - y\|_{\mathcal{Y}}^2 \, \mathrm{d}q(x, y)$

- *Target Regression function* $f_q(x) = \int_{\mathcal{Y}} y \, \mathrm{d}q(y|x)$. (Vector valued) integral in the sense of Lebesgue-Bochner.

## Problem

- *Given:* sequence $f_1, \dots, f_l : \mathcal{X} \to \mathcal{Y}$ of models, source sample $(\mathbf{x}, \mathbf{y})$ and unlabeled target sample $\mathbf{x}'$

- *Aim:* find aggregation $f = \sum_{i=1}^{l} c_i f_i$ with minimal $\mathcal{R}_q(f)$.

## Main Assumptions

- *covariate shift:* $p(y|x) = q(y|x)$ and thus $f_p = f_q$.

- *bounded density ratio:* there is $\beta : \mathcal{X} \to [0, B]$ such that $\mathrm{d}q_{\mathcal{X}}(x) = \beta(x) \, \mathrm{d}p_{\mathcal{X}}(x)$.

Existence of the associated conditional probability measures is guaranteed by the fact that $\mathcal{X} \times \mathcal{Y}$ is Polish (a separable and complete metric space), c.f. Dudley (2002, Theorem 10.2.2.).

**Notation from functional analysis/operator theory**  Let $\mathbf{U}$ and $\mathbf{V}$ denote separable Hilbert spaces (i.e. they admit countable orthonormal bases) with associated inner products $\langle .,. \rangle_{\mathbf{U}}$ (or $\langle .,. \rangle_{\mathbf{V}}$, respectively). Let us briefly recall some notions from functional analysis that we need in order to set up our theory. There are lots of standard references on these aspects, e.g. Teschl (2022a) and Teschl (2022b):

- $\mathcal{L}(\mathbf{U}, \mathbf{V})$: space of bounded linear operators $\mathbf{U} \to \mathbf{V}$ with uniform norm $\|.\|_{\mathcal{L}(\mathbf{U},\mathbf{V})}$. $\mathcal{L}(\mathbf{U})$: space of bounded linear operators $\mathbf{U} \to \mathbf{U}$.

- For $A \in \mathcal{L}(\mathbf{U}, \mathbf{V})$, its *adjoint* is denoted by $A^* \in \mathcal{L}(\mathbf{V}, \mathbf{U})$ (and uniquely defined by the equation $\langle Au, v \rangle_{\mathbf{V}} = \langle u, A^*v \rangle_{\mathbf{U}}$ for any $u \in \mathbf{U}, v \in \mathbf{V}$).

- If $A \in \mathcal{L}(\mathbf{U})$ and $A = A^*$: $A$ is called *self-adjoint*.

- If $A \in \mathcal{L}(\mathbf{U})$ is self adjoint and $\langle Au, u \rangle_{\mathbf{U}} \geq 0$ for any $u \in \mathbf{U}$, then $A$ is called *positive*. Equivalently: there exists (unique) bounded and self-adjoint $B := \sqrt{A} \in \mathcal{L}(\mathbf{U})$ such that $B^2 = A$.

- *Trace* of an operator $A \in \mathcal{L}(\mathbf{U})$: $Tr(A) = \sum_k \langle Ae_k, e_k \rangle_{\mathbf{U}}$ for any orthonormal basis $(e_k)_{k=1}^{\infty}$ of $\mathbf{U}$ (independent of choice of basis). If $Tr(A) < \infty$: $A$ is called *trace class*.
- $\mathcal{L}_2(\mathbf{U})$: separable Hilbert space of *Hilbert-Schmidt operators* on $\mathbf{U}$ with scalar product $\langle A, B \rangle_{\mathcal{L}_2(\mathbf{U})} = Tr(B^*A)$ and norm $\|A\|_{\mathcal{L}_2(\mathbf{U})} = \sqrt{Tr(A^*A)} \geq \|A\|_{\mathcal{L}(\mathbf{U})}$.
- $A : \mathbf{U} \to \mathbf{V}$ is called *Hilbert-Schmidt*, if $A^*A$ is trace class. Also here: $\|A\|_{\mathcal{L}(\mathbf{U}, \mathbf{V})} \leq \sqrt{Tr(A^*A)}$
- For (probability) measure $q$ on $\mathcal{X}$ (or $\mathcal{Y}$) and appropriate functions $F : \mathcal{X} \to \mathbf{U}$ (e.g. strongly measurable and $\|F\|_{\mathbf{U}}$ is integrable wrt. $q$) we denote the usual ($\mathbf{U}$-valued) *Bochner integral* of $F$ as $\int_{\mathcal{X}} F(x) \, dq(x)$. We denote the associated $L^p$-spaces by $L^p(\mathcal{X}, q, \mathbf{U})$, or $L^p(q)$ for short, if the associated spaces are clear from the context.

**Assumptions on models**  We assume that the regression function $f^* = f_p = f_q$ as well as the models $f_1, ..., f_l$ belong to a *hypothesis space* $\mathcal{H} \subseteq C(\mathcal{X}, \mathcal{Y}) \subseteq L^2(p_{\mathcal{X}}) \cap L^2(q_{\mathcal{X}})$, where $C(\mathcal{X}, \mathcal{Y})$ denotes the space of bounded continous functions $\mathcal{X} \to \mathcal{Y}$. The space $\mathcal{H}$ should satisfy the following assumptions, which are discussed in much greater detail in Caponnetto & De Vito (2007) and Caponnetto & De Vito (2005):

**Hypothesis 1.** *(Caponnetto & De Vito, 2007) The space $\mathcal{H}$ is a separable Hilbert space of functions $f : \mathcal{X} \to \mathcal{Y}$ such that:*

- *For all $x \in \mathcal{X}$ there is a Hilbert-Schmidt operator $K_x : \mathcal{Y} \to \mathcal{H}$ satisfying*

$$f(x) = K_x^* f, \quad f \in \mathcal{H}, \tag{5}$$

- *The function from $\mathcal{X} \times \mathcal{X}$ to $\mathbb{R}$*

$$(x, t) \mapsto \langle K_t v, K_x w \rangle_{\mathcal{H}} \text{ is measurable } \forall v, w \in \mathcal{Y}; \tag{6}$$

- *There is $\kappa > 0$ such that*

$$Tr(K_x^* K_x) \leq \kappa, \quad \forall x \in \mathcal{X}. \tag{7}$$

Moreover we assume that the norms $\|f_k\|_{\mathcal{H}}$, $k = 1, 2, \dots, l$, are under our control, such that we can put a threshold $\gamma_l > 0$ and consider $\|f_k\|_{\mathcal{H}} \leq \gamma_l$.

**Further useful observations**  Then we have

$$K_t^* K_x = K(t, x) \in \mathcal{L}_2(\mathcal{Y}) \quad \forall x, t \in \mathcal{X}. \tag{8}$$

Given $x \in \mathcal{X}$ the operator

$$T_x = K_x K_x^* \in \mathcal{L}_2(\mathcal{H}), \tag{9}$$

is a positive Hilbert-Schmidt operator and (9) ensures

$$\|T_x\|_{\mathcal{L}(\mathcal{H})} \leq \|T_x\|_{\mathcal{L}_2(\mathcal{H})} = \|K(x, x)\|_{\mathcal{L}_2(\mathcal{Y})} \leq \kappa. \tag{10}$$

Let $T_{q_{\mathcal{X}}} : \mathcal{H} \to \mathcal{H}$ be

$$T_{q_{\mathcal{X}}} = \int_{\mathcal{X}} T_x \, dq_{\mathcal{X}}(x),$$

where the integral converges in $\mathcal{L}_2(\mathcal{H})$ to a positive trace class operator with

$$\|T_{q_{\mathcal{X}}}\|_{\mathcal{L}(\mathcal{H})} \leq \|T_{q_{\mathcal{X}}}\|_{\mathcal{L}_2(\mathcal{H})} \leq Tr(T_{q_{\mathcal{X}}}) = \int_{\mathcal{X}} Tr(T_x) \, dq_{\mathcal{X}}(x) \leq \kappa. \tag{11}$$

Following Proposition 1 in Caponnetto & De Vito (2007), we have the minimizers $f_q$ of expected risk $\mathcal{R}_q$ are the solution of the following equation:

$$T_{q_{\mathcal{X}}} f_q = g,$$

where

$$g = \int_{\mathcal{X}} K_x f_q(x) \, \mathrm{d}q_{\mathcal{X}}(x) \in \mathcal{H},$$

with integral converging in $\mathcal{H}$.

Next we define the operators

$$T_{\mathbf{x}'} = \frac{1}{m} \sum_{j=1}^{m} K_{x_j'} K_{x_j'}^*,$$

$$T_{\mathbf{x},\beta} = \frac{1}{n} \sum_{i=1}^{n} \beta(x_i) K_{x_i} K_{x_i}^*,$$

$$g_{\mathbf{x},\mathbf{y},\beta} = \frac{1}{n} \sum_{i=1}^{n} \beta(x_i) K_{x_i} y_i.$$

In the sequel we adopt the convention that $C$ denotes a generic positive coefficient, which can vary from appearance to appearance and may only depend on basic parameter such as $p_{\mathcal{X}}$, $q_{\mathcal{X}}$, $\kappa$, $B$, $y_0$ and others introduced below, but not on $n, m$ and error probability $\delta > 0$.

We will need the following statements.

**Lemma 1.** *With probability at least $1 - \delta$ we have*

$$\|T_{q_{\mathcal{X}}} - T_{\mathbf{x}'}\|_{\mathcal{L}(\mathcal{H})} \le \|T_{q_{\mathcal{X}}} - T_{\mathbf{x}'}\|_{\mathcal{L}_2(\mathcal{H})} \le C \left( \log^{\frac{1}{2}} \frac{1}{\delta} \right) m^{-\frac{1}{2}}, \tag{12}$$

$$\|T_{\mathbf{x}'} - T_{\mathbf{x},\beta}\|_{\mathcal{L}(\mathcal{H})} \le C \left( \log^{\frac{1}{2}} \frac{1}{\delta} \right) \left( n^{-\frac{1}{2}} + m^{-\frac{1}{2}} \right), \tag{13}$$

$$\|T_{\mathbf{x},\beta} f^* - g_{\mathbf{x},\mathbf{y},\beta}\|_{\mathcal{H}} \le C \left( \log^{\frac{1}{2}} \frac{1}{\delta} \right) n^{-\frac{1}{2}}, \tag{14}$$

*where $C > 0$ does not depend on $n, m$ and $\delta$.*

The proof of Lemma 1 is based on Lemma 4 of Huang et al. (2006), which we formulate in our notations as follows

**Lemma 2.** *((Huang et al., 2006)) Let $\phi$ be a map from $\mathbf{U}$ to $\mathbf{U}$ such that $\|\phi(x)\|_{\mathbf{U}} \le R$ for all $x \in \mathcal{X}$. Then with probability at least $1 - \delta$ it holds*

$$\left\| \frac{1}{m} \sum_{j=1}^{m} \phi(x_j') - \frac{1}{n} \sum_{i=1}^{n} \beta(x_i) \phi(x_i) \right\|_{\mathbf{U}} \le \left( 1 + \sqrt{2 \log \frac{2}{\delta}} \right) R \sqrt{\frac{B^2}{n} + \frac{1}{m}}.$$

Moreover, we will need a concentration inequality that follows from Pinelis (1992), see also Rosasco et al. (2010).

**Lemma 3** (Concentration lemma). *If $\xi_1, \xi_2, \ldots, \xi_n$ are zero mean independent random variables with values in a separable Hilbert space $\mathbf{U}$, and for some $D > 0$ one has $\|\xi_i\|_{\mathbf{U}} \le D$, $i = 1, 2, \ldots, n$, then the following bound*

$$\left\| \frac{1}{n} \sum_{i=1}^{n} \xi_i \right\|_{\mathbf{U}} \le \frac{D \sqrt{2 \log \frac{2}{\delta}}}{\sqrt{n}}$$

*holds true with probability at least $1 - \delta$.*

**Proof of Lemma 1.**

Let us start by proving (12) by introducing the map $\xi : \mathcal{X} \to \mathcal{L}_2(\mathcal{H})$ as $\xi(x) = K_x K_x^* - T_{q_{\mathcal{X}}}$. From (10) and (11) it follows that

$$\|\xi(x)\|_{\mathcal{L}_2(\mathcal{H})} \le \|K_x K_x^*\|_{\mathcal{L}_2(\mathcal{H})} + \|T_{q_{\mathcal{X}}}\|_{\mathcal{L}_2(\mathcal{H})} \le 2\kappa.$$

Moreover, we have

$$\int_{\mathcal{X}} \xi(x) dq_{\mathcal{X}}(x) = \int_{\mathcal{X}} K_x K_x^* dq_{\mathcal{X}}(x) - T_{q_{\mathcal{X}}} = 0.$$

Therefore, for $x'_j, j = 1, 2, \dots, m$, drawn i.i.d from the marginal probability measure $q_{\mathcal{X}}$, the corresponding operators $\xi_j = \xi(x'_j)$ can be treated as zero mean independent random variables in $\mathcal{L}_2(\mathcal{H})$, such that the condition of Concentration lemma are satisfied with $D = 2\kappa$, and

$$\|T_{\mathbf{x}'} - T_{q_{\mathcal{X}}}\|_{\mathcal{L}_2(\mathcal{H})} = \left\| \frac{1}{m} \sum_{j=1}^{m} K_{x'_j} K_{x'_j}^* - T_{q_{\mathcal{X}}} \right\|_{\mathcal{L}_2(\mathcal{H})} = \left\| \frac{1}{m} \sum_{j=1}^{m} \xi_j \right\|_{\mathcal{L}_2(\mathcal{H})} \le \frac{2\kappa \sqrt{2 \log \frac{2}{\delta}}}{\sqrt{m}}.$$

To obtain (13), for any $f \in \mathcal{H}$ we define a map $\phi = \phi_f : \mathcal{X} \to \mathcal{H}$ as $\phi_f(x) = K_x K_x^* f$. It clear that

$$\|\phi_f(x)\|_{\mathcal{H}} = \|K_x K_x^*\|_{\mathcal{L}(\mathcal{H})} \|f\|_{\mathcal{H}} \le \kappa \|f\|_{\mathcal{H}}.$$

Therefore, for the map $\phi = \phi_f$ the condition of the above Lemma 2 is satisfied with $R = \kappa \|f\|_{\mathcal{H}}$. Then directly from that lemma for any $f \in \mathcal{H}$ we have

$$
\begin{aligned}
\|T_{\mathbf{x}'} f - T_{\mathbf{x},\beta} f\|_{\mathcal{H}} &= \left\| \frac{1}{m} \sum_{j=1}^{m} \phi_f(x'_j) - \frac{1}{n} \sum_{i=1}^{n} \beta(x_i) \phi_f(x_i) \right\|_{\mathcal{H}} \\
&\le \left( 1 + \sqrt{2 \log \frac{2}{\delta}} \right) \left( \sqrt{\frac{B^2}{n} + \frac{1}{m}} \right) \kappa \|f\|_{\mathcal{H}} \\
&\le C \left( \log^{\frac{1}{2}} \frac{1}{\delta} \right) \left( m^{-\frac{1}{2}} + n^{-\frac{1}{2}} \right) \|f\|_{\mathcal{H}},
\end{aligned}
$$

that proves (13).

Consider now the map $F : \mathcal{X} \times \mathcal{Y} \to \mathcal{H}$ defined by

$$F(x, y) = \beta(x) K_x (f_p(x) - y).$$

Recall that $\|K_x\|_{\mathcal{L}(\mathcal{Y},\mathcal{H})} \le \sqrt{Tr(K_x^* K_x)} \le \sqrt{\kappa}$. Then we obtain:

$$\|F(x, y)\|_{\mathcal{H}} \le \|K_x\|_{\mathcal{L}(\mathcal{Y},\mathcal{H})} \left\| \int_{\mathcal{Y}} y' dp(y'|x) - y \right\|_{\mathcal{Y}} |\beta(x)| \le 2 y_0 B \sqrt{\kappa}.$$

Moreover, for $p(x, y) = p(y|x) p_{\mathcal{X}}(x)$ we have

$$\int_{\mathcal{X} \times \mathcal{Y}} F(x, y) dp(x, y) = \int_{\mathcal{X}} K_x \beta(x) \int_{\mathcal{Y}} \left( \int_{\mathcal{Y}} y' dp(y'|x) - y \right) dp(y|x) dp_{\mathcal{X}}(x) = 0,$$

such that for $(x_i, y_i)$, $i = 1, 2, \dots, n$, drawn i.i.d from the measure $p(x, y)$ the corresponding values $F_i = F(x_i, y_i)$ are zero mean independent random variables in $\mathcal{H}$.
Then for the just defined $F_i = \beta(x_i) K_{x_i} (f_q(x_i) - y_i)$ the conditions of Lemma 3 are satisfied with $D = 2 y_0 B \sqrt{\kappa}$, such that

$$
\begin{aligned}
\left\| \frac{1}{n} \sum_{i=1}^{n} F_i \right\|_{\mathcal{H}} &= \left\| \frac{1}{n} \sum_{i=1}^{n} \beta(x_i) K_{x_i} (f_q(x_i) - y_i) \right\|_{\mathcal{H}} \\
&= \left\| \sum_{i=1}^{n} \beta(x_i) K_{x_i} K_{x_i}^* f_q - \sum_{i=1}^{n} \beta(x_i) K_{x_i} y_i \right\|_{\mathcal{H}} \\
&= \|T_{\mathbf{x},\beta} f_q - g_{\mathbf{x},\mathbf{y},\beta}\|_{\mathcal{H}} \le \frac{2 y_0 B \sqrt{\kappa} \sqrt{2 \log \frac{2}{\delta}}}{\sqrt{n}}.
\end{aligned}
$$

This bound gives us (14).

**Aggregation for vector-valued functions**   Next we construct a new approximant in the form of a linear combination of approximants $f_1, f_2, \ldots, f_l$, computed for all tried parameter values. The linear combination of the approximants is computed as

$$f = \sum_{k=1}^{l} c_k f_k. \tag{15}$$

Since $f_1, f_2, \ldots, f_l$ belong to RKHS $\mathcal{H}$, it is clear that $f \in \mathcal{H}$. Now we want to argue on how close we can get to $f_q$. Following Proposition 1 in Caponnetto & De Vito (2007), we have

$$\mathcal{R}_q(f) - \mathcal{R}_q(f_q) = \|f - f_q\|_{L^2(q_{\mathcal{X}})}^2 = \left\| \sqrt{T_{q_{\mathcal{X}}}}(f - f_q) \right\|_{\mathcal{H}}^2. \tag{16}$$

Next we observe that the best approximation $f^*$ of the target regression function $f_q$ by linear combinations corresponds to the vector $c^* = (c_1^*, \ldots, c_l^*)$ of ideal coefficients in (15) that solves the linear system $Gc^* = \bar{g}$ with the Gram matrix $G = \left( \left\langle \sqrt{T_{q_{\mathcal{X}}}} f_k, \sqrt{T_{q_{\mathcal{X}}}} f_u \right\rangle_{\mathcal{H}} \right)_{k,u=1}^l$ and the right-hand side vector $\bar{g} = \left( \left\langle \sqrt{T_{q_{\mathcal{X}}}} f_q, \sqrt{T_{q_{\mathcal{X}}}} f_k \right\rangle_{\mathcal{H}} \right)_{k=1}^l$. Let us provide a prove of this short observation in the next lemma. Note that the entries $G$ and $g$ can equivalently also be formulated in terms of $\langle ., . \rangle_{L^2(q_{\mathcal{X}})}$, as done in the main text. We are going to use this formulation in the next lemma in order to be compatible with the main text (switching to the inner products in terms of $\mathcal{H}$ would not change the argument of the proof at all):

**Lemma 4.** *The best $L^2(q_{\mathcal{X}})$-approximation $f^*$ of the target regression function $f_q$ by linear combinations corresponds to the vector $c^* = (c_1^*, \ldots, c_l^*) = G^{-1}\bar{g}$.*

*Proof.* Let us denote (16) by $f(c)$ and rewrite this expression appropriately:

$$f(c) = \sum_{i,j=1}^{l} c_i c_j \langle f_i, f_j \rangle_{L^2(q_{\mathcal{X}})} - 2 \sum_{i=1}^{l} c_i \langle f_i, f_q \rangle_{L^2(q_{\mathcal{X}})} + \langle f_q, f_q \rangle_{L^2(q_{\mathcal{X}})}.$$

Taking the derivative with respect to $c_i$ yield:

$$\frac{\partial f(c)}{\partial c_i} = 2 \left( \sum_{j=1}^{l} c_j \langle f_i, f_j \rangle_{L^2(q_{\mathcal{X}})} - \langle f_i, f_q \rangle_{L^2(q_{\mathcal{X}})} \right).$$

Setting these derivatives to zero (for all $i \in \{1, \ldots, l\}$) gives the claimed equation. Noting that the Hessian is equal to $2G$ (and thus positive-definite) ensures that $c^*$ is a global minimum of $f$.   □

But, of course, neither Gram matrix $G$ nor the vector $\bar{g}$ is accessible, because there is no access to the target measure $q_{\mathcal{X}}$, so we switch to the empirical counterparts $\widetilde{G}$ and $\widetilde{g}$.

Then the following lemma is helpful to gain some information on the error made by the empirical average:

**Lemma 5.** *With probability $1 - \delta$ we have*

$$\left| \left\langle \sqrt{T_{q_{\mathcal{X}}}} f_u, \sqrt{T_{q_{\mathcal{X}}}} f_k \right\rangle_{\mathcal{H}} - \frac{1}{m} \sum_{j=1}^{m} \langle f_k(x_j'), f_u(x_j') \rangle_{\mathcal{Y}} \right| \leq C \left( \log^{\frac{1}{2}} \frac{1}{\delta} \right) m^{-\frac{1}{2}},$$

$$\left| \left\langle \sqrt{T_{q_{\mathcal{X}}}} f_k, \sqrt{T_{q_{\mathcal{X}}}} f_q \right\rangle_{\mathcal{H}} - \frac{1}{n} \sum_{i=1}^{n} \beta(x_i) \langle f_k(x_i), y_i \rangle_{\mathcal{Y}} \right| \leq C \left( \log^{\frac{1}{2}} \frac{1}{\delta} \right) (n^{-\frac{1}{2}} + m^{-\frac{1}{2}}),$$

*where $C > 0$ does not depend on $n, m$ and $\delta$.*

*Proof.* Keeping in mind that $f_q, f_k \in \mathcal{H}$ we have

$$
\begin{aligned}
\left\langle \sqrt{T_{q\mathcal{X}}} f_u, \sqrt{T_{q\mathcal{X}}} f_k \right\rangle_{\mathcal{H}} &= \langle T_{\mathbf{x}'} f_k, f_u \rangle_{\mathcal{H}} + \langle (T_{q\mathcal{X}} - T_{\mathbf{x}'}) f_u, f_k \rangle_{\mathcal{H}} \\
&= \left\langle \frac{1}{m} \sum_{j=1}^{m} K_{x'_j} K_{x'_j}^* f_k, f_u \right\rangle_{\mathcal{H}} + \langle (T_{q\mathcal{X}} - T_{\mathbf{x}'}) f_u, f_k \rangle_{\mathcal{H}} \\
&= \frac{1}{m} \sum_{j=1}^{m} \left\langle K_{x'_j}^* f_k, K_{x'_j}^* f_u \right\rangle_{\mathcal{Y}} + \langle (T_{q\mathcal{X}} - T_{\mathbf{x}'}) f_u, f_k \rangle_{\mathcal{H}} \\
&= \frac{1}{m} \sum_{j=1}^{m} \langle f_k(x'_j), f_u(x'_j) \rangle_{\mathcal{Y}} + \langle (T_{q\mathcal{X}} - T_{\mathbf{x}'}) f_u, f_k \rangle_{\mathcal{H}}.
\end{aligned}
$$

Moreover, from (12) with probability $1 - \delta$ we have that

$$
\left| \langle (T_{q\mathcal{X}} - T_{\mathbf{x}'}) f_u, f_k \rangle_{\mathcal{H}} \right| \leq C \|f_u\|_{\mathcal{H}} \|f_k\|_{\mathcal{H}} \left( \log^{\frac{1}{2}} \frac{1}{\delta} \right) m^{-\frac{1}{2}}.
$$

Then

$$
\left| \left\langle \sqrt{T_{q\mathcal{X}}} f_u, \sqrt{T_{q\mathcal{X}}} f_k \right\rangle_{\mathcal{H}} - \frac{1}{m} \sum_{j=1}^{m} \langle f_k(x'_j), f_u(x'_j) \rangle_{\mathcal{Y}} \right| \leq C \left( \log^{\frac{1}{2}} \frac{1}{\delta} \right) m^{-\frac{1}{2}}.
$$

Now, we prove the second statement in Lemma 5. We have

$$
\begin{aligned}
\left\langle \sqrt{T_{q\mathcal{X}}} f_k, \sqrt{T_{q\mathcal{X}}} f_q \right\rangle_{\mathcal{H}} &= \langle f_k, T_{q\mathcal{X}} f_q \rangle_{\mathcal{H}} = \langle f_k, T_{q\mathcal{X}} f_q - g_{\mathbf{x},\mathbf{y},\beta} \rangle_{\mathcal{H}} + \langle f_k, g_{\mathbf{x},\mathbf{y},\beta} \rangle_{\mathcal{H}} \\
&= \frac{1}{n} \sum_{i=1}^{n} \beta(x_i) \langle f_k, K_{x_i} y_i \rangle_{\mathcal{H}} + \langle f_k, T_{q\mathcal{X}} f_q - g_{\mathbf{x},\mathbf{y},\beta} \rangle_{\mathcal{H}} \\
&= \frac{1}{n} \sum_{i=1}^{n} \beta(x_i) \langle K_{x_i}^* f_k, y_i \rangle_{\mathcal{Y}} + \langle f_k, T_{q\mathcal{X}} f_q - g_{\mathbf{x},\mathbf{y},\beta} \rangle_{\mathcal{H}} \\
&= \frac{1}{n} \sum_{i=1}^{n} \beta(x_i) \langle f_k(x_i), y_i \rangle_{\mathcal{Y}} + \langle f_k, T_{q\mathcal{X}} f_q - g_{\mathbf{x},\mathbf{y},\beta} \rangle_{\mathcal{H}}.
\end{aligned}
$$

From Lemma 1, with probability $1 - \delta$ we have

$$
\begin{aligned}
&\|T_{q\mathcal{X}} f_q - g_{\mathbf{x},\mathbf{y},\beta}\|_{\mathcal{H}} \\
&\leq \|T_{q\mathcal{X}} f_q - T_{\mathbf{x}'} f_q\|_{\mathcal{H}} + \|T_{\mathbf{x}'} f_q - g_{\mathbf{x},\mathbf{y},\beta}\|_{\mathcal{H}} \\
&\leq \|T_{q\mathcal{X}} f_q - T_{\mathbf{x}'} f_q\|_{\mathcal{H}} + \|T_{\mathbf{x}'} f_q - T_{\mathbf{x},\beta} f_q\|_{\mathcal{H}} + \|T_{\mathbf{x},\beta} f_q - g_{\mathbf{x},\mathbf{y},\beta}\|_{\mathcal{H}} \\
&\leq C \|f_q\|_{\mathcal{H}} \left( \log^{\frac{1}{2}} \frac{1}{\delta} \right) m^{-\frac{1}{2}} + C \|f_q\|_{\mathcal{H}} \left( \log^{\frac{1}{2}} \frac{1}{\delta} \right) (n^{-\frac{1}{2}} + m^{-\frac{1}{2}}) + \|T_{\mathbf{x},\beta} f_q - g_{\mathbf{x},\mathbf{y},\beta}\|_{\mathcal{H}} \\
&\leq C \|f_q\|_{\mathcal{H}} \left( \log^{\frac{1}{2}} \frac{1}{\delta} \right) m^{-\frac{1}{2}} + C \|f_q\|_{\mathcal{H}} \left( \log^{\frac{1}{2}} \frac{1}{\delta} \right) (n^{-\frac{1}{2}} + m^{-\frac{1}{2}}) + C \left( \log^{\frac{1}{2}} \frac{1}{\delta} \right) n^{-\frac{1}{2}}.
\end{aligned}
$$

Then

$$
\langle f_k, T_{q\mathcal{X}} f_q - g_{\mathbf{x},\mathbf{y},\beta} \rangle_{\mathcal{H}} \leq C \|f_k\|_{\mathcal{H}} \left( \log^{\frac{1}{2}} \frac{1}{\delta} \right) \left( n^{-\frac{1}{2}} + m^{-\frac{1}{2}} \right).
$$

Therefore,

$$
\left| \left\langle \sqrt{T_{q\mathcal{X}}} f_k, \sqrt{T_{q\mathcal{X}}} f_q \right\rangle_{\mathcal{H}} - \frac{1}{n} \sum_{i=1}^{n} \beta(x_i) \langle f_k(x_i), y_i \rangle_{\mathcal{Y}} \right| \leq C \left( \log^{\frac{1}{2}} \frac{1}{\delta} \right) (n^{-\frac{1}{2}} + m^{-\frac{1}{2}}).
$$

$\square$

**Towards our main generalization bound**  Next we use similar arguments as in Theorem 4 from Gizewski et al. (2022) to obtain our main result, Theorem 1. Lemma 5 suggests to approximate $G$ and $\bar{g}$ by their empirical counterparts:

$$\tilde{G} = \left( \frac{1}{m} \sum_{j=1}^{m} \left\langle f_k(x_j'), f_u(x_j') \right\rangle_{\mathcal{Y}} \right)_{k,u=1}^{l}, \tag{17}$$

$$\tilde{g} = \left( \frac{1}{n} \sum_{i=1}^{n} \beta(x_i) \left\langle y_i, f_k(x_i) \right\rangle_{\mathcal{Y}} \right)_{k=1}^{l} \tag{18}$$

which can be effectively computed from data samples. Moreover, again from Lemma 5 we can argue that with probability $1 - \delta$ it holds:

$$\|\bar{g} - \tilde{g}\|_{\mathbb{R}^l} \leq C \left( \log^{\frac{1}{2}} \frac{1}{\delta} \right) (n^{-\frac{1}{2}} + m^{-\frac{1}{2}}), \tag{19}$$

$$\|G - \tilde{G}\|_{\mathcal{L}(\mathbb{R}^l)} \leq C \left( \log^{\frac{1}{2}} \frac{1}{\delta} \right) m^{-\frac{1}{2}}. \tag{20}$$

With the matrix $\tilde{G}$ at hand one can easily check whether or not it is well-conditioned and $\tilde{G}^{-1}$ exists (otherwise one needs to get rid of models with similar performance). Thus the norms $\left\| \tilde{G} \right\|_{\mathcal{L}(\mathbb{R}^l)}$ and $\left\| \tilde{G}^{-1} \right\|_{\mathcal{L}(\mathbb{R}^l)}$ can be bounded independently of $m$ and $n$, due to the fact that all their entries can be bounded as follows (we only do the calculation for the entries of $\tilde{G}$):

$$|\tilde{G}_{k,u}| \leq \frac{1}{m} \sum_{j=1}^{m} \left| \left\langle f_k(x_j'), f_u(x_j') \right\rangle_{\mathcal{Y}} \right| = \frac{1}{m} \sum_{j=1}^{m} \left| \left\langle K_{x_j'}^* f_k, K_{x_j'}^* f_u \right\rangle_{\mathcal{Y}} \right|$$

$$= \frac{1}{m} \sum_{j=1}^{m} \left| \left\langle K_{x_j'} K_{x_j'}^* f_k, f_u \right\rangle_{\mathcal{H}} \right| = \frac{1}{m} \sum_{j=1}^{m} \left| \left\langle T_{x_j'} f_k, f_u \right\rangle_{\mathcal{H}} \right|$$

$$\leq \frac{1}{m} \sum_{j=1}^{m} \left\| T_{x_j'} \right\|_{\mathcal{L}(\mathcal{H})} \|f_k\|_{\mathcal{H}} \|f_u\|_{\mathcal{H}} \leq \kappa \gamma_l^2,$$

where we used the reproducing property (5) to obtain the equality in the first line and (10) for the last inequality. Now assume that $m$ is so large that with probability $1 - \delta$ we have

$$\|G - \tilde{G}\|_{\mathcal{L}(\mathbb{R}^l)} < \frac{1}{\left\| \tilde{G}^{-1} \right\|_{\mathcal{L}(\mathbb{R}^l)}}. \tag{21}$$

Moreover we can use the following simple manipulation:

$$G^{-1} = \tilde{G}^{-1} (G \tilde{G}^{-1})^{-1} = \tilde{G}^{-1} (I - (I - G \tilde{G}^{-1}))^{-1} = \tilde{G}^{-1} (I - (\tilde{G} - G) \tilde{G}^{-1})^{-1}.$$

Then (21) ensures that the Neumann series for $(I - (\tilde{G} - G) \tilde{G}^{-1})^{-1}$ converges and we obtain the following bound:

$$\left\| G^{-1} \right\|_{\mathcal{L}(\mathbb{R}^l)} \leq \frac{\left\| \tilde{G}^{-1} \right\|_{\mathcal{L}(\mathbb{R}^l)}}{1 - \left\| \tilde{G}^{-1} \right\|_{\mathcal{L}(\mathbb{R}^l)} \left\| G - \tilde{G} \right\|_{\mathcal{L}(\mathbb{R}^l)}} = O(1). \tag{22}$$

Now we are in the position to prove our main generalization bound (4) for unsupervised domain adaptation:

*Proof of Theorem 1.*  We have already discussed that the coefficients in the best approximation $f^*$ to $f_q$ are given by $c^* = (c_1^*, c_2^*, \ldots, c_l^*) = G^{-1} \bar{g}$. Since:

$$G^{-1}(\tilde{g} - \bar{g}) + G^{-1}(G - \tilde{G})\tilde{c} = G^{-1}\tilde{g} - c^* + \tilde{c} - G^{-1}\tilde{g} = \tilde{c} - c^*$$

then from (19)–(22) with probability $1 - \delta$ we have

$$
\begin{aligned}
\|\tilde{c} - c^*\|_{\mathbb{R}^l} = &\leq \|G^{-1}\|_{\mathcal{L}(\mathbb{R}^l)} \left( \|\tilde{g} - \bar{g}\|_{\mathbb{R}^l} + \|G - \tilde{G}\|_{\mathcal{L}(\mathbb{R}^l)} \|\tilde{c}\|_{\mathbb{R}^l} \right) \\
&\leq C \left( \log^{\frac{1}{2}} \frac{1}{\delta} \right) (n^{-\frac{1}{2}} + m^{-\frac{1}{2}}).
\end{aligned}
\tag{23}
$$

Moreover:

$$
\begin{aligned}
\mathcal{R}_q(\tilde{f}) - \mathcal{R}_q(f_q) &= \left\| \sqrt{T_{q_x}}(\tilde{f} - f_q) \right\|_{\mathcal{H}}^2 \\
&\leq \left( \left\| \sqrt{T_{q_x}}(f^* - f_q) \right\|_{\mathcal{H}} + \left\| \sqrt{T_{q_x}}(\tilde{f} - f^*) \right\|_{\mathcal{H}} \right)^2 \\
&\leq 2 \left\| \sqrt{T_{q_x}}(f^* - f_q) \right\|_{\mathcal{H}}^2 + 2 \left\| \sqrt{T_{q_x}}(\tilde{f} - f^*) \right\|_{\mathcal{H}}^2 \\
&= 2 \left( \mathcal{R}_q(f^*) - \mathcal{R}_q(f_q) \right) + 2 \left\| \sqrt{T_{q_x}}(\tilde{f} - f^*) \right\|_{\mathcal{H}}^2 \\
&\leq 2 \left( \mathcal{R}_q(f^*) - \mathcal{R}_q(f_q) \right) + 2 \left( \sum_{k=1}^l |c_k^* - \tilde{c}_k| \left\| \sqrt{T_{q_x}} f_k \right\|_{\mathcal{H}} \right)^2 \\
&\leq 2 \left( \mathcal{R}_q(f^*) - \mathcal{R}_q(f_q) \right) + 2l \|c^* - \tilde{c}\|_{\mathbb{R}^l}^2 \max_k \left\| \sqrt{T_{q_x}} f_k \right\|_{\mathcal{H}}^2 \\
&\leq 2 \left( \mathcal{R}_q(f^*) - \mathcal{R}_q(f_q) \right) + 2 \left\| \sqrt{T_{q_x}} \right\|_{\mathcal{L}(\mathcal{H})}^2 l \gamma_l^2 \|c^* - \tilde{c}\|_{\mathbb{R}^l}^2,
\end{aligned}
\tag{24}
$$

The statement of the theorem follows now from (23)–(24) (using again the inequality $(a + b)^2 \leq 2(a^2 + b^2)$).  □

**On the dependence of the error bound on the number $l$ of models**  An interesting question is, how the bound in Eq. (4) depends on $l$. To this end, let us have a look at the second term in the last line in Eq. (24): $\left\| \sqrt{T_{q_x}} \right\|_{\mathcal{L}}$ analyzes a sampling operator, thus does not depend on the number of models, same goes for $\gamma_l$, which is just a uniform bound on all our models. To analyze $\|c^* - \tilde{c}\|_{\mathbb{R}^l}^2$, let us have a look at the individual factors in the first inequality of Eq. (23). To not overload notation, $C > 0$ is used here for any absolute constant that is independent of $l, m, n$ and $\delta$.

- $\|G - \tilde{G}\|_{\mathcal{L}(\mathbb{R}^l)}$: The individual entries of $G - \tilde{G}$ are (in absolute values) bounded by Lemma 5. The proof arguments only involve norm bounds on the associated sampling operators, the uniform bound $\gamma_l$ on all the models and the bound $B$ on $\beta$, thus the absolute constant $C$ there is independent of $l$. By the definition of the matrix Frobenius norm, we thus get

$$
\|G - \tilde{G}\|_{\mathcal{L}(\mathbb{R}^l)} \leq Cl \left( \log^{\frac{1}{2}} \frac{1}{\delta} \right) (n^{-\frac{1}{2}} + m^{-\frac{1}{2}})
\tag{25}
$$

- $\|\tilde{g} - \bar{g}\|_{\mathbb{R}^l}$: Similar arguments as before lead to $\|\tilde{g} - \bar{g}\|_{\mathcal{L}(\mathbb{R}^l)} \leq C\sqrt{l} \left( \log^{\frac{1}{2}} \frac{1}{\delta} \right) (n^{-\frac{1}{2}} + m^{-\frac{1}{2}})$

- $\|G^{-1}\|_{\mathcal{L}(\mathbb{R}^l)}$: It is natural to assume that there is some constant $c \geq \left\| \tilde{G}^{-1} \right\|_{\mathcal{L}(\mathbb{R}^l)}$. Otherwise, we can, e.g., orthogonalize our models and coefficients without changing the aggregation, but with reducing the conditioning number (i.e., with reducing $\left\| \tilde{G}^{-1} \right\|$). It is also natural to assume that $m$ and $n$ are large enough such that $l(n^{-\frac{1}{2}} + m^{-\frac{1}{2}}) < \frac{1}{2c}$. Then applying Eq. (25) to Eq. (22), we can deduce that $\left\| G^{-1} \right\|_{\mathcal{L}(\mathbb{R}^l)} \leq 2c$.

- $\|\tilde{c}\|_{\mathbb{R}^l}$: This quantity can also be assumed to be known independently of $l$, since it is given by our data only.

Combining the previous points gives us $\|c^* - \tilde{c}\|_{\mathbb{R}^l}^2 \leq Cl^2 \left(\log \frac{1}{\delta}\right) (n^{-1} + m^{-1})$ which finally leads to the refined bound

$$\mathcal{R}_q(\tilde{f}) - \mathcal{R}_q(f_q) \leq 2\left(\mathcal{R}_q\left(f^*\right) - \mathcal{R}_q(f_q)\right) + Cl^3 \left(\log \frac{1}{\delta}\right) (n^{-1} + m^{-1})$$

for sufficiently large $l$, $m$ and $n$ and error probability $\delta > 0$.

## B  CONSTRUCTION OF FUNCTION SPACES

Let us give a short discussion on the construction of our required function space mentioned in the previous Section A, the reproducing kernel space $\mathcal{H}$. As mentioned already in the main text, the explicit knowledge of $\mathcal{H}$ is not required, we just need to rely on its existence. First, any of our models $f$ can be regarded as an element of some reproducing kernel space (RKHS) $\tilde{\mathcal{H}}$ satisfying the assumptions 1. This is immediate if $f : \mathcal{X} \to \mathbb{R}$ is a real valued continuous function and we take $k(x, y) = f(x)f(y)$ as the associated reproducing kernel. In the case $f : \mathcal{X} \to \mathcal{Y}$ and $\mathcal{Y}$ is finite dimensional, it is not hard to see that a similar construction is possible, as this case can again be boiled down to the construction of a kernel with real-valued output, see e.g. Remark 1 in Caponnetto & De Vito (2007) for details.

Overall we end up with a finite sequence of spaces $(\mathcal{H}_k)_{k=1}^{l+1}$ of functions living on the same domain $\mathcal{X}$ (we have $l + 1$ as we also take into account the regression function), and the existence of a RKHS containing all given models and the regression function is not a real restriction. For example, in case of real valued functions, this assumption is automatically satisfied, as linear combinations of functions with the same domain which stem from a finite sequence of RKHSs belong to an RKHS. This follows from a classical result by N. Aronszajn and R. Godement, see e.g. Pereverzyev (2022, Theorem 1.4.).

There is also ongoing research on constructing function spaces (and especially associated reproducing kernels) for families of neural networks that are used in applications, see e.g. Ma & Wu (2022) for ReLU networks, Fermanian et al. (2021) for recurrent networks and Bietti & Mairal (2017; 2019) for convolutional neural networks. Incorporating these into our work may lead to refined generalization bounds that also reflect the nature of our models. We leave the details open for future work.

## C  DATASETS

This section provides an overview over all applied datasets from language, image, and time-series domains.

**Illustrative example:**  For the illustrative example (Figure 1 in the main paper) we rely on the following setting, taken from Shimodaira (2000); Sugiyama et al. (2007); You et al. (2019): The data points are labelled with $y = \frac{\sin(\pi x)}{\pi x}$ with random noise sampled from the normal distribution $\mathcal{N}\left(0, \left(\frac{1}{4}\right)^2\right)$. Moreover $p_{\mathcal{X}} \sim \mathcal{N}\left(1, \frac{1}{4}\right)$, $q_{\mathcal{X}} \sim \mathcal{N}\left(2, \left(\frac{1}{4}\right)^2\right)$. The density ratio $\beta$ can be computed analytically and is bounded. We aggregate several linear models with our approach and compare it to the optimal linear model, whose coefficients have been evaluated using a computer algebra system.

**Academic Dataset**  We rely on the Transformed Moons dataset (Zellinger et al., 2021), allowing us to visualize and address low-dimensional input data. The dataset consists of two-dimensional input data points forming two classes with a "moon-shaped" support. The shift from source to target domain is simulated by a transformation in input space as depicted in Figure 3. The results are shown in the following table:

**Language Dataset**  To evaluate our method on a language task, we rely on the Amazon Reviews (Blitzer et al., 2006) dataset. This dataset consists of text reviews from four domains: books (B), DVDs (D), electronics (E), and kitchen appliances (K). Reviews are encoded in 5000 dimensional feature vectors of bag-of-words unigrams and bigrams with binary labels: label 0 if the product is ranked by 1 to 3 stars, and label 1 if the product is ranked by 4 or 5 stars. From the four categories, we obtain twelve domain adaptation tasks, where each category serves once as source domain and once

Table 2: Mean and standard deviation (after ±) of target classification accuracy on Transformed Moons dataset over three different random initialization of model weights and 11 domain adaptation methods.

| | | | | Transformed Moons | | | | | |
| | | | Heuristic | | | | Theoretical error guarantees | | |
| Method | SO | TMV | TMR | TCR | SOR | IWV | DEV | IWA (ours) | TB |
|---|---|---|---|---|---|---|---|---|---|
| HoMM | 0.994(±0.003) | 0.994(±0.000) | *1.000(±0.003)* | 1.000(±0.003) | 0.985(±0.012) | 0.994(±0.003) | **0.994(±0.003)** | 0.992(±0.000) | 0.998(±0.000) |
| AdvSKM | 0.985(±0.000) | *1.000(±0.000)* | 0.998(±0.000) | 0.998(±0.002) | 0.987(±0.014) | 0.985(±0.000) | 0.985(±0.000) | **1.000(±0.000)** | 1.000(±0.000) |
| DIRT | 0.964(±0.009) | 0.827(±0.018) | 0.830(±0.017) | 0.981(±0.003) | *0.990(±0.006)* | 0.964(±0.009) | 0.869(±0.009) | **0.983(±0.003)** | 0.966(±0.000) |
| DDC | 0.994(±0.023) | *0.998(±0.000)* | 0.996(±0.000) | 0.996(±0.000) | 0.981(±0.009) | 0.994(±0.023) | 0.994(±0.023) | **0.998(±0.000)** | 0.998(±0.000) |
| CMD | 0.990(±0.006) | 0.956(±0.000) | 0.964(±0.000) | *0.996(±0.000)* | 0.994(±0.006) | 0.990(±0.006) | 0.990(±0.006) | **0.996(±0.000)** | 1.000(±0.000) |
| MMDA | 0.990(±0.006) | *1.000(±0.003)* | 1.000(±0.003) | 1.000(±0.003) | 0.996(±0.009) | 0.990(±0.006) | 0.990(±0.006) | **1.000(±0.003)** | 1.000(±0.003) |
| CoDATS | 0.979(±0.016) | *1.000(±0.044)* | 1.000(±0.052) | 1.000(±0.016) | 0.981(±0.012) | 0.979(±0.016) | 0.979(±0.174) | **1.000(±0.010)** | 1.000(±0.045) |
| Deep-Coral | 1.000(±0.009) | *1.000(±0.000)* | 1.000(±0.000) | 1.000(±0.000) | 0.985(±0.000) | 1.000(±0.009) | 1.000(±0.009) | **1.000(±0.000)** | 1.000(±0.000) |
| CDAN | 1.000(±0.000) | *1.000(±0.000)* | 1.000(±0.000) | 1.000(±0.000) | 0.985(±0.026) | 1.000(±0.000) | 1.000(±0.000) | **1.000(±0.000)** | 1.000(±0.000) |
| DANN | 0.994(±0.006) | *1.000(±0.000)* | 1.000(±0.000) | 1.000(±0.000) | 0.994(±0.016) | 0.994(±0.006) | 0.994(±0.006) | **1.000(±0.003)** | 1.000(±0.003) |
| DSAN | 0.990(±0.009) | *1.000(±0.000)* | 1.000(±0.000) | 1.000(±0.000) | 1.000(±0.003) | 0.990(±0.009) | 0.990(±0.009) | **1.000(±0.000)** | 1.000(±0.000) |
| Avg. | 0.989(±0.008) | 0.980(±0.006) | 0.981(±0.007) | *0.997(±0.002)* | 0.989(±0.010) | 0.989(±0.008) | 0.981(±0.022) | **0.997(±0.002)** | 0.997(±0.005) |

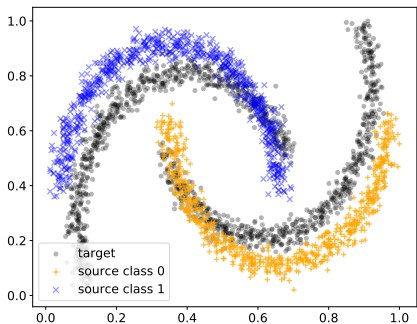

Figure 3: Transformed Moons dataset. Source data is depicted as blue + and orange ×. Target data points are shown as black dots.

as target domain (e.g., see Table 15). We follow similar data splits as previous works (Chen et al., 2012; Louizos et al., 2016; Ganin et al., 2016). In particular, we use 4000 labeled source examples and 4000 unlabeled target examples for training, and over 1000 examples for testing.

**Image Dataset** Our third dataset is MiniDomainNet, which is based on the DomainNet-2019 dataset (Peng et al., 2019) consisting of six different image domains (Quickdraw: Q, Real: R, Clipart: C, Sketch: S, Infograph: I, and Painting: P). We follow Zellinger et al. (2021) and rely on the reduced version of DomainNet-2019, referred to as MiniDomainNet, which reduces the number of classes to the top-five largest representatives in the training set across all six domains. To further improve computation time, we rely on a ImageNet (Krizhevsky et al., 2012) pre-trained ResNet-18 (He et al., 2016) backbone. Therefore, we assume that the backbone has learned lower-level filters suitable for the "Real" image category, and we only need to adapt to the remaining five domains (e.g., Clipart, Sketch). This results in five domain adaptation tasks.

**Time-Series Dataset** We based our time-series experiments on the four datasets included in the AdaTime benchmark suite (Ragab et al., 2022), which consists of UCI-HAR, WISDM, HHAR, and Sleep-EDF. The suite includes four representative datasets spanning 20 cross-domain real-world scenarios, i.e., human activity recognition and sleep stage classification. The first dataset is the *Human Activity Recognition* (HAR) (Anguita et al., 2013) dataset from the UC Irvine Repository denoted as UCI-HAR, which contains data from three motion sensors (accelerometer, gyroscope and body-worn sensors) gathered using smartphones from 30 different subjects. It classifies their activities in several categories, namely, walking, walking upstairs, downstairs, standing, sitting, and lying down. The WISDM (Kwapisz et al., 2011) dataset is a class-imbalanced variant from collected accelerometer sensors, including GPS data, from 29 different subjects which are performing similar activities as in the UCI-HAR dataset. The *Heterogeneity Human Activity Recognition* (HHAR) (Stisen et al., 2015) dataset investigate sensor-, device- and workload-specific heterogeneities using 36 smartphones and smartwatches, consisting of 13 different device models from four manufacturers. Finally, the *Sleep Stage Classification* time-series setting aims to classify the electroencephalography

(EEG) signals into five stages i.e., Wake (W), Non-Rapid Eye Movement stages (N1, N2, N3), and Rapid Eye Movement (REM). Analogous to Ragab et al. (2022); Eldele et al. (2021), we adopt the Sleep-EDF-20 dataset obtained from PhysioBank (Goldberger et al., 2000), which contains EEG readings from 20 healthy subjects. For all datasets, each subject is treated as an own domain, and adopt from a source subject to a target subject.

## D EXPERIMENTAL SETUP

This section is meant to provide further details on the overall computational setting of our experiments. We start by giving an overview on the used computational resources for the specific datasets and the implementation tools. Next, we describe the network architectures for the individual datasets in greater detail. In the third subsection we elaborate on the construction of our models, and the fourth subsection is devoted to matrix inversion. Finally, in the last subsection, we describe the detailed empirical results and give the complete tables.

### D.1 COMPUTATIONAL RESOURCES AND IMPLEMENTATIONS

Overall, to compute the results in our tables, we trained 16680 models with an approximate computational budget of 1500 GPU/hours on one high-performance computing station using $8\times$NVIDIA P100 16GB, 512GB RAM, 40 Cores Xeon(R) CPU E5-2698 v4 @ 2.20GHz on CentOS Linux 7.

Transformed Moons: 11 methods $\times$ 14 parameters $\times$ 1 domain adaptation tasks $\times$ 3 seeds $+$ 3 density estimator classifier $=$ 465 trained models

Amazon Reviews: 11 methods $\times$ 14 parameters $\times$ 12 domain adaptation tasks $\times$ 3 seeds $+$ 12 $\times$ 3 density estimator classifier $=$ 5580 trained models

MiniDomainNet: 11 methods $\times$ 8 parameters $\times$ 5 domain adaptation tasks $\times$ 3 seeds $+$ 5 $\times$ 3 density estimator classifier $=$ 1335 trained models

UCI-HAR: 11 methods $\times$ 14 parameters $\times$ 5 domain adaptation tasks $\times$ 3 seeds $+$ 5 $\times$ 3 density estimator classifier $=$ 2325 trained models

Sleep-EDF: 11 methods $\times$ 14 parameters $\times$ 5 domain adaptation tasks $\times$ 3 seeds $+$ 5 $\times$ 3 density estimator classifier $=$ 2325

HHAR: 11 methods $\times$ 14 parameters $\times$ 5 domain adaptation tasks $\times$ 3 seeds $+$ 5 $\times$ 3 density estimator classifier $=$ 2325 trained models

WISDM: 11 methods $\times$ 14 parameters $\times$ 5 domain adaptation tasks $\times$ 3 seeds $+$ 5 $\times$ 3 density estimator classifier $=$ 2325 trained models

In Total: $465 + 5580 + 1335 + 4 \times 2325 = 16680$ trained models

All methods have been implemented in Python using the *Pytorch* (Paszke et al., 2017, BSD license) library. For monitoring the runs we used Weights & Biases (Biewald, 2020, MIT license). We use *Scikit-learn* (Pedregosa et al., 2011) library for evaluation measures and toy datasets, and the *TQDM* (da Costa-Luis, 2019) library, and *Tensorboard* (Abadi et al., 2015) for keeping track of the progress of our experiments. We built parts of our implementation on the codebase of Zellinger et al. (2021, MIT License) and Ragab et al. (2022, MIT License).

### D.2 ARCHITECTURES AND TRAINING SETUP

In this subsection, we provide details on the model architectures and the training setup for every dataset. Our base architectures are based on the AdaTime benchmark suite, which is a large-scale evaluation of domain adaptation algorithms on time-series data. We extended the benchmark suite to support 11 state-of-the-art model architectures on multiple dataset types ranging from language, image to time-series data, addressed by Transformed Moons, Amazon Reviews, MiniDomainNet and the four time-series datasets (UCI-HAR, WISDM, HHAR, and Sleep-EDF) spanning in total 38 cross-domain real-world scenarios.

**Transformed Moons**  For the Transformed Moons dataset we use two sequential blocks with fully-connected layers, 1D-BatchNorm, ReLU activation functions and Dropout. The full architecture specification can be found in Table 3. The domain classifier (density ratio estimator) uses the same architecture. We train the class prediction models for 50 epochs and the domain classifier for 80 epochs with learning rate 0.001, weight decay 0.0001 and batchsize 128 using the Adam optimizer (Kingma & Ba, 2014). We share the same base architecture and training setup across every domain adaption method (e.g., DANN, HoMM, CMD). Additional hyper-parameters are reported in Table 9.

Table 3: Model architecture for the Transformed Moons dataset. The values for neural network layers correspond to the number of output units.

| | Architecture | |
| --- | --- | --- |
| | **Layers** | Values |
| | Input units | 2 |
| MLP Block 1 | Fully-connected Layer | 128 |
| | Batch Normalization 1D Layer | 128 |
| | ReLU Activation | |
| | Dropout | |
| MLP Block 2 | Fully-connected Layer | 128 |
| | Batch Normalization 1D Layer | 128 |
| | ReLU Activation | |
| | Dropout | |
| | Fully-connected Layer | 128 |
| | **Methods** | |
| | See Table 7 and Table 8 for details. | |

**Amazon Reviews**  For the Amazon Reviews dataset we use two sequential blocks with fully-connected layers, 1D-BatchNorm, ReLU activation function and Dropout, analogous to the setup for Transformed Moons. We also use the same architecture for the domain classifier. We train the class prediction models for 50 epochs and the domain classifier for 80 epochs with learning rate 0.001, weight decay 0.0001 and batchsize 128 using the Adam optimizer (Kingma & Ba, 2014). We share the same base architecture and training setup across every domain adaption method (e.g., DANN, HoMM, CMD). Additional hyper-parameters are reported in Table 9.

Table 4: Model architecture for the Amazon Reviews dataset. The values for neural network layers correspond to the number of output units.

| | Architecture | |
| --- | --- | --- |
| | **Layers** | Values |
| | Input units | 5000 |
| MLP Block 1 | Fully-connected Layer | 128 |
| | Batch Normalization 1D Layer | 128 |
| | ReLU Activation | |
| | Dropout | |
| MLP Block 2 | Fully-connected Layer | 128 |
| | Batch Normalization 1D Layer | 128 |
| | ReLU Activation | |
| | Dropout | |
| | Fully-connected Layer | 128 |
| | **Methods** | |
| | See Table 7 and Table 8 for details. | |

**MiniDomainNet**  Following the pre-trained setup from Peng et al. (2019), we use a frozen ResNet-18 backbone model which was trained on ImageNet, and operate subsequent computations on the

512 dimensional extracted features. To alleviate overfitting effects on pre-computed features, we perform data augmentation on the images of each batch and forward each batch through the backbone. We incorporate zero padding before resizing the images to 256x256 to avoid image distortions. Furthermore, in alignment with data augmentation techniques from Shorten & Khoshgoftaar (2019), we perform random resized cropping to 224x224 with a random viewport between 70% and 100% of the original image, random horizontal flipping, color jittering of 0.25% on each RGB channel, and a $\pm 2$ degree rotation.

After the ResNet-18 backbone output, we add a projection layer, and define the domain adaptation layers on which we use the domain adaptation methods to align the representations. The backbone and projection layers are defined as a common architecture across the different domain adaptation methods. Additional layers are further added for the classification networks, according to the requirements of the individual domain adaptation methods (e.g., CMD, HoMM). The number of layers/neurons in the upper layers of our architecture have been tuned in order to achieve the best performance in the source-only setup. See Table 5 for a detailed description of the architecture used. We perform experiments on all 5 domain adaptation tasks as defined in Section C for each of the previously listed methods, and with 3 repetitions based on different random weights initialization. All class prediction models have been trained for 60 epochs and domain classifiers for 100 epochs with Adam optimizer, a learning rate of 0.001, $\beta_1 = 0.9$, $\beta_2 = 0.999$, batchsize of 128 and weight decay of 0.0001. Additional hyper-parameters are reported in Table 9.

Table 5: Model architecture for the MiniDomainNet dataset. The values for neural network layers correspond to the number of output units.

| | Architecture | |
|---|---|---|
| | **Layers** | Values |
| Backbone Output Layer | ResNet-18 (Adaptive Average Pooling Layer) | 512 |
| | Fully-connected Layer | 128 |
| | **Methods** See Table 7 and Table 8 for details. | |

**AdaTime** Unless stated otherwise, we follow the implementation and hyper-parameter settings as reported in Ragab et al. (2022). We extended the AdaTime suite to comprise a collection of 11 domain adaptation algorithms. We learned all domain adaptations models according to the following approaches (see also Table 6, Table 7 and Table 8): Deep Domain Confusion (DDC) (Tzeng et al., 2014), Correlation Alignment via Deep Neural Networks (Deep-Coral) (Sun et al., 2017), Higher-order Moment Matching (HoMM) (Chen et al., 2020), Minimum Discrepancy Estimation for Deep Domain Adaptation (MMDA) (Rahman et al., 2020), Central Moment Discrepancy (CMD) (Zellinger et al., 2017), Deep Subdomain Adaptation (DSAN) (Zhu et al., 2021), Domain-Adversarial Neural Networks (DANN) (Ganin et al., 2016), Conditional Adversarial Domain Adaptation (CDAN) (Long et al., 2018), A DIRT-T Approach to Unsupervised Domain Adaptation (DIRT) (Shu et al., 2018), Convolutional deep Domain Adaptation model for Time-Series data (CoDATS) (Wilson et al., 2020), and Adversarial Spectral Kernel Matching (AdvSKM) (Liu & Xue, 2021). The backbone architecture of all models is a 1D-CNN network. It consists of three CNN blocks and each block has a 1D convolutional layer, followed by 1D batch normalization layer, ReLU activation function, 1D max pooling and Dropout. In the first block, the kernel size of the convolutional layer is set according to the dataset as reported in Ragab et al. (2022). After the convolutional blocks, we apply an 1D adaptive pooling layer. All methods are trained for 100 epochs on all datasets. The batch size is 32, except for Sleep-EDF, where we use batch size of 128. All models are trained with Adam optimizer (Kingma & Ba, 2014) and weight decay of $10^{-4}$. Additional hyper-parameters are reported in Table 10.

## D.3 MODEL SEQUENCE

Our algorithm, IWA, constructs an ensemble from a sequence of different classifiers, e.g. obtained from a sequence of possible hyper-parameter configurations in domain adaptation algorithms. To obtain this sequence of models, we train multiple models for every domain adaptation task across all datasets with different hyper-parameter choices. For the experiments on the language, im-

Table 6: Model backbone for the AdaTime suite. Kernel size, stride, output channels of the convolutional layers are dataset dependent and are chosen according to Ragab et al. (2022).

| Architecture | |
|---|---|
| **Layers** | |
| Conv Block 1 | Convolutional 1D Layer |
| | Batch Normalization 1D Layer |
| | ReLU Activation |
| | Max Pooling 1D Layer |
| | Dropout |
| Conv Block 2 | Convolutional 1D Layer |
| | Batch Normalization 1D Layer |
| | ReLU Activation |
| | Max Pooling 1D Layer |
| | Dropout |
| Conv Block 3 | Convolutional 1D Layer |
| | Batch Normalization 1D Layer |
| | ReLU Activation |
| | Max Pooling 1D Layer |
| | Dropout |
| | Adaptive Pooling 1D Layer |
| **Methods** | |
| See Table 7 and Table 8 for details. | |

Table 7: Model architecture for the AdaTime dataset. Layer hyper-parameters are dataset dependent and are chosen according to Ragab et al. (2022).

| Method Architectures (Part 1) | |
|---|---|
| **DANN** | |
| Class Output Head | Fully-connected Layer |
| Domain Classifier Head | Fully-connected Layer |
| | ReLU Activation |
| | Fully-connected Layer |
| | ReLU Activation |
| | Fully-connected Layer |
| **DeepCoral** | |
| Class Output Head | Fully-connected Layer |
| **DDC** | |
| Class Output Head | Fully-connected Layer |
| **HoMM** | |
| Class Output Head | Fully-connected Layer |
| **CoDATS** | |
| Class Output Head | Fully-connected Layer |
| | ReLU Activation |
| | Fully-connected Layer |
| | ReLU Activation |
| | Fully-connected Layer |
| **DSAN** | |
| Class Output Head | Fully-connected Layer |

age and academic dataset, the values of the hyper-parameters are shown in Table 9. For the time-series datasets we use the best hyper-parameters in Ragab et al. (2022), and, to obtain a good sequence of values. For all settings except MiniDomainnet, we multiply each parameter by

Table 8: Model architecture for the AdaTime dataset. Hyper-parameters are dataset dependent and are chosen according to Ragab et al. (2022).

| Method Architectures (Part 2) | |
|---|---|
| **AdvSKM** | |
| Class Output Head | Fully-connected Layer |
| AdvSKM Embedder 1 | Fully-connected Layer |
| | Fully-connected Layer |
| | Batch Normalization 1D Layer |
| | Cosine Activation |
| | Fully-connected Layer |
| | Fully-connected Layer |
| | Batch Normalization 1D Layer |
| | Cosine Activation |
| AdvSKM Embedder 2 | Fully-connected Layer |
| | Fully-connected Layer |
| | Batch Normalization 1D Layer |
| | ReLU Activation |
| | Fully-connected Layer |
| | Fully-connected Layer |
| | Batch Normalization 1D Layer |
| | ReLU Activation |
| **MMDA** | |
| Class Output Head | Fully-connected Layer |
| **CMD** | |
| Class Output Head | Fully-connected Layer |
| **CDAN** | |
| Class Output Head | Fully-connected Layer |
| Domain Classifier Head | Fully-connected Layer |
| | ReLU Activation |
| | Fully-connected Layer |
| | ReLU Activation |
| | Fully-connected Layer |
| **DIRT** | |
| Class Output Head | Fully-connected Layer |
| Domain Classifier Head | Fully-connected Layer |
| | ReLU Activation |
| | Fully-connected Layer |
| | ReLU Activation |
| | Fully-connected Layer |

$\lambda \in \{0, 0.0001, 0.001, 0.01, 0.05, 0.1, 0.25, 0.5, 0.75, 1, 1.5, 2, 5, 10\}$. In this way, we generate a sequence of 14 hyper-parameter choices. Due to computational limitations, in MiniDomainNet we use $\lambda \in \{0, 0.0001, 0.001, 0.01, 0.1, 1, 5, 10\}$. All values are listed in Table 9 and Table 10.

Table 9: Domain adaptation hyper-parameter sequences for experiments on the datasets Transformed Moons, AmazonReviews, and MiniDomainNet. We multiply each hyper-paramter with a set of scaling factors $\lambda \in \{0, 0.0001, 0.001, 0.01, 0.05, 0.1, 0.25, 0.5, 0.75, 1, 1.5, 2, 5, 10\}$ to obtain a sequence. Due to computational limitations, for MiniDomainNet we use only $\lambda \in \{0, 0.0001, 0.001, 0.01, 0.1, 1, 5, 10\}$.

| Method | Hyper-parameter | Transformed Moons | Amazon Reviews | MiniDomainNet |
|---|---|---|---|---|
| DANN | Classification loss weight | 0.9603 | 0.9603 | 0.9603 |
| | Domain loss weight | $\lambda \times 0.9238$ | $\lambda \times 0.9238$ | $\lambda \times 0.9238$ |
| DeepCoral | Classification loss weight | 0.05931 | 0.05931 | 0.05931 |
| | Coral loss weight | $\lambda \times 8.452$ | $\lambda \times 8.452$ | $\lambda \times 8.452$ |
| DDC | Classification loss weight | 0.1593 | 0.1593 | 0.1593 |
| | MMD loss weight | $\lambda \times 0.2048$ | $\lambda \times 0.2048$ | $\lambda \times 0.2048$ |
| CMD | Classification loss weight | 0.96 | 0.96 | 0.96 |
| | CMD loss weight | $\lambda \times 5.52$ | $\lambda \times 5.52$ | $\lambda \times 5.52$ |
| HoMM | Classification loss weight | 0.2429 | 0.2429 | 0.2429 |
| | Higher-order-MMD loss weight | $\lambda \times 0.9824$ | $\lambda \times 0.9824$ | $\lambda \times 0.9824$ |
| CoDATS | Classification loss weight | 0.5416 | 0.5416 | 0.5416 |
| | Adversarial loss weight | $\lambda \times 0.5582$ | $\lambda \times 0.5582$ | $\lambda \times 0.5582$ |
| DSAN | Classification loss weight | 0.4133 | 0.4133 | 0.4133 |
| | Local MMD loss weight | $\lambda \times 0.16$ | $\lambda \times 0.16$ | $\lambda \times 0.16$ |
| AdvSKM | Classification loss weight | 0.4637 | 0.4637 | 0.4637 |
| | Adversarial MMD loss weight | $\lambda \times 0.1511$ | $\lambda \times 0.1511$ | $\lambda \times 0.1511$ |
| MMDA | Classification loss weight | 0.9505 | 0.9505 | 0.9505 |
| | MMD loss weight | $\lambda \times 0.5476$ | $\lambda \times 0.5476$ | $\lambda \times 0.5476$ |
| | Conditional loss weight | $\lambda \times 0.5167$ | $\lambda \times 0.5167$ | $\lambda \times 0.5167$ |
| | Coral loss weight | $\lambda \times 0.5838$ | $\lambda \times 0.5838$ | $\lambda \times 0.5838$ |
| CDAN | Classification loss weight | 0.6636 | 0.6636 | 0.6636 |
| | Adversarial loss weight | $\lambda \times 0.1954$ | $\lambda \times 0.1954$ | $\lambda \times 0.1954$ |
| | Conditional loss weight | $\lambda \times 0.0124$ | $\lambda \times 0.0124$ | $\lambda \times 0.0124$ |
| DIRT | Classification loss weight | 0.9752 | 0.9752 | 0.9752 |
| | Adversarial loss weight | $\lambda \times 0.3892$ | $\lambda \times 0.3892$ | $\lambda \times 0.3892$ |
| | Conditional loss weight | $\lambda \times 0.09228$ | $\lambda \times 0.09228$ | $\lambda \times 0.09228$ |
| | Virtual adversarial loss weight | $\lambda \times 0.1947$ | $\lambda \times 0.1947$ | $\lambda \times 0.1947$ |

Table 10: Domain adaptation hyper-parameters for experiments on the time-series data. We multiply each hyper-paramter with a set of scaling factors $\lambda \in \{0, 0.0001, 0.001, 0.01, 0.05, 0.1, 0.25, 0.5, 0.75, 1, 1.5, 2, 5, 10\}$ to obtain a sequence.

| | | Datasets | | | |
|---|---|---|---|---|---|
| **Method** | **Hyper-parameter** | **UCI-HAR** | **Sleep-EDF** | **WISDM** | **HHAR** |
| DANN | Classification loss weight | 9.74 | 8.3 | 5.613 | 0.9603 |
| | Domain loss weight | $\lambda \times 5.43$ | $\lambda \times 0.324$ | $\lambda \times 1.857$ | $\lambda \times 0.9238$ |
| DeepCoral | Classification loss weight | 8.67 | 9.39 | 8.876 | 0.05931 |
| | Coral loss weight | $\lambda \times 0.44$ | $\lambda \times 0.19$ | $\lambda \times 5.56$ | $\lambda \times 8.452$ |
| DDC | Classification loss weight | 6.24 | 2.951 | 7.01 | 0.1593 |
| | MMD loss weight | $\lambda \times 6.36$ | $\lambda \times 8.923$ | $\lambda \times 7.595$ | $\lambda \times 0.2048$ |
| CMD | Classification loss weight | 0.96 | 0.96 | 0.96 | 0.96 |
| | CMD loss weight | $\lambda \times 5.52$ | $\lambda \times 5.52$ | $\lambda \times 5.52$ | $\lambda \times 5.52$ |
| HoMM | Classification loss weight | 2.15 | 0.197 | 0.1913 | 0.2429 |
| | Higher-order-MMD loss weight | $\lambda \times 9.13$ | $\lambda \times 1.102$ | $\lambda \times 4.239$ | $\lambda \times 0.9824$ |
| CoDATS | Classification loss weight | 6.21 | 9.239 | 7.187 | 0.5416 |
| | Adversarial loss weight | $\lambda \times 1.72$ | $\lambda \times 1.342$ | $\lambda \times 6.439$ | $\lambda \times 0.5582$ |
| DSAN | Classification loss weight | 1.76 | 6.713 | 0.1 | 0.4133 |
| | Local MMD loss weight | $\lambda \times 1.59$ | $\lambda \times 6.708$ | $\lambda \times 0.1$ | $\lambda \times 0.16$ |
| AdvSKM | Classification loss weight | 3.05 | 2.5 | 3.05 | 0.4637 |
| | Adversarial MMD loss weight | $\lambda \times 2.876$ | $\lambda \times 2.5$ | $\lambda \times 2.876$ | $\lambda \times 0.1511$ |
| MMDA | Classification loss weight | 6.13 | 4.48 | 0.1 | 0.9505 |
| | MMD loss weight | $\lambda \times 2.37$ | $\lambda \times 5.951$ | $\lambda \times 0.1$ | $\lambda \times 0.5476$ |
| | Conditional loss weight | $\lambda \times 7.16$ | $\lambda \times 6.13$ | $\lambda \times 0.4753$ | $\lambda \times 0.5167$ |
| | Coral loss weight | $\lambda \times 8.63$ | $\lambda \times 3.36$ | $\lambda \times 0.1$ | $\lambda \times 0.5838$ |
| CDAN | Classification loss weight | 5.19 | 6.803 | 9.54 | 0.6636 |
| | Adversarial loss weight | $\lambda \times 2.91$ | $\lambda \times 4.726$ | $\lambda \times 3.283$ | $\lambda \times 0.1954$ |
| | Conditional loss weight | $\lambda \times 1.73$ | $\lambda \times 1.307$ | $\lambda \times 0.1$ | $\lambda \times 0.0124$ |
| DIRT | Classification loss weight | 7.0 | 9.183 | 0.1 | 0.9752 |
| | Adversarial loss weight | $\lambda \times 4.51$ | $\lambda \times 7.411$ | $\lambda \times 0.1$ | $\lambda \times 0.3892$ |
| | Conditional loss weight | $\lambda \times 0.79$ | $\lambda \times 2.564$ | $\lambda \times 0.1$ | $\lambda \times 0.09228$ |
| | Virtual adversarial loss weight | $\lambda \times 9.31$ | $\lambda \times 3.583$ | $\lambda \times 0.1$ | $\lambda \times 0.1947$ |

### D.4 Matrix Inversion

Matrix inversion is a well-known numerical task, especially in cases of limited computing precision and ill-conditioned matrices. In our case, similar models in the given sequence can cause numerical instability due to limited compute precision. That is, occasionally a tabula rasa inversion of the matrix $\widetilde{G}$ in Algorithm 1 is numerically unstable. Various standard approaches can be applied to handle this common issue, including the exclusion of similar models and various regularization techniques. In our computational setup, we rely on the Python routine *numpy.linalg.pinv*, which is based on the eigendecompostion of $\widetilde{G}$ (coinciding with the singular value decomposition in our case due to positive-definiteness) and an eigenvalue-based regularization based on a treshold value *rcond* for small eigenvalues, see Strang (1980, pages 138–140) for details. The choice of *rcond* depends on the scale of the Gram matrix and can therefore be chosen by source data only. Based on evaluating our method on source data only (target domain is fixed to be source domain) on several choices for *rcond*, we obtain a stable choice for *rcond* of $10^{-1}$ for all datasets.

### D.5 Correlation Analysis: Target Accuracy of Individual Models vs. Aggregation Weight

Figure 2 in the main paper suggests that there is a positive correlation between the target accuracy of each individual model (orange dashed line, Figure 2 top) and the respective aggregation weight (red bars, Figure 2 bottom), if the linear aggregation of the models is computed by our method IWA. In the following, we analyze whether this trend holds throughout all other experiments.

More precisely, for a given sequence $f_1, \ldots, f_l$ of models, we compute the Pearson correlation coefficient between the aggregation weights $\widetilde{c}_1, \ldots, \widetilde{c}_l$ (see Algorithm 1) and the corresponding target accuracies $\frac{1}{t} \sum_{i=1}^{t} \mathbf{1}[y_i = f_1(x_i)], \ldots, \frac{1}{t} \sum_{i=1}^{t} \mathbf{1}[y_i = f_l(x_i)]$ for the target test data $(x_1, y_1), \ldots, (x_t, y_t)$, where $\mathbf{1}[P] = 1$ iff $P$ is true and $\mathbf{1}[P] = 0$ otherwise. In this context, a positive correlation coefficient means that the aggregation algorithm assigns a higher weight to models performing better on target samples. We calculate these coefficients for our method IWA and the other linear regression baselines SOR, TCR, and TMR for all domain adaptation methods across all datasets and cross-domain scenarios. Note that for this analysis we cannot compare to the other baseline TMV, as the count-based aggregation by majority voting does not involve the computation of aggregation weights.

In Figure 4 and 6 we compare the resulting correlation coefficient distribution of our method IWA to the ones for the heuristic baselines SOR, TCR, and TMR. Figure 5 compares the correlation coefficients for IWA on different datasets.

We find that IWA shows a stronger positive correlation than other methods, between a model's target accuracy and its aggregation weight.

### D.6 Sensitivity Analysis: Effect of Adding Inaccurate Models

We study the sensitivity of our method with respect to adding inaccurate models in the given sequence of models. We define an innacurate model as having a target accuracy lower than $80\%$ of the target accuracy of the model computed without domain adaptation (SO). In particular, we add $+10, +50$ and $+100$ inaccurate models to the given sequences of models. One inaccurate model is constructed as follows: First, a model is chosen uniformly at random from the given sequence of models. Second, the outputs of the chosen model are corrupted by adding, to half of the elements of its vector-valued output, random Gaussian noise with zero-mean and unit variance.

In Figure 7, we show the median performance over all domain adaptation methods for each dataset. We see, that the performance of our method (IWA) is not very sensitive w.r.t. an increase in the number of inaccurate models. This is in contrast to the heuristics (e.g. TMV, TMR or TCR) and model selection method DEV, which show a high sensitivity concerning adding inaccurate models. SOR also shows stable results; it is, however, clearly outperformed by our method in five out of six datasets. We conclude, that IWA is not only overall the best performing method, but also the most stable choice concerning inaccurate models in the given sequence.

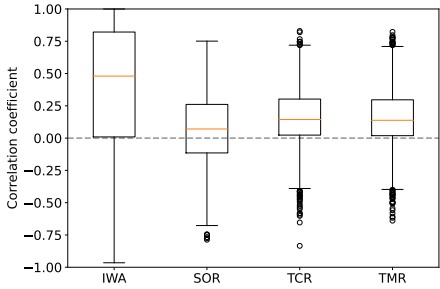

Figure 4: Boxplots of the correlation coefficients of IWA and the linear regression heuristic baselines SOR, TCR, and TMR over all datasets.

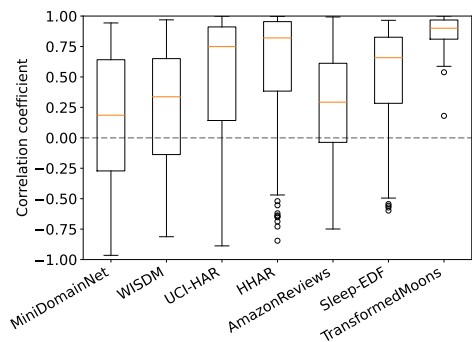

Figure 5: Boxplots of the correlation coefficients of IWA for each dataset.

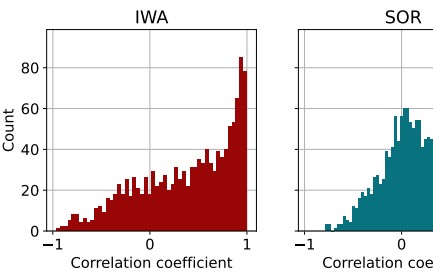
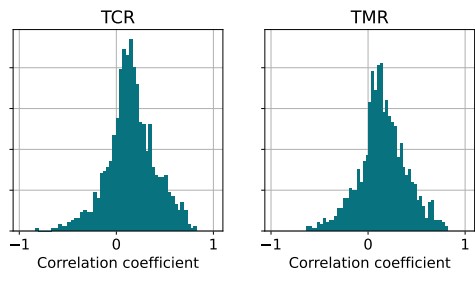

Figure 6: Histogram of the correlation coefficients of IWA and the linear regression heuristic baselines SOR, TCR and TMR over all datasets (showing the same data as Figure 4). In comparison to the heuristic baselines, IWA shows a stronger positive correlation between a model's target accuracy and its aggregation weight.

## D.7 DETAILED EMPIRICAL RESULTS

In this section, we add all result tables for the datasets described in the main paper. Table 15, Table 16 and Table 17 show all domain adaptation tasks for the Amazon Review dataset. Table 13 shows all domain adaptation tasks for the MiniDomainNet experiments. Table 21, Table 22, Table 23, Table 24, Table 25, Table 26, Table 27, and Table 28 show all domain adaptation task results for the time-series datasets.

**Baselines** As addressed in the main paper, our method, IWA, is compared to ensemble learning methods that use linear regression and majority voting as *heuristic* for model aggregation, and, model selection methods with *theoretical error guarantees*. The heuristic baselines are majority voting on target data (TMV), source-only regression (SOR), target majority voting regression (TMR), target confidence average regression (TCR). The model selection methods with theoretical error guarantees are importance weighted validation (IWV) (Sugiyama et al., 2007) and deep embedded validation (DEV) (Kouw et al., 2019). The tables also provide a column for source-only (SO) performance and target-best (TB) performance. We highlight in bold the performance of the best performing method with theoretical error guarantees, and in italic the best performing heuristic.

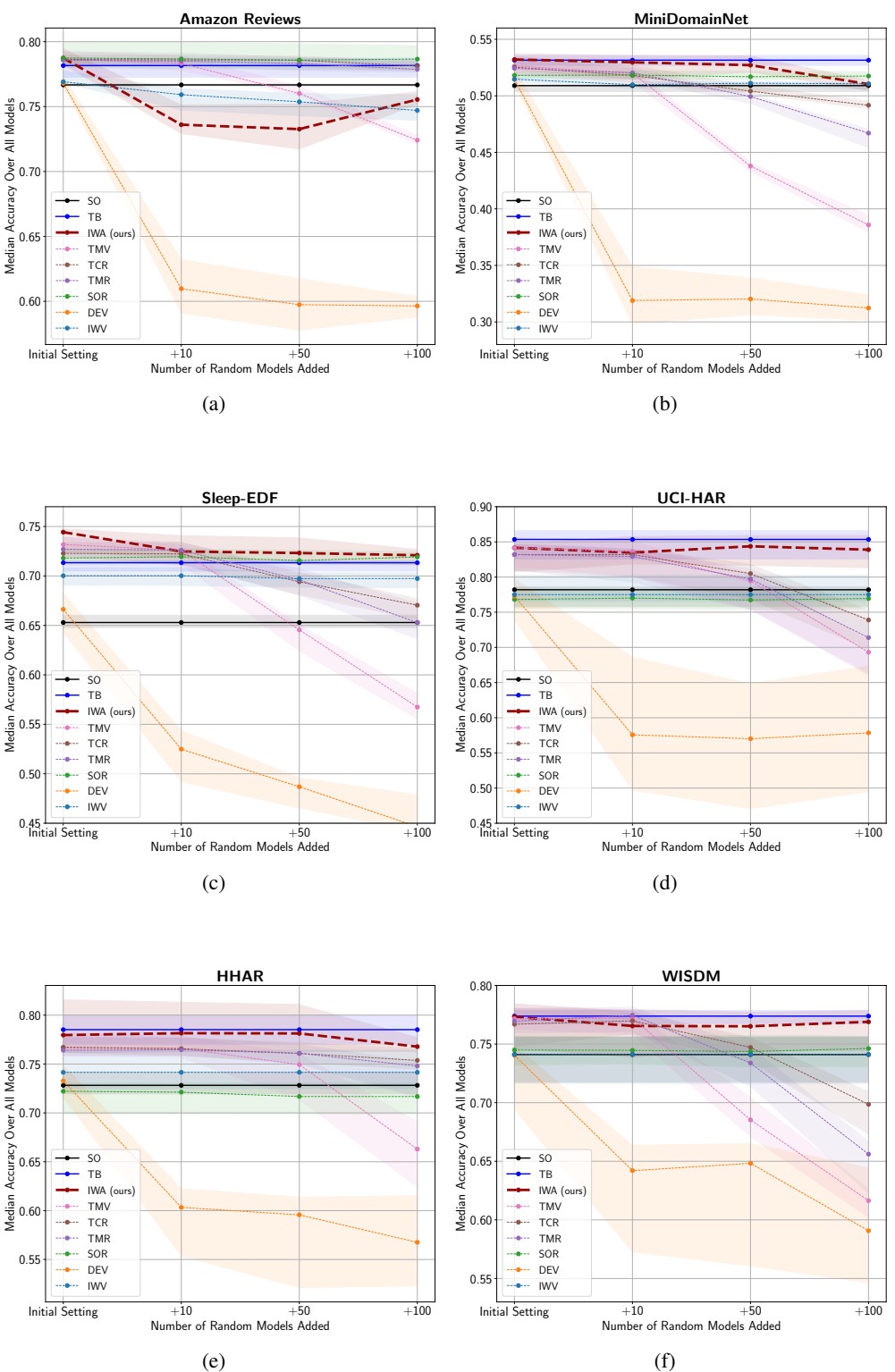

Figure 7: Sensitivity of methods for parameter choice issues w.r.t. adding inaccurate models in the given sequence of models; separate for each dataset (a–f), averaged over all domain adaptation methods, source-target pairs, and random seeds. Horizontal axes: Number of inaccurate models added to the initial sequence of models. Vertical axes: Target accuracy. Solid lines indicate median and shaded area indicate 50% confidence intervals.

### D.7.1 DETAILED SUMMARY RESULTS

Table 11: Average target accuracies (and average standard deviations) for all 7 datasets (e.g., Sleep-EDF, MiniDomainNet, Amazon Reviews) taken over several domain adaptation tasks (e.g., 5 on Sleep-EDF, 5 on MiniDomainNet, 12 on Amazon Reviews), 11 domain adaptation methods (e.g., DANN, HoMM, CMD) and 3 repetitions with different random initialization of model weights. The input sequences of the approaches (e.g., DEV, IWA) consist of neural networks computed by runs of the domain adaptation methods with different hyper-parameters (e.g., 8 different values of $\lambda$ for DANN).

| Dataset | SO | Heuristic | | | | Theoretical error guarantees | | | TB |
|---|---|---|---|---|---|---|---|---|---|
| | | TMV | TMR | TCR | SOR | IWV | DEV | IWA (ours) | |
| Transformed Moons | 0.989($\pm$0.008) | 0.980($\pm$0.006) | 0.981($\pm$0.007) | *0.997($\pm$0.002)* | 0.989($\pm$0.010) | 0.989($\pm$0.008) | 0.981($\pm$0.022) | **0.997($\pm$0.002)** | 0.997($\pm$0.005) |
| Amazon Reviews | 0.767($\pm$0.011) | 0.787($\pm$0.009) | 0.786($\pm$0.010) | 0.786($\pm$0.010) | *0.789($\pm$0.010)* | 0.772($\pm$0.014) | 0.764($\pm$0.019) | **0.788($\pm$0.009)** | 0.781($\pm$0.012) |
| MiniDomainNet | 0.507($\pm$0.022) | *0.526($\pm$0.011)* | 0.525($\pm$0.014) | 0.526($\pm$0.013) | 0.518($\pm$0.012) | 0.513($\pm$0.022) | 0.515($\pm$0.028) | **0.531($\pm$0.011)** | 0.534($\pm$0.022) |
| Sleep-EDF | 0.655($\pm$0.054) | *0.729($\pm$0.018)* | 0.729($\pm$0.024) | 0.725($\pm$0.023) | 0.717($\pm$0.028) | 0.700($\pm$0.052) | 0.660($\pm$0.057) | **0.737($\pm$0.020)** | 0.712($\pm$0.045) |
| UCI-HAR | 0.770($\pm$0.046) | *0.840($\pm$0.017)* | 0.833($\pm$0.023) | 0.832($\pm$0.024) | 0.769($\pm$0.060) | 0.774($\pm$0.070) | 0.765($\pm$0.090) | **0.835($\pm$0.020)** | 0.850($\pm$0.029) |
| HHAR | 0.732($\pm$0.042) | 0.771($\pm$0.015) | 0.768($\pm$0.017) | *0.771($\pm$0.018)* | 0.722($\pm$0.068) | 0.746($\pm$0.037) | 0.722($\pm$0.063) | **0.787($\pm$0.012)** | 0.784($\pm$0.028) |
| WISDM | 0.736($\pm$0.050) | *0.768($\pm$0.027)* | 0.768($\pm$0.036) | 0.765($\pm$0.037) | 0.737($\pm$0.062) | 0.736($\pm$0.052) | 0.726($\pm$0.077) | **0.764($\pm$0.025)** | 0.771($\pm$0.046) |

Table 12: Mean and standard deviation (after $\pm$) of target classification accuracy on Amazon Reviews dataset over three different random initialization of model weights and 12 domain adaptation tasks.

| | Amazon Reviews | | | | | | | | |
|---|---|---|---|---|---|---|---|---|---|
| Method | SO | Heuristic | | | | Theoretical error guarantees | | | TB |
| | | TMV | TMR | TCR | SOR | IWV | DEV | IWA (ours) | |
| HoMM | 0.769($\pm$0.009) | 0.777($\pm$0.010) | 0.778($\pm$0.010) | *0.778($\pm$0.011)* | 0.777($\pm$0.010) | 0.765($\pm$0.011) | 0.766($\pm$0.012) | **0.778($\pm$0.010)** | 0.769($\pm$0.012) |
| AdvSKM | 0.766($\pm$0.012) | *0.780($\pm$0.009)* | 0.779($\pm$0.010) | 0.779($\pm$0.008) | 0.778($\pm$0.011) | 0.769($\pm$0.012) | 0.766($\pm$0.012) | **0.780($\pm$0.009)** | 0.770($\pm$0.012) |
| DIRT | 0.764($\pm$0.009) | 0.786($\pm$0.008) | 0.786($\pm$0.010) | 0.786($\pm$0.008) | *0.800($\pm$0.008)* | 0.778($\pm$0.022) | 0.773($\pm$0.056) | **0.787($\pm$0.008)** | 0.786($\pm$0.009) |
| DDC | 0.766($\pm$0.012) | 0.779($\pm$0.010) | *0.780($\pm$0.009)* | 0.779($\pm$0.010) | 0.778($\pm$0.010) | 0.767($\pm$0.017) | 0.768($\pm$0.011) | **0.780($\pm$0.010)** | 0.770($\pm$0.013) |
| CMD | 0.767($\pm$0.012) | 0.791($\pm$0.009) | 0.792($\pm$0.009) | 0.789($\pm$0.010) | *0.792($\pm$0.010)* | 0.765($\pm$0.015) | 0.710($\pm$0.015) | **0.794($\pm$0.009)** | 0.785($\pm$0.009) |
| MMDA | 0.767($\pm$0.011) | 0.787($\pm$0.011) | 0.785($\pm$0.010) | 0.785($\pm$0.010) | *0.787($\pm$0.012)* | 0.769($\pm$0.011) | 0.766($\pm$0.010) | **0.787($\pm$0.011)** | 0.782($\pm$0.011) |
| CoDATS | 0.766($\pm$0.013) | 0.795($\pm$0.009) | 0.793($\pm$0.010) | 0.794($\pm$0.012) | *0.799($\pm$0.010)* | 0.779($\pm$0.016) | 0.773($\pm$0.020) | **0.796($\pm$0.009)** | 0.791($\pm$0.015) |
| Deep-Coral | 0.766($\pm$0.012) | *0.784($\pm$0.009)* | 0.783($\pm$0.009) | 0.783($\pm$0.009) | 0.782($\pm$0.009) | 0.769($\pm$0.016) | 0.769($\pm$0.037) | **0.785($\pm$0.009)** | 0.776($\pm$0.013) |
| CDAN | 0.767($\pm$0.012) | *0.788($\pm$0.010)* | 0.787($\pm$0.009) | 0.787($\pm$0.010) | 0.787($\pm$0.011) | 0.775($\pm$0.011) | 0.776($\pm$0.014) | **0.788($\pm$0.010)** | 0.777($\pm$0.011) |
| DANN | 0.767($\pm$0.012) | 0.796($\pm$0.010) | 0.792($\pm$0.010) | 0.793($\pm$0.010) | *0.800($\pm$0.011)* | 0.776($\pm$0.011) | 0.778($\pm$0.012) | **0.797($\pm$0.009)** | 0.798($\pm$0.012) |
| DSAN | 0.769($\pm$0.009) | 0.796($\pm$0.009) | 0.792($\pm$0.009) | 0.791($\pm$0.010) | *0.800($\pm$0.010)* | 0.779($\pm$0.012) | 0.763($\pm$0.017) | **0.795($\pm$0.009)** | 0.789($\pm$0.012) |
| Avg. | 0.767($\pm$0.011) | 0.787($\pm$0.009) | 0.786($\pm$0.010) | 0.786($\pm$0.010) | *0.789($\pm$0.010)* | 0.772($\pm$0.014) | 0.764($\pm$0.019) | **0.788($\pm$0.009)** | 0.781($\pm$0.012) |

Table 13: Mean and standard deviation (after $\pm$) of target classification accuracy on MiniDomainNet dataset over three different random initialization of model weights and five domain adaptation tasks.

| | MiniDomainNet | | | | | | | | |
|---|---|---|---|---|---|---|---|---|---|
| Method | SO | Heuristic | | | | Theoretical error guarantees | | | TB |
| | | TMV | TMR | TCR | SOR | IWV | DEV | IWA (ours) | |
| HoMM | 0.509($\pm$0.018) | *0.526($\pm$0.009)* | 0.524($\pm$0.010) | 0.523($\pm$0.010) | 0.518($\pm$0.015) | 0.511($\pm$0.018) | 0.511($\pm$0.018) | **0.531($\pm$0.007)** | 0.537($\pm$0.020) |
| AdvSKM | 0.509($\pm$0.015) | *0.516($\pm$0.011)* | 0.514($\pm$0.014) | 0.514($\pm$0.017) | 0.514($\pm$0.009) | 0.515($\pm$0.032) | 0.512($\pm$0.032) | **0.516($\pm$0.011)** | 0.522($\pm$0.017) |
| DIRT | 0.499($\pm$0.022) | *0.517($\pm$0.006)* | 0.515($\pm$0.009) | 0.508($\pm$0.006) | 0.507($\pm$0.018) | 0.493($\pm$0.028) | 0.498($\pm$0.033) | **0.519($\pm$0.006)** | 0.525($\pm$0.020) |
| DDC | 0.510($\pm$0.032) | *0.514($\pm$0.012)* | 0.512($\pm$0.018) | 0.511($\pm$0.019) | 0.514($\pm$0.015) | 0.511($\pm$0.028) | 0.512($\pm$0.043) | **0.516($\pm$0.015)** | 0.521($\pm$0.019) |
| CMD | 0.509($\pm$0.019) | 0.528($\pm$0.011) | *0.531($\pm$0.016)* | 0.531($\pm$0.016) | 0.522($\pm$0.013) | 0.518($\pm$0.025) | 0.490($\pm$0.037) | **0.533($\pm$0.009)** | 0.533($\pm$0.024) |
| MMDA | 0.509($\pm$0.021) | *0.524($\pm$0.004)* | 0.522($\pm$0.005) | 0.523($\pm$0.007) | 0.519($\pm$0.012) | 0.517($\pm$0.008) | 0.524($\pm$0.006) | **0.527($\pm$0.009)** | 0.531($\pm$0.029) |
| CoDATS | 0.502($\pm$0.026) | 0.535($\pm$0.026) | 0.533($\pm$0.027) | *0.536($\pm$0.028)* | 0.525($\pm$0.015) | 0.518($\pm$0.031) | 0.524($\pm$0.051) | **0.536($\pm$0.025)** | 0.529($\pm$0.042) |
| Deep-Coral | 0.505($\pm$0.022) | *0.538($\pm$0.009)* | 0.534($\pm$0.013) | 0.533($\pm$0.012) | 0.520($\pm$0.010) | 0.518($\pm$0.022) | 0.524($\pm$0.034) | **0.539($\pm$0.012)** | 0.535($\pm$0.024) |
| CDAN | 0.514($\pm$0.028) | *0.526($\pm$0.013)* | 0.524($\pm$0.011) | 0.525($\pm$0.008) | 0.517($\pm$0.005) | 0.513($\pm$0.011) | 0.516($\pm$0.014) | **0.532($\pm$0.010)** | 0.542($\pm$0.017) |
| DANN | 0.496($\pm$0.023) | 0.530($\pm$0.011) | 0.535($\pm$0.011) | *0.538($\pm$0.005)* | 0.516($\pm$0.011) | 0.519($\pm$0.015) | 0.515($\pm$0.013) | **0.541($\pm$0.006)** | 0.532($\pm$0.014) |
| DSAN | 0.509($\pm$0.022) | 0.537($\pm$0.011) | 0.534($\pm$0.013) | *0.543($\pm$0.011)* | 0.525($\pm$0.013) | 0.513($\pm$0.018) | 0.535($\pm$0.023) | **0.546($\pm$0.007)** | 0.563($\pm$0.013) |
| Avg. | 0.507($\pm$0.022) | *0.526($\pm$0.011)* | 0.525($\pm$0.014) | 0.526($\pm$0.013) | 0.518($\pm$0.012) | 0.513($\pm$0.022) | 0.515($\pm$0.028) | **0.531($\pm$0.011)** | 0.534($\pm$0.022) |

Table 14: Mean and standard deviation (after ±) of target classification accuracy on four time series datasets over three different random initialization of model weights and five domain adaptation tasks.

**Sleep-EDF**

| Method | SO | Heuristic | | | | Theoretical error guarantees | | | TB |
|---|---|---|---|---|---|---|---|---|---|
| | | TMV | TMR | TCR | SOR | IWV | DEV | IWA (ours) | |
| HoMM | 0.676(±0.036) | 0.722(±0.017) | 0.719(±0.023) | 0.718(±0.021) | 0.724(±0.032) | 0.726(±0.046) | 0.678(±0.035) | **0.747(±0.025)** | 0.715(±0.047) |
| AdvSKM | 0.665(±0.058) | 0.708(±0.023) | 0.712(±0.027) | 0.712(±0.032) | 0.718(±0.030) | 0.703(±0.069) | 0.692(±0.038) | **0.722(±0.025)** | 0.706(±0.054) |
| DIRT | 0.656(±0.058) | 0.743(±0.009) | 0.745(±0.012) | 0.748(±0.019) | 0.742(±0.031) | 0.679(±0.038) | 0.686(±0.066) | **0.749(±0.010)** | 0.728(±0.037) |
| DDC | 0.646(±0.035) | 0.717(±0.029) | 0.721(±0.037) | 0.712(±0.031) | 0.695(±0.020) | 0.694(±0.056) | 0.666(±0.031) | **0.724(±0.012)** | 0.704(±0.031) |
| CMD | 0.653(±0.057) | 0.740(±0.022) | 0.736(±0.016) | 0.723(±0.020) | 0.709(±0.015) | 0.716(±0.052) | 0.640(±0.068) | **0.729(±0.018)** | 0.725(±0.053) |
| MMDA | 0.650(±0.051) | 0.736(±0.014) | 0.727(±0.021) | 0.723(±0.018) | 0.714(±0.028) | 0.704(±0.033) | 0.660(±0.034) | **0.745(±0.031)** | 0.715(±0.042) |
| CoDATS | 0.672(±0.084) | 0.738(±0.029) | 0.739(±0.036) | 0.736(±0.030) | 0.723(±0.039) | 0.683(±0.090) | 0.690(±0.107) | **0.744(±0.021)** | 0.715(±0.045) |
| Deep-Coral | 0.643(±0.049) | 0.716(±0.018) | 0.717(±0.028) | 0.712(±0.027) | 0.694(±0.032) | 0.700(±0.053) | 0.675(±0.077) | **0.713(±0.021)** | 0.702(±0.070) |
| CDAN | 0.652(±0.056) | 0.732(±0.016) | 0.739(±0.024) | 0.739(±0.018) | 0.728(±0.029) | 0.697(±0.031) | 0.642(±0.065) | **0.748(±0.019)** | 0.713(±0.045) |
| DANN | 0.641(±0.047) | 0.722(±0.017) | 0.723(±0.026) | 0.721(±0.025) | 0.714(±0.024) | 0.687(±0.034) | 0.644(±0.046) | **0.724(±0.018)** | 0.710(±0.035) |
| DSAN | 0.653(±0.060) | 0.748(±0.008) | 0.740(±0.016) | 0.732(±0.016) | 0.728(±0.026) | 0.712(±0.070) | 0.589(±0.063) | **0.757(±0.016)** | 0.700(±0.033) |
| Avg. | 0.655(±0.054) | 0.729(±0.018) | 0.729(±0.024) | 0.725(±0.023) | 0.717(±0.028) | 0.700(±0.052) | 0.660(±0.057) | **0.737(±0.020)** | 0.712(±0.045) |

**UCI-HAR**

| Method | SO | Heuristic | | | | Theoretical error guarantees | | | TB |
|---|---|---|---|---|---|---|---|---|---|
| | | TMV | TMR | TCR | SOR | IWV | DEV | IWA (ours) | |
| HoMM | 0.782(±0.078) | 0.833(±0.020) | 0.818(±0.023) | 0.818(±0.022) | 0.783(±0.040) | 0.809(±0.095) | 0.800(±0.098) | **0.826(±0.010)** | 0.854(±0.039) |
| AdvSKM | 0.724(±0.059) | 0.791(±0.024) | 0.800(±0.022) | 0.810(±0.022) | 0.768(±0.042) | 0.707(±0.100) | 0.711(±0.167) | **0.800(±0.022)** | 0.811(±0.039) |
| DIRT | 0.783(±0.044) | 0.912(±0.013) | 0.907(±0.009) | 0.890(±0.016) | 0.756(±0.036) | 0.807(±0.107) | 0.808(±0.112) | **0.900(±0.015)** | 0.928(±0.034) |
| DDC | 0.790(±0.061) | 0.806(±0.019) | 0.807(±0.026) | 0.810(±0.017) | 0.756(±0.108) | 0.724(±0.066) | 0.734(±0.109) | **0.804(±0.028)** | 0.792(±0.013) |
| CMD | 0.788(±0.058) | 0.869(±0.012) | 0.849(±0.041) | 0.839(±0.023) | 0.731(±0.066) | 0.804(±0.064) | 0.812(±0.080) | **0.842(±0.025)** | 0.888(±0.037) |
| MMDA | 0.785(±0.018) | 0.819(±0.022) | 0.812(±0.028) | 0.800(±0.032) | 0.759(±0.085) | 0.773(±0.073) | 0.767(±0.107) | **0.807(±0.025)** | 0.840(±0.055) |
| CoDATS | 0.760(±0.037) | 0.854(±0.022) | 0.832(±0.027) | 0.832(±0.006) | 0.785(±0.057) | 0.804(±0.064) | 0.794(±0.078) | **0.846(±0.016)** | 0.867(±0.012) |
| Deep-Coral | 0.790(±0.035) | 0.810(±0.007) | 0.800(±0.022) | 0.808(±0.030) | 0.771(±0.023) | 0.768(±0.044) | 0.773(±0.087) | **0.808(±0.016)** | 0.806(±0.022) |
| CDAN | 0.756(±0.055) | 0.842(±0.009) | 0.843(±0.020) | 0.840(±0.044) | 0.802(±0.080) | 0.781(±0.072) | 0.687(±0.068) | **0.846(±0.018)** | 0.853(±0.026) |
| DANN | 0.756(±0.026) | 0.858(±0.016) | 0.856(±0.033) | 0.856(±0.033) | 0.800(±0.057) | 0.763(±0.032) | 0.780(±0.043) | **0.849(±0.023)** | 0.847(±0.007) |
| DSAN | 0.762(±0.032) | 0.849(±0.023) | 0.843(±0.033) | 0.854(±0.025) | 0.749(±0.065) | 0.775(±0.043) | 0.744(±0.035) | **0.858(±0.023)** | 0.865(±0.038) |
| Avg. | 0.770(±0.046) | 0.840(±0.017) | 0.833(±0.023) | 0.832(±0.024) | 0.769(±0.060) | 0.774(±0.070) | 0.765(±0.090) | **0.835(±0.020)** | 0.850(±0.029) |

**HHAR**

| Method | SO | Heuristic | | | | Theoretical error guarantees | | | TB |
|---|---|---|---|---|---|---|---|---|---|
| | | TMV | TMR | TCR | SOR | IWV | DEV | IWA (ours) | |
| HoMM | 0.739(±0.044) | 0.757(±0.014) | 0.759(±0.013) | 0.759(±0.011) | 0.700(±0.058) | 0.720(±0.027) | 0.733(±0.031) | **0.759(±0.007)** | 0.764(±0.023) |
| AdvSKM | 0.718(±0.042) | 0.749(±0.027) | 0.742(±0.032) | 0.748(±0.034) | 0.676(±0.046) | 0.730(±0.051) | 0.728(±0.051) | **0.752(±0.031)** | 0.749(±0.025) |
| DIRT | 0.728(±0.026) | 0.803(±0.011) | 0.792(±0.016) | 0.803(±0.017) | 0.796(±0.066) | 0.743(±0.028) | 0.739(±0.075) | **0.816(±0.008)** | 0.820(±0.015) |
| DDC | 0.716(±0.063) | 0.748(±0.014) | 0.750(±0.009) | 0.748(±0.007) | 0.717(±0.075) | 0.711(±0.048) | 0.705(±0.066) | **0.748(±0.012)** | 0.729(±0.027) |
| CMD | 0.748(±0.027) | 0.760(±0.014) | 0.764(±0.006) | 0.767(±0.007) | 0.737(±0.100) | **0.775(±0.031)** | 0.643(±0.031) | 0.766(±0.016) | 0.794(±0.023) |
| MMDA | 0.738(±0.036) | 0.783(±0.017) | 0.781(±0.016) | 0.780(±0.015) | 0.698(±0.038) | 0.719(±0.036) | 0.731(±0.047) | **0.780(±0.017)** | 0.785(±0.035) |
| CoDATS | 0.710(±0.030) | 0.766(±0.023) | 0.772(±0.040) | 0.773(±0.050) | 0.722(±0.064) | 0.739(±0.028) | 0.739(±0.040) | **0.812(±0.009)** | 0.785(±0.039) |
| Deep-Coral | 0.745(±0.046) | 0.766(±0.012) | 0.762(±0.015) | 0.766(±0.027) | 0.681(±0.073) | 0.754(±0.054) | 0.758(±0.244) | **0.764(±0.006)** | 0.776(±0.023) |
| CDAN | 0.728(±0.039) | 0.762(±0.012) | 0.758(±0.017) | 0.764(±0.046) | 0.765(±0.063) | 0.774(±0.035) | 0.775(±0.036) | **0.816(±0.011)** | 0.790(±0.038) |
| DANN | 0.757(±0.057) | 0.779(±0.012) | 0.774(±0.009) | 0.773(±0.011) | 0.722(±0.103) | 0.798(±0.041) | 0.793(±0.045) | **0.818(±0.009)** | 0.807(±0.020) |
| DSAN | 0.721(±0.053) | 0.803(±0.010) | 0.797(±0.014) | 0.802(±0.007) | 0.724(±0.065) | 0.741(±0.033) | 0.596(±0.031) | **0.825(±0.008)** | 0.826(±0.046) |
| Avg. | 0.732(±0.042) | 0.771(±0.015) | 0.768(±0.017) | 0.771(±0.018) | 0.722(±0.068) | 0.746(±0.037) | 0.722(±0.063) | **0.787(±0.012)** | 0.784(±0.028) |

**WISDM**

| Method | SO | Heuristic | | | | Theoretical error guarantees | | | TB |
|---|---|---|---|---|---|---|---|---|---|
| | | TMV | TMR | TCR | SOR | IWV | DEV | IWA (ours) | |
| HoMM | 0.753(±0.054) | 0.741(±0.026) | 0.738(±0.031) | 0.739(±0.047) | 0.775(±0.062) | **0.753(±0.054)** | 0.740(±0.054) | 0.740(±0.021) | 0.774(±0.037) |
| AdvSKM | 0.747(±0.050) | 0.771(±0.043) | 0.781(±0.055) | 0.779(±0.035) | 0.742(±0.062) | 0.747(±0.050) | 0.747(±0.135) | **0.777(±0.031)** | 0.779(±0.041) |
| DIRT | 0.738(±0.038) | 0.792(±0.015) | 0.797(±0.024) | 0.797(±0.037) | 0.756(±0.071) | 0.738(±0.059) | 0.797(±0.059) | **0.798(±0.018)** | 0.816(±0.063) |
| DDC | 0.741(±0.071) | 0.780(±0.032) | 0.779(±0.052) | 0.787(±0.049) | 0.737(±0.071) | 0.741(±0.076) | 0.741(±0.063) | **0.782(±0.038)** | 0.770(±0.060) |
| CMD | 0.710(±0.088) | 0.772(±0.021) | 0.765(±0.032) | 0.767(±0.040) | 0.728(±0.092) | 0.713(±0.084) | 0.686(±0.113) | **0.773(±0.032)** | 0.742(±0.071) |
| MMDA | 0.759(±0.047) | 0.789(±0.017) | 0.772(±0.030) | 0.745(±0.035) | 0.754(±0.050) | 0.759(±0.047) | 0.750(±0.047) | **0.790(±0.018)** | 0.775(±0.030) |
| CoDATS | 0.711(±0.039) | 0.775(±0.018) | 0.757(±0.027) | 0.751(±0.020) | 0.682(±0.057) | 0.709(±0.039) | 0.735(±0.054) | **0.764(±0.015)** | 0.770(±0.019) |
| Deep-Coral | 0.694(±0.030) | 0.717(±0.041) | 0.723(±0.037) | 0.713(±0.035) | 0.664(±0.055) | 0.694(±0.030) | 0.670(±0.149) | **0.723(±0.026)** | 0.736(±0.044) |
| CDAN | 0.760(±0.057) | 0.762(±0.048) | 0.762(±0.046) | 0.781(±0.051) | 0.768(±0.074) | 0.760(±0.057) | 0.750(±0.091) | **0.765(±0.040)** | 0.779(±0.049) |
| DANN | 0.724(±0.042) | 0.789(±0.018) | 0.802(±0.028) | 0.796(±0.036) | 0.745(±0.026) | 0.720(±0.042) | 0.702(±0.053) | **0.778(±0.019)** | 0.765(±0.044) |
| DSAN | 0.759(±0.030) | 0.765(±0.024) | 0.769(±0.034) | 0.756(±0.025) | 0.757(±0.068) | **0.759(±0.030)** | 0.663(±0.025) | 0.722(±0.013) | 0.779(±0.044) |
| Avg. | 0.736(±0.050) | 0.768(±0.027) | 0.768(±0.036) | 0.765(±0.037) | 0.737(±0.062) | 0.736(±0.052) | 0.726(±0.077) | **0.764(±0.025)** | 0.771(±0.046) |

### D.7.2 DETAILED AMAZON REVIEWS RESULTS

Table 15: Mean and standard deviation (after ±) of target classification accuracy on Amazon Reviews (Part 1) over 3 repetitions with different random initialization of model weights.

**HoMM**

| Task | SO | Heuristic | | | | Theoretical error guarantees | | | |
|---|---|---|---|---|---|---|---|---|---|
| | | TMV | TMR | TCR | SOR | IWV | DEV | IWA (ours) | TB |
| B→D | 0.778(±0.01) | 0.793(±0.008) | 0.793(±0.008) | 0.795(±0.002) | 0.795(±0.008) | 0.782(±0.021) | 0.782(±0.021) | **0.794(±0.007)** | 0.789(±0.008) |
| B→E | 0.745(±0.014) | 0.759(±0.013) | 0.759(±0.013) | 0.761(±0.012) | 0.763(±0.012) | 0.747(±0.005) | 0.749(±0.025) | **0.761(±0.012)** | 0.758(±0.012) |
| B→K | 0.77(±0.018) | 0.782(±0.018) | 0.782(±0.018) | 0.782(±0.017) | 0.779(±0.013) | 0.761(±0.018) | 0.767(±0.003) | **0.781(±0.017)** | 0.776(±0.018) |
| D→B | 0.785(±0.006) | 0.791(±0.004) | 0.791(±0.004) | 0.79(±0.002) | 0.788(±0.008) | 0.778(±0.002) | 0.777(±0.015) | **0.79(±0.003)** | 0.787(±0.003) |
| D→E | 0.772(±0.012) | 0.781(±0.003) | 0.781(±0.003) | 0.779(±0.004) | 0.778(±0.004) | 0.764(±0.017) | 0.769(±0.009) | **0.781(±0.004)** | 0.774(±0.003) |
| D→K | 0.783(±0.015) | 0.791(±0.016) | 0.791(±0.016) | 0.791(±0.016) | 0.791(±0.017) | 0.781(±0.01) | 0.782(±0.01) | **0.789(±0.016)** | 0.786(±0.017) |
| E→B | 0.702(±0.026) | 0.7(±0.02) | 0.7(±0.02) | 0.701(±0.019) | 0.699(±0.021) | 0.69(±0.011) | 0.692(±0.014) | **0.698(±0.011)** | 0.703(±0.03) |
| E→D | 0.733(±0.003) | 0.733(±0.006) | 0.733(±0.006) | 0.736(±0.011) | 0.733(±0.008) | 0.728(±0.011) | 0.719(±0.005) | **0.735(±0.011)** | 0.735(±0.011) |
| E→K | 0.862(±0.009) | 0.874(±0.008) | 0.874(±0.008) | 0.876(±0.009) | 0.873(±0.008) | 0.856(±0.01) | 0.858(±0.012) | **0.873(±0.009)** | 0.863(±0.009) |
| K→B | 0.715(±0.008) | 0.721(±0.004) | 0.721(±0.004) | 0.719(±0.004) | 0.721(±0.01) | 0.714(±0.007) | 0.708(±0.003) | **0.722(±0.007)** | 0.717(±0.007) |
| K→D | 0.745(±0.011) | 0.752(±0.01) | 0.752(±0.01) | 0.747(±0.013) | 0.755(±0.008) | 0.735(±0.018) | 0.742(±0.022) | **0.749(±0.006)** | 0.745(±0.008) |
| K→E | 0.844(±0.009) | 0.854(±0.008) | 0.854(±0.008) | 0.855(±0.005) | 0.857(±0.009) | 0.845(±0.005) | 0.844(±0.006) | **0.856(±0.006)** | 0.849(±0.009) |
| Avg. | 0.769(±0.012) | 0.778(±0.01) | 0.778(±0.01) | 0.778(±0.01) | 0.777(±0.011) | 0.765(±0.011) | 0.766(±0.012) | **0.778(±0.009)** | 0.774(±0.011) |

**AdvSKM**

| Task | SO | Heuristic | | | | Theoretical error guarantees | | | |
|---|---|---|---|---|---|---|---|---|---|
| | | TMV | TMR | TCR | SOR | IWV | DEV | IWA (ours) | TB |
| B→D | 0.784(±0.004) | 0.79(±0.003) | 0.79(±0.003) | 0.789(±0.003) | 0.79(±0.003) | 0.777(±0.008) | 0.778(±0.008) | **0.794(±0.005)** | 0.79(±0.005) |
| B→E | 0.754(±0.009) | 0.761(±0.014) | 0.761(±0.014) | 0.763(±0.012) | 0.762(±0.011) | 0.754(±0.011) | 0.751(±0.007) | **0.761(±0.012)** | 0.756(±0.012) |
| B→K | 0.769(±0.021) | 0.784(±0.019) | 0.784(±0.019) | 0.782(±0.021) | 0.78(±0.016) | 0.771(±0.012) | 0.764(±0.016) | **0.782(±0.017)** | 0.774(±0.019) |
| D→B | 0.779(±0.004) | 0.795(±0.007) | 0.795(±0.007) | 0.794(±0.007) | 0.792(±0.003) | 0.785(±0.007) | 0.78(±0.011) | **0.796(±0.009)** | 0.789(±0.009) |
| D→E | 0.766(±0.013) | 0.785(±0.006) | 0.785(±0.006) | 0.784(±0.008) | 0.777(±0.005) | 0.773(±0.005) | 0.769(±0.009) | **0.785(±0.005)** | 0.776(±0.002) |
| D→K | 0.777(±0.015) | 0.79(±0.015) | 0.79(±0.015) | 0.79(±0.015) | 0.791(±0.013) | 0.775(±0.014) | 0.775(±0.014) | **0.789(±0.013)** | 0.786(±0.013) |
| E→B | 0.688(±0.015) | 0.709(±0.022) | 0.709(±0.022) | 0.705(±0.024) | 0.709(±0.021) | 0.704(±0.022) | 0.695(±0.022) | **0.712(±0.012)** | 0.707(±0.021) |
| E→D | 0.714(±0.005) | 0.734(±0.002) | 0.734(±0.002) | 0.737(±0.005) | 0.733(±0.008) | 0.725(±0.004) | 0.725(±0.004) | **0.732(±0.006)** | 0.737(±0.011) |
| E→K | 0.858(±0.004) | 0.874(±0.01) | 0.874(±0.01) | 0.874(±0.012) | 0.876(±0.01) | 0.86(±0.012) | 0.86(±0.012) | **0.875(±0.011)** | 0.865(±0.01) |
| K→B | 0.713(±0.002) | 0.723(±0.006) | 0.723(±0.006) | 0.724(±0.009) | 0.72(±0.004) | 0.708(±0.014) | 0.705(±0.011) | **0.726(±0.006)** | 0.718(±0.006) |
| K→D | 0.741(±0.01) | 0.752(±0.012) | 0.752(±0.012) | 0.751(±0.013) | 0.752(±0.017) | **0.75(±0.019)** | 0.742(±0.008) | 0.749(±0.012) | 0.756(±0.014) |
| K→E | 0.848(±0.008) | 0.857(±0.007) | 0.857(±0.007) | 0.858(±0.008) | 0.856(±0.008) | 0.846(±0.012) | 0.847(±0.007) | **0.858(±0.008)** | 0.85(±0.008) |
| Avg. | 0.766(±0.009) | 0.779(±0.01) | 0.779(±0.01) | 0.779(±0.011) | 0.778(±0.01) | 0.769(±0.011) | 0.766(±0.011) | **0.78(±0.01)** | 0.775(±0.011) |

**DIRT**

| Task | SO | Heuristic | | | | Theoretical error guarantees | | | |
|---|---|---|---|---|---|---|---|---|---|
| | | TMV | TMR | TCR | SOR | IWV | DEV | IWA (ours) | TB |
| B→D | 0.777(±0.01) | 0.816(±0.008) | 0.816(±0.008) | 0.817(±0.008) | 0.813(±0.007) | 0.791(±0.012) | 0.787(±0.032) | **0.813(±0.004)** | 0.809(±0.008) |
| B→E | 0.756(±0.013) | 0.792(±0.006) | 0.792(±0.006) | 0.792(±0.01) | 0.782(±0.008) | 0.758(±0.011) | 0.758(±0.011) | **0.788(±0.008)** | 0.804(±0.005) |
| B→K | 0.77(±0.013) | 0.815(±0.003) | 0.815(±0.003) | 0.814(±0.005) | 0.813(±0.006) | 0.793(±0.009) | 0.782(±0.025) | **0.814(±0.003)** | 0.814(±0.003) |
| D→B | 0.773(±0.008) | 0.812(±0.014) | 0.812(±0.014) | 0.809(±0.019) | 0.814(±0.01) | 0.805(±0.015) | 0.808(±0.004) | **0.81(±0.01)** | 0.811(±0.011) |
| D→E | 0.757(±0.02) | 0.813(±0.009) | 0.813(±0.009) | 0.814(±0.012) | 0.808(±0.012) | 0.787(±0.028) | 0.807(±0.024) | **0.814(±0.009)** | 0.812(±0.009) |
| D→K | 0.775(±0.016) | 0.836(±0.009) | 0.836(±0.009) | 0.835(±0.008) | 0.837(±0.004) | 0.789(±0.025) | 0.787(±0.038) | **0.832(±0.006)** | 0.828(±0.004) |
| E→B | 0.702(±0.018) | 0.672(±0.015) | 0.672(±0.015) | 0.674(±0.015) | 0.714(±0.016) | **0.702(±0.01)** | 0.675(±0.05) | 0.678(±0.015) | 0.708(±0.016) |
| E→D | 0.716(±0.008) | 0.689(±0.019) | 0.689(±0.019) | 0.69(±0.022) | 0.731(±0.029) | **0.718(±0.01)** | 0.704(±0.027) | 0.702(±0.021) | 0.73(±0.029) |
| E→K | 0.855(±0.013) | 0.888(±0.01) | 0.888(±0.01) | 0.885(±0.009) | 0.894(±0.005) | 0.874(±0.008) | 0.873(±0.009) | **0.884(±0.009)** | 0.889(±0.005) |
| K→B | 0.709(±0.012) | 0.707(±0.007) | 0.707(±0.007) | 0.711(±0.003) | 0.749(±0.005) | **0.73(±0.02)** | 0.715(±0.005) | 0.711(±0.008) | 0.723(±0.005) |
| K→D | 0.736(±0.012) | 0.721(±0.01) | 0.721(±0.01) | 0.727(±0.019) | 0.779(±0.003) | **0.745(±0.022)** | 0.731(±0.007) | 0.727(±0.006) | 0.756(±0.003) |
| K→E | 0.842(±0.009) | 0.869(±0.01) | 0.869(±0.01) | 0.868(±0.01) | 0.872(±0.009) | 0.848(±0.012) | 0.848(±0.008) | **0.868(±0.01)** | 0.863(±0.009) |
| Avg. | 0.764(±0.013) | 0.786(±0.01) | 0.786(±0.01) | 0.786(±0.012) | 0.8(±0.01) | 0.778(±0.016) | 0.773(±0.02) | **0.787(±0.009)** | 0.795(±0.009) |

**DDC**

| Task | SO | Heuristic | | | | Theoretical error guarantees | | | |
|---|---|---|---|---|---|---|---|---|---|
| | | TMV | TMR | TCR | SOR | IWV | DEV | IWA (ours) | TB |
| B→D | 0.781(±0.016) | 0.796(±0.008) | 0.796(±0.008) | 0.793(±0.013) | 0.794(±0.002) | 0.775(±0.017) | 0.778(±0.02) | **0.789(±0.009)** | 0.786(±0.008) |
| B→E | 0.752(±0.007) | 0.762(±0.014) | 0.762(±0.014) | 0.762(±0.013) | 0.766(±0.01) | 0.752(±0.013) | 0.749(±0.006) | **0.763(±0.014)** | 0.756(±0.01) |
| B→K | 0.766(±0.015) | 0.781(±0.015) | 0.781(±0.015) | 0.782(±0.017) | 0.778(±0.02) | 0.776(±0.016) | 0.774(±0.008) | **0.783(±0.014)** | 0.78(±0.014) |
| D→B | 0.781(±0.008) | 0.793(±0.007) | 0.793(±0.007) | 0.791(±0.006) | 0.789(±0.012) | 0.787(±0.009) | 0.783(±0.005) | **0.796(±0.006)** | 0.788(±0.006) |
| D→E | 0.767(±0.015) | 0.782(±0.004) | 0.782(±0.004) | 0.78(±0.004) | 0.783(±0.008) | 0.778(±0.002) | 0.776(±0.004) | **0.784(±0.004)** | 0.778(±0.004) |
| D→K | 0.782(±0.013) | 0.792(±0.015) | 0.792(±0.015) | 0.793(±0.015) | 0.792(±0.018) | 0.782(±0.014) | 0.783(±0.014) | **0.793(±0.017)** | 0.788(±0.017) |
| E→B | 0.693(±0.019) | 0.703(±0.023) | 0.703(±0.023) | 0.706(±0.021) | 0.7(±0.023) | 0.696(±0.011) | 0.689(±0.021) | **0.705(±0.025)** | 0.704(±0.021) |
| E→D | 0.725(±0.006) | 0.736(±0.004) | 0.736(±0.004) | 0.731(±0.006) | 0.735(±0.012) | 0.711(±0.009) | 0.736(±0.004) | **0.736(±0.008)** | 0.736(±0.007) |
| E→K | 0.857(±0.011) | 0.874(±0.009) | 0.874(±0.009) | 0.872(±0.004) | 0.874(±0.006) | 0.857(±0.014) | 0.853(±0.009) | **0.877(±0.011)** | 0.864(±0.011) |
| K→B | 0.713(±0.002) | 0.728(±0.002) | 0.728(±0.002) | 0.727(±0.005) | 0.721(±0.003) | 0.709(±0.015) | 0.708(±0.007) | **0.726(±0.003)** | 0.719(±0.002) |
| K→D | 0.741(±0.018) | 0.752(±0.011) | 0.752(±0.011) | 0.753(±0.01) | 0.752(±0.016) | 0.743(±0.012) | 0.748(±0.01) | **0.755(±0.013)** | 0.748(±0.012) |
| K→E | 0.841(±0.001) | 0.855(±0.011) | 0.855(±0.011) | 0.856(±0.008) | 0.858(±0.008) | 0.84(±0.003) | 0.84(±0.01) | **0.856(±0.008)** | 0.849(±0.008) |
| Avg. | 0.766(±0.011) | 0.78(±0.01) | 0.78(±0.01) | 0.779(±0.01) | 0.778(±0.012) | 0.767(±0.011) | 0.768(±0.01) | **0.78(±0.011)** | 0.775(±0.01) |

Table 16: Mean and standard deviation (after $\pm$) of target classification accuracy on Amazon Reviews (Part 2) over 3 repetitions with different random initialization of model weights.

| | | CMD | | | | | | | |
|---|---|---|---|---|---|---|---|---|---|
| | | Heuristic | | | | Theoretical error guarantees | | | |
| Task | SO | TMV | TMR | TCR | SOR | IWV | DEV | IWA (ours) | TB |
| B → D | 0.772(±0.009) | 0.794(±0.008) | 0.794(±0.008) | 0.794(±0.004) | 0.798(±0.003) | 0.779(±0.018) | 0.748(±0.036) | **0.8(±0.003)** | 0.789(±0.003) |
| B → E | 0.745(±0.012) | 0.785(±0.014) | 0.785(±0.014) | 0.779(±0.011) | 0.779(±0.011) | 0.746(±0.032) | 0.651(±0.077) | **0.782(±0.01)** | 0.78(±0.014) |
| B → K | 0.763(±0.015) | 0.796(±0.01) | 0.796(±0.01) | 0.79(±0.007) | 0.794(±0.013) | 0.787(±0.011) | 0.749(±0.028) | **0.797(±0.007)** | 0.793(±0.008) |
| D → B | 0.788(±0.01) | 0.801(±0.007) | 0.801(±0.007) | 0.798(±0.003) | 0.804(±0.002) | 0.785(±0.011) | 0.754(±0.022) | **0.803(±0.008)** | 0.794(±0.002) |
| D → E | 0.768(±0.003) | 0.804(±0.006) | 0.804(±0.006) | 0.798(±0.003) | 0.801(±0.013) | 0.777(±0.031) | 0.744(±0.095) | **0.802(±0.004)** | 0.798(±0.007) |
| D → K | 0.777(±0.012) | 0.803(±0.009) | 0.803(±0.009) | 0.797(±0.014) | 0.804(±0.009) | 0.773(±0.006) | 0.673(±0.094) | **0.801(±0.011)** | 0.811(±0.007) |
| E → B | 0.699(±0.014) | 0.72(±0.013) | 0.72(±0.013) | 0.717(±0.014) | 0.724(±0.013) | 0.71(±0.016) | 0.68(±0.039) | **0.719(±0.013)** | 0.712(±0.013) |
| E → D | 0.722(±0.008) | 0.752(±0.006) | 0.752(±0.006) | 0.752(±0.004) | 0.751(±0.005) | 0.693(±0.041) | 0.573(±0.01) | **0.755(±0.003)** | 0.738(±0.003) |
| E → K | 0.86(±0.012) | 0.874(±0.013) | 0.874(±0.013) | 0.872(±0.007) | 0.875(±0.008) | 0.822(±0.058) | 0.807(±0.054) | **0.876(±0.009)** | 0.871(±0.009) |
| K → B | 0.718(±0.005) | 0.749(±0.007) | 0.749(±0.007) | 0.738(±0.003) | 0.74(±0.009) | 0.716(±0.022) | 0.679(±0.056) | **0.746(±0.004)** | 0.74(±0.007) |
| K → D | 0.748(±0.003) | 0.761(±0.016) | 0.761(±0.016) | 0.766(±0.022) | 0.77(±0.017) | 0.746(±0.011) | 0.655(±0.119) | **0.775(±0.016)** | 0.761(±0.016) |
| K → E | 0.842(±0.005) | 0.864(±0.006) | 0.864(±0.006) | 0.862(±0.002) | 0.869(±0.004) | 0.85(±0.011) | 0.806(±0.041) | **0.867(±0.008)** | 0.858(±0.004) |
| Avg. | 0.767(±0.009) | 0.792(±0.01) | 0.792(±0.01) | 0.789(±0.008) | 0.792(±0.008) | 0.765(±0.022) | 0.71(±0.056) | **0.794(±0.008)** | 0.787(±0.008) |

| | | MMDA | | | | | | | |
|---|---|---|---|---|---|---|---|---|---|
| | | Heuristic | | | | Theoretical error guarantees | | | |
| Task | SO | TMV | TMR | TCR | SOR | IWV | DEV | IWA (ours) | TB |
| B → D | 0.775(±0.005) | 0.796(±0.005) | 0.796(±0.005) | 0.797(±0.009) | 0.794(±0.015) | 0.783(±0.004) | 0.786(±0.006) | **0.797(±0.003)** | 0.792(±0.004) |
| B → E | 0.752(±0.006) | 0.775(±0.01) | 0.775(±0.01) | 0.776(±0.008) | 0.771(±0.014) | 0.766(±0.022) | 0.756(±0.012) | **0.779(±0.012)** | 0.781(±0.006) |
| B → K | 0.765(±0.009) | 0.793(±0.016) | 0.793(±0.016) | 0.792(±0.017) | 0.803(±0.019) | 0.763(±0.009) | 0.767(±0.024) | **0.795(±0.017)** | 0.802(±0.019) |
| D → B | 0.783(±0.007) | 0.796(±0.007) | 0.796(±0.007) | 0.795(±0.008) | 0.796(±0.006) | 0.789(±0.013) | 0.789(±0.012) | **0.798(±0.009)** | 0.79(±0.009) |
| D → E | 0.764(±0.003) | 0.792(±0.005) | 0.792(±0.005) | 0.792(±0.005) | 0.788(±0.005) | 0.775(±0.009) | 0.773(±0.009) | **0.795(±0.006)** | 0.799(±0.006) |
| D → K | 0.775(±0.012) | 0.797(±0.014) | 0.797(±0.014) | 0.796(±0.013) | 0.811(±0.008) | 0.791(±0.018) | 0.79(±0.019) | **0.8(±0.012)** | 0.8(±0.008) |
| E → B | 0.701(±0.015) | 0.707(±0.017) | 0.707(±0.017) | 0.707(±0.021) | 0.713(±0.014) | 0.688(±0.022) | 0.699(±0.004) | **0.709(±0.016)** | 0.707(±0.014) |
| E → D | 0.738(±0.01) | 0.741(±0.004) | 0.741(±0.004) | 0.744(±0.004) | 0.738(±0.001) | 0.732(±0.007) | 0.703(±0.026) | **0.749(±0.001)** | 0.738(±0.001) |
| E → K | 0.855(±0.015) | 0.875(±0.007) | 0.875(±0.007) | 0.875(±0.007) | 0.878(±0.008) | 0.852(±0.01) | 0.856(±0.014) | **0.878(±0.009)** | 0.867(±0.009) |
| K → B | 0.715(±0.008) | 0.73(±0.003) | 0.73(±0.003) | 0.726(±0.002) | 0.739(±0.006) | 0.72(±0.009) | 0.716(±0.015) | **0.732(±0.004)** | 0.73(±0.006) |
| K → D | 0.736(±0.014) | 0.761(±0.013) | 0.761(±0.013) | 0.755(±0.011) | 0.764(±0.018) | 0.734(±0.012) | 0.728(±0.045) | **0.758(±0.017)** | 0.75(±0.018) |
| K → E | 0.842(±0.008) | 0.856(±0.009) | 0.856(±0.009) | 0.858(±0.009) | 0.855(±0.012) | 0.835(±0.01) | 0.832(±0.014) | **0.858(±0.005)** | 0.847(±0.009) |
| Avg. | 0.767(±0.009) | 0.785(±0.009) | 0.785(±0.009) | 0.785(±0.01) | 0.787(±0.01) | 0.769(±0.012) | 0.766(±0.017) | **0.787(±0.009)** | 0.784(±0.009) |

| | | CoDATS | | | | | | | |
|---|---|---|---|---|---|---|---|---|---|
| | | Heuristic | | | | Theoretical error guarantees | | | |
| Task | SO | TMV | TMR | TCR | SOR | IWV | DEV | IWA (ours) | TB |
| B → D | 0.783(±0.013) | 0.8(±0.002) | 0.8(±0.002) | 0.8(±0.004) | 0.803(±0.003) | 0.788(±0.008) | 0.792(±0.001) | **0.805(±0.009)** | 0.801(±0.008) |
| B → E | 0.755(±0.015) | 0.788(±0.004) | 0.788(±0.004) | 0.787(±0.005) | 0.792(±0.005) | 0.771(±0.011) | 0.771(±0.011) | **0.788(±0.006)** | 0.802(±0.009) |
| B → K | 0.771(±0.022) | 0.808(±0.007) | 0.808(±0.007) | 0.808(±0.008) | 0.807(±0.008) | 0.775(±0.022) | 0.793(±0.023) | **0.806(±0.009)** | 0.817(±0.007) |
| D → B | 0.774(±0.01) | 0.796(±0.005) | 0.796(±0.005) | 0.795(±0.005) | 0.807(±0.007) | 0.791(±0.007) | 0.792(±0.007) | **0.802(±0.004)** | 0.8(±0.007) |
| D → E | 0.769(±0.003) | 0.808(±0.009) | 0.808(±0.009) | 0.81(±0.009) | 0.808(±0.008) | 0.789(±0.02) | 0.777(±0.008) | **0.811(±0.009)** | 0.817(±0.011) |
| D → K | 0.782(±0.021) | 0.819(±0.011) | 0.819(±0.011) | 0.82(±0.01) | 0.824(±0.01) | 0.796(±0.021) | 0.789(±0.011) | **0.82(±0.01)** | 0.828(±0.01) |
| E → B | 0.687(±0.013) | 0.714(±0.019) | 0.714(±0.019) | 0.713(±0.015) | 0.725(±0.021) | 0.7(±0.027) | 0.688(±0.01) | **0.717(±0.018)** | 0.723(±0.021) |
| E → D | 0.72(±0.013) | 0.745(±0.023) | 0.745(±0.023) | 0.744(±0.026) | 0.756(±0.017) | 0.738(±0.025) | 0.731(±0.027) | **0.747(±0.025)** | 0.736(±0.017) |
| E → K | 0.859(±0.016) | 0.883(±0.009) | 0.883(±0.009) | 0.882(±0.008) | 0.879(±0.007) | 0.867(±0.01) | 0.866(±0.01) | **0.883(±0.012)** | 0.87(±0.012) |
| K → B | 0.712(±0.012) | 0.732(±0.004) | 0.732(±0.004) | 0.732(±0.009) | 0.747(±0.012) | 0.726(±0.037) | 0.694(±0.008) | **0.737(±0.006)** | 0.751(±0.016) |
| K → D | 0.73(±0.022) | 0.759(±0.006) | 0.759(±0.006) | 0.766(±0.009) | 0.775(±0.016) | 0.752(±0.015) | 0.738(±0.007) | **0.766(±0.004)** | 0.764(±0.016) |
| K → E | 0.845(±0.0) | 0.867(±0.01) | 0.867(±0.01) | 0.866(±0.011) | 0.868(±0.007) | 0.853(±0.003) | 0.851(±0.011) | **0.867(±0.007)** | 0.856(±0.007) |
| Avg. | 0.766(±0.012) | 0.793(±0.009) | 0.793(±0.009) | 0.794(±0.01) | 0.799(±0.01) | 0.779(±0.017) | 0.773(±0.011) | **0.796(±0.01)** | 0.797(±0.012) |

| | | Deep-Coral | | | | | | | |
|---|---|---|---|---|---|---|---|---|---|
| | | Heuristic | | | | Theoretical error guarantees | | | |
| Task | SO | TMV | TMR | TCR | SOR | IWV | DEV | IWA (ours) | TB |
| B → D | 0.778(±0.018) | 0.799(±0.011) | 0.799(±0.011) | 0.799(±0.006) | 0.794(±0.011) | 0.792(±0.023) | 0.789(±0.025) | **0.801(±0.01)** | 0.794(±0.012) |
| B → E | 0.749(±0.013) | 0.771(±0.01) | 0.771(±0.01) | 0.771(±0.008) | 0.764(±0.018) | 0.759(±0.002) | 0.752(±0.011) | **0.773(±0.008)** | 0.767(±0.008) |
| B → K | 0.769(±0.011) | 0.785(±0.013) | 0.785(±0.013) | 0.784(±0.016) | 0.783(±0.015) | 0.762(±0.013) | 0.77(±0.014) | **0.785(±0.017)** | 0.779(±0.017) |
| D → B | 0.783(±0.009) | 0.796(±0.003) | 0.796(±0.003) | 0.796(±0.006) | 0.792(±0.006) | 0.78(±0.005) | 0.773(±0.006) | **0.797(±0.004)** | 0.786(±0.004) |
| D → E | 0.76(±0.013) | 0.785(±0.002) | 0.785(±0.002) | 0.785(±0.002) | 0.778(±0.006) | 0.762(±0.006) | 0.764(±0.008) | **0.784(±0.003)** | 0.785(±0.005) |
| D → K | 0.76(±0.016) | 0.794(±0.011) | 0.794(±0.011) | 0.793(±0.012) | 0.795(±0.014) | 0.787(±0.007) | 0.783(±0.019) | **0.795(±0.013)** | 0.789(±0.013) |
| E → B | 0.696(±0.009) | 0.708(±0.022) | 0.708(±0.022) | 0.709(±0.024) | 0.708(±0.024) | 0.688(±0.016) | 0.698(±0.018) | **0.71(±0.024)** | 0.708(±0.027) |
| E → D | 0.722(±0.009) | 0.738(±0.005) | 0.738(±0.005) | 0.741(±0.005) | 0.741(±0.001) | 0.73(±0.016) | 0.72(±0.017) | **0.74(±0.004)** | 0.739(±0.001) |
| E → K | 0.859(±0.01) | 0.879(±0.005) | 0.879(±0.005) | 0.878(±0.007) | 0.877(±0.007) | 0.857(±0.007) | 0.858(±0.009) | **0.878(±0.007)** | 0.864(±0.005) |
| K → B | 0.72(±0.01) | 0.733(±0.009) | 0.733(±0.009) | 0.736(±0.009) | 0.733(±0.009) | 0.719(±0.011) | 0.722(±0.022) | **0.739(±0.01)** | 0.74(±0.005) |
| K → D | 0.733(±0.011) | 0.755(±0.012) | 0.755(±0.012) | 0.751(±0.009) | 0.759(±0.009) | 0.75(±0.009) | 0.752(±0.013) | **0.755(±0.015)** | 0.751(±0.009) |
| K → E | 0.841(±0.01) | 0.858(±0.009) | 0.858(±0.009) | 0.858(±0.011) | 0.857(±0.008) | 0.844(±0.011) | 0.849(±0.005) | **0.857(±0.01)** | 0.849(±0.011) |
| Avg. | 0.766(±0.012) | 0.783(±0.009) | 0.783(±0.009) | 0.783(±0.01) | 0.782(±0.011) | 0.769(±0.011) | 0.769(±0.014) | **0.785(±0.01)** | 0.779(±0.01) |

Table 17: Mean and standard deviation (after ±) of target classification accuracy on Amazon Reviews (Part 3) over 3 repetitions with different random initialization of model weights.

| | | | CDAN | | | | | | |
|---|---|---|---|---|---|---|---|---|---|
| | | Heuristic | | | | Theoretical error guarantees | | | |
| Task | SO | TMV | TMR | TCR | SOR | IWV | DEV | IWA (ours) | TB |
| B → D | 0.784(±0.022) | 0.803(±0.007) | 0.803(±0.007) | 0.803(±0.004) | 0.795(±0.009) | 0.795(±0.009) | 0.792(±0.006) | **0.804(±0.009)** | 0.797(±0.006) |
| B → E | 0.758(±0.013) | 0.776(±0.009) | 0.776(±0.009) | 0.774(±0.009) | 0.776(±0.012) | 0.764(±0.012) | 0.764(±0.012) | **0.778(±0.009)** | 0.775(±0.009) |
| B → K | 0.769(±0.019) | 0.796(±0.008) | 0.796(±0.008) | 0.796(±0.008) | 0.792(±0.014) | 0.78(±0.006) | 0.791(±0.007) | **0.796(±0.01)** | 0.796(±0.01) |
| D → B | 0.786(±0.01) | 0.797(±0.007) | 0.797(±0.007) | 0.795(±0.004) | 0.79(±0.004) | 0.793(±0.01) | 0.782(±0.01) | **0.798(±0.007)** | 0.794(±0.008) |
| D → E | 0.761(±0.003) | 0.8(±0.004) | 0.8(±0.004) | 0.8(±0.006) | 0.8(±0.005) | 0.775(±0.007) | 0.77(±0.009) | **0.799(±0.004)** | 0.794(±0.006) |
| D → K | 0.778(±0.018) | 0.797(±0.012) | 0.797(±0.012) | 0.797(±0.01) | 0.804(±0.015) | 0.792(±0.02) | 0.793(±0.015) | **0.801(±0.013)** | 0.801(±0.015) |
| E → B | 0.692(±0.014) | 0.707(±0.015) | 0.707(±0.015) | 0.707(±0.015) | 0.713(±0.024) | 0.695(±0.012) | 0.707(±0.021) | **0.71(±0.018)** | 0.711(±0.024) |
| E → D | 0.72(±0.018) | 0.741(±0.021) | 0.741(±0.021) | 0.738(±0.011) | 0.741(±0.005) | 0.722(±0.017) | 0.723(±0.028) | **0.741(±0.006)** | 0.735(±0.009) |
| E → K | 0.86(±0.006) | 0.879(±0.011) | 0.879(±0.011) | 0.878(±0.01) | 0.874(±0.012) | 0.861(±0.018) | 0.861(±0.018) | **0.879(±0.011)** | 0.865(±0.011) |
| K → B | 0.706(±0.004) | 0.727(±0.003) | 0.727(±0.003) | 0.728(±0.001) | 0.732(±0.005) | 0.724(±0.011) | 0.722(±0.004) | **0.733(±0.004)** | 0.724(±0.004) |
| K → D | 0.748(±0.013) | 0.765(±0.013) | 0.765(±0.013) | 0.764(±0.011) | 0.766(±0.021) | 0.752(±0.021) | 0.755(±0.005) | **0.762(±0.01)** | 0.752(±0.021) |
| K → E | 0.845(±0.007) | 0.86(±0.005) | 0.86(±0.005) | 0.861(±0.007) | 0.863(±0.003) | 0.848(±0.005) | 0.849(±0.007) | **0.861(±0.006)** | 0.85(±0.003) |
| Avg. | 0.767(±0.012) | 0.787(±0.01) | 0.787(±0.01) | 0.787(±0.008) | 0.787(±0.011) | 0.775(±0.012) | 0.776(±0.012) | **0.788(±0.009)** | 0.783(±0.01) |

| | | | DANN | | | | | | |
|---|---|---|---|---|---|---|---|---|---|
| | | Heuristic | | | | Theoretical error guarantees | | | |
| Task | SO | TMV | TMR | TCR | SOR | IWV | DEV | IWA (ours) | TB |
| B → D | 0.783(±0.004) | 0.794(±0.003) | 0.794(±0.003) | 0.794(±0.008) | 0.803(±0.007) | 0.782(±0.005) | 0.782(±0.005) | **0.804(±0.008)** | 0.813(±0.017) |
| B → E | 0.752(±0.011) | 0.787(±0.009) | 0.787(±0.009) | 0.787(±0.013) | 0.796(±0.006) | 0.783(±0.013) | 0.782(±0.013) | **0.789(±0.008)** | 0.8(±0.005) |
| B → K | 0.767(±0.022) | 0.804(±0.007) | 0.804(±0.007) | 0.804(±0.008) | 0.805(±0.004) | 0.796(±0.003) | **0.806(±0.015)** | 0.804(±0.008) | 0.82(±0.004) |
| D → B | 0.779(±0.002) | 0.8(±0.007) | 0.8(±0.007) | 0.799(±0.006) | 0.802(±0.003) | 0.796(±0.004) | 0.778(±0.014) | **0.804(±0.004)** | 0.799(±0.004) |
| D → E | 0.767(±0.016) | 0.805(±0.006) | 0.805(±0.006) | 0.807(±0.009) | 0.808(±0.012) | 0.775(±0.023) | 0.78(±0.035) | **0.81(±0.007)** | 0.815(±0.007) |
| D → K | 0.784(±0.013) | 0.815(±0.008) | 0.815(±0.008) | 0.817(±0.011) | 0.829(±0.014) | 0.788(±0.007) | 0.779(±0.012) | **0.816(±0.011)** | 0.827(±0.014) |
| E → B | 0.701(±0.014) | 0.712(±0.014) | 0.712(±0.014) | 0.712(±0.015) | 0.724(±0.014) | 0.69(±0.03) | 0.702(±0.022) | **0.712(±0.012)** | 0.721(±0.014) |
| E → D | 0.736(±0.005) | 0.743(±0.014) | 0.743(±0.014) | 0.747(±0.012) | 0.758(±0.012) | 0.735(±0.053) | 0.738(±0.019) | **0.751(±0.008)** | 0.757(±0.012) |
| E → K | 0.854(±0.016) | 0.877(±0.011) | 0.877(±0.011) | 0.878(±0.012) | 0.879(±0.014) | 0.86(±0.009) | 0.858(±0.007) | **0.88(±0.013)** | 0.875(±0.012) |
| K → B | 0.711(±0.014) | 0.741(±0.004) | 0.741(±0.004) | 0.74(±0.004) | 0.759(±0.007) | 0.718(±0.012) | 0.729(±0.003) | **0.75(±0.004)** | 0.747(±0.007) |
| K → D | 0.738(±0.006) | 0.768(±0.011) | 0.768(±0.011) | 0.765(±0.011) | 0.778(±0.013) | 0.733(±0.013) | 0.758(±0.019) | **0.778(±0.012)** | 0.762(±0.012) |
| K → E | 0.837(±0.013) | 0.864(±0.008) | 0.864(±0.008) | 0.864(±0.007) | 0.865(±0.009) | 0.86(±0.013) | 0.851(±0.008) | **0.866(±0.011)** | 0.859(±0.011) |
| Avg. | 0.767(±0.012) | 0.792(±0.009) | 0.792(±0.009) | 0.793(±0.01) | 0.8(±0.01) | 0.776(±0.015) | 0.778(±0.015) | **0.797(±0.009)** | 0.8(±0.01) |

| | | | DSAN | | | | | | |
|---|---|---|---|---|---|---|---|---|---|
| | | Heuristic | | | | Theoretical error guarantees | | | |
| Task | SO | TMV | TMR | TCR | SOR | IWV | DEV | IWA (ours) | TB |
| B → D | 0.779(±0.013) | 0.801(±0.014) | 0.801(±0.014) | 0.803(±0.015) | 0.807(±0.01) | 0.79(±0.02) | 0.689(±0.108) | **0.805(±0.014)** | 0.805(±0.01) |
| B → E | 0.752(±0.005) | 0.788(±0.005) | 0.788(±0.005) | 0.783(±0.005) | 0.798(±0.002) | 0.775(±0.022) | 0.764(±0.02) | **0.787(±0.006)** | 0.796(±0.002) |
| B → K | 0.768(±0.012) | 0.799(±0.008) | 0.799(±0.008) | 0.799(±0.01) | 0.815(±0.007) | 0.786(±0.036) | 0.798(±0.031) | **0.801(±0.011)** | 0.816(±0.005) |
| D → B | 0.782(±0.008) | 0.795(±0.004) | 0.795(±0.004) | 0.795(±0.002) | 0.799(±0.007) | 0.797(±0.01) | 0.729(±0.111) | **0.801(±0.005)** | 0.802(±0.007) |
| D → E | 0.771(±0.002) | 0.804(±0.008) | 0.804(±0.008) | 0.804(±0.008) | 0.816(±0.012) | 0.79(±0.02) | 0.792(±0.02) | **0.81(±0.003)** | 0.813(±0.012) |
| D → K | 0.786(±0.013) | 0.804(±0.01) | 0.804(±0.01) | 0.802(±0.012) | 0.821(±0.014) | 0.789(±0.013) | 0.801(±0.025) | **0.807(±0.012)** | 0.83(±0.016) |
| E → B | 0.702(±0.021) | 0.716(±0.017) | 0.716(±0.017) | 0.718(±0.021) | 0.718(±0.018) | 0.71(±0.021) | 0.707(±0.018) | **0.721(±0.02)** | 0.711(±0.02) |
| E → D | 0.725(±0.007) | 0.743(±0.008) | 0.743(±0.008) | 0.741(±0.005) | 0.744(±0.008) | 0.723(±0.021) | 0.735(±0.012) | **0.749(±0.005)** | 0.732(±0.005) |
| E → K | 0.865(±0.013) | 0.883(±0.009) | 0.883(±0.009) | 0.882(±0.008) | 0.886(±0.009) | 0.874(±0.003) | 0.876(±0.006) | **0.883(±0.01)** | 0.876(±0.009) |
| K → B | 0.716(±0.016) | 0.741(±0.01) | 0.741(±0.01) | 0.736(±0.007) | 0.746(±0.011) | 0.712(±0.014) | 0.707(±0.016) | **0.742(±0.008)** | 0.739(±0.011) |
| K → D | 0.739(±0.023) | 0.773(±0.006) | 0.773(±0.006) | 0.768(±0.003) | 0.777(±0.01) | 0.746(±0.016) | 0.726(±0.035) | **0.776(±0.005)** | 0.763(±0.012) |
| K → E | 0.84(±0.01) | 0.86(±0.005) | 0.86(±0.005) | 0.858(±0.007) | 0.868(±0.005) | 0.854(±0.001) | 0.835(±0.04) | **0.865(±0.008)** | 0.857(±0.005) |
| Avg. | 0.769(±0.012) | 0.792(±0.009) | 0.792(±0.009) | 0.791(±0.009) | 0.8(±0.009) | 0.779(±0.016) | 0.763(±0.037) | **0.795(±0.009)** | 0.795(±0.009) |

### D.7.3 DETAILED MINIDOMAINNET RESULTS

Table 18: Mean and standard deviation (after $\pm$) of target classification accuracy on MiniDomainNet dataset over three different random initialization of model weights and five domain adaptation tasks.

| | | MiniDomainNet | | | | | | | |
| --- | --- | --- | --- | --- | --- | --- | --- | --- |
| | | Heuristic | | | | Theoretical error guarantees | | | |
| Method | SO | TMV | TMR | TCR | SOR | IWV | DEV | IWA (ours) | TB |
| HoMM | 0.509($\pm$0.018) | *0.526($\pm$0.009)* | 0.524($\pm$0.010) | 0.523($\pm$0.010) | 0.518($\pm$0.015) | 0.511($\pm$0.018) | 0.511($\pm$0.018) | **0.531($\pm$0.007)** | 0.537($\pm$0.020) |
| AdvSKM | 0.509($\pm$0.015) | *0.516($\pm$0.011)* | 0.514($\pm$0.014) | 0.514($\pm$0.017) | 0.514($\pm$0.009) | 0.515($\pm$0.032) | 0.512($\pm$0.032) | **0.516($\pm$0.011)** | 0.522($\pm$0.017) |
| DIRT | 0.499($\pm$0.022) | *0.517($\pm$0.006)* | 0.515($\pm$0.009) | 0.508($\pm$0.006) | 0.507($\pm$0.018) | 0.493($\pm$0.028) | 0.498($\pm$0.033) | **0.519($\pm$0.006)** | 0.525($\pm$0.020) |
| DDC | 0.510($\pm$0.032) | *0.514($\pm$0.012)* | 0.512($\pm$0.018) | 0.511($\pm$0.019) | 0.514($\pm$0.015) | 0.511($\pm$0.028) | 0.512($\pm$0.043) | **0.516($\pm$0.015)** | 0.521($\pm$0.019) |
| CMD | 0.509($\pm$0.019) | 0.528($\pm$0.011) | *0.531($\pm$0.016)* | 0.531($\pm$0.016) | 0.522($\pm$0.013) | 0.518($\pm$0.025) | 0.490($\pm$0.037) | **0.533($\pm$0.009)** | 0.533($\pm$0.024) |
| MMDA | 0.509($\pm$0.021) | *0.524($\pm$0.004)* | 0.522($\pm$0.005) | 0.523($\pm$0.007) | 0.519($\pm$0.012) | 0.517($\pm$0.008) | 0.524($\pm$0.006) | **0.527($\pm$0.009)** | 0.531($\pm$0.029) |
| CoDATS | 0.502($\pm$0.026) | 0.535($\pm$0.026) | 0.533($\pm$0.027) | *0.536($\pm$0.028)* | 0.525($\pm$0.015) | 0.518($\pm$0.031) | 0.524($\pm$0.051) | **0.536($\pm$0.025)** | 0.529($\pm$0.042) |
| Deep-Coral | 0.505($\pm$0.022) | *0.538($\pm$0.009)* | 0.534($\pm$0.013) | 0.533($\pm$0.012) | 0.520($\pm$0.010) | 0.518($\pm$0.022) | 0.524($\pm$0.034) | **0.539($\pm$0.012)** | 0.535($\pm$0.024) |
| CDAN | 0.514($\pm$0.028) | *0.526($\pm$0.013)* | 0.524($\pm$0.011) | 0.525($\pm$0.008) | 0.517($\pm$0.005) | 0.513($\pm$0.011) | 0.516($\pm$0.014) | **0.532($\pm$0.010)** | 0.542($\pm$0.017) |
| DANN | 0.496($\pm$0.023) | 0.530($\pm$0.011) | 0.535($\pm$0.011) | *0.538($\pm$0.005)* | 0.516($\pm$0.011) | 0.519($\pm$0.015) | 0.515($\pm$0.013) | **0.541($\pm$0.006)** | 0.532($\pm$0.014) |
| DSAN | 0.509($\pm$0.022) | 0.537($\pm$0.011) | 0.534($\pm$0.013) | *0.543($\pm$0.011)* | 0.525($\pm$0.013) | 0.513($\pm$0.018) | 0.535($\pm$0.023) | **0.546($\pm$0.007)** | 0.563($\pm$0.013) |
| Avg. | 0.507($\pm$0.022) | *0.526($\pm$0.011)* | 0.525($\pm$0.014) | 0.526($\pm$0.013) | 0.518($\pm$0.012) | 0.513($\pm$0.022) | 0.515($\pm$0.028) | **0.531($\pm$0.011)** | 0.534($\pm$0.022) |

Table 19: Mean and standard deviation (after ±) of target classification accuracy on MiniDomainNet (Part 1) over 3 repetitions with different random initialization of model weights.

| | | HoMM | | | | | | | |
|---|---|---|---|---|---|---|---|---|---|
| | | Heuristic | | | | Theoretical error guarantees | | | |
| Task | SO | TMV | TMR | TCR | SOR | IWV | DEV | IWA (ours) | TB |
| R → C | 0.552(±0.041) | 0.587(±0.017) | 0.587(±0.017) | 0.587(±0.005) | 0.557(±0.008) | 0.538(±0.029) | 0.56(±0.017) | **0.596(±0.009)** | 0.607(±0.022) |
| R → I | 0.373(±0.014) | 0.376(±0.012) | 0.376(±0.012) | 0.373(±0.005) | 0.386(±0.007) | 0.37(±0.01) | 0.37(±0.01) | **0.392(±0.007)** | 0.413(±0.041) |
| R → P | 0.709(±0.026) | 0.722(±0.01) | 0.722(±0.01) | 0.718(±0.006) | 0.709(±0.011) | 0.704(±0.013) | 0.705(±0.014) | **0.73(±0.003)** | 0.721(±0.003) |
| R → Q | 0.332(±0.009) | 0.35(±0.006) | 0.35(±0.006) | 0.353(±0.008) | 0.356(±0.02) | **0.367(±0.019)** | 0.342(±0.018) | 0.353(±0.003) | 0.37(±0.013) |
| R → S | 0.579(±0.022) | 0.587(±0.012) | 0.587(±0.012) | 0.586(±0.003) | 0.582(±0.01) | 0.577(±0.004) | 0.577(±0.004) | **0.586(±0.008)** | 0.609(±0.011) |
| Avg. | 0.509(±0.023) | 0.524(±0.011) | 0.524(±0.011) | 0.523(±0.005) | 0.518(±0.011) | 0.511(±0.015) | 0.511(±0.013) | **0.531(±0.006)** | 0.544(±0.018) |

| | | AdvSKM | | | | | | | |
|---|---|---|---|---|---|---|---|---|---|
| | | Heuristic | | | | Theoretical error guarantees | | | |
| Task | SO | TMV | TMR | TCR | SOR | IWV | DEV | IWA (ours) | TB |
| R → C | 0.544(±0.017) | 0.56(±0.025) | 0.56(±0.025) | 0.566(±0.03) | 0.544(±0.024) | 0.549(±0.022) | 0.549(±0.022) | **0.568(±0.009)** | 0.582(±0.046) |
| R → I | 0.38(±0.018) | 0.373(±0.016) | 0.373(±0.016) | 0.371(±0.009) | 0.394(±0.037) | 0.394(±0.021) | **0.394(±0.021)** | 0.377(±0.013) | 0.395(±0.02) |
| R → P | 0.723(±0.009) | 0.721(±0.004) | 0.721(±0.004) | 0.719(±0.003) | 0.712(±0.006) | 0.706(±0.041) | 0.695(±0.034) | **0.721(±0.003)** | 0.723(±0.009) |
| R → Q | 0.322(±0.027) | 0.332(±0.006) | 0.332(±0.006) | 0.328(±0.005) | 0.335(±0.002) | **0.335(±0.002)** | 0.332(±0.006) | 0.333(±0.005) | 0.335(±0.004) |
| R → S | 0.579(±0.017) | 0.586(±0.001) | 0.586(±0.001) | 0.584(±0.003) | 0.586(±0.005) | 0.589(±0.007) | **0.589(±0.007)** | 0.582(±0.004) | 0.592(±0.009) |
| Avg. | 0.509(±0.018) | 0.514(±0.01) | 0.514(±0.01) | 0.514(±0.01) | 0.514(±0.015) | 0.515(±0.018) | 0.512(±0.018) | **0.516(±0.007)** | 0.525(±0.018) |

| | | DIRT | | | | | | | |
|---|---|---|---|---|---|---|---|---|---|
| | | Heuristic | | | | Theoretical error guarantees | | | |
| Task | SO | TMV | TMR | TCR | SOR | IWV | DEV | IWA (ours) | TB |
| R → C | 0.53(±0.013) | 0.577(±0.033) | 0.577(±0.033) | 0.546(±0.031) | 0.571(±0.017) | 0.541(±0.025) | 0.544(±0.025) | **0.577(±0.025)** | 0.587(±0.037) |
| R → I | 0.373(±0.038) | 0.365(±0.041) | 0.365(±0.041) | 0.374(±0.037) | 0.37(±0.007) | 0.36(±0.011) | **0.385(±0.062)** | 0.38(±0.031) | 0.431(±0.045) |
| R → P | 0.704(±0.031) | 0.708(±0.026) | 0.708(±0.026) | 0.706(±0.02) | 0.689(±0.025) | 0.663(±0.042) | | **0.713(±0.03)** | 0.715(±0.028) |
| R → Q | 0.34(±0.02) | 0.345(±0.031) | 0.345(±0.031) | 0.34(±0.046) | 0.336(±0.024) | 0.326(±0.086) | **0.365(±0.038)** | 0.342(±0.022) | 0.382(±0.027) |
| R → S | 0.55(±0.028) | 0.578(±0.006) | 0.578(±0.006) | 0.571(±0.006) | 0.571(±0.012) | 0.572(±0.005) | 0.525(±0.088) | **0.583(±0.016)** | 0.588(±0.033) |
| Avg. | 0.499(±0.026) | 0.515(±0.027) | 0.515(±0.027) | 0.508(±0.028) | 0.507(±0.015) | 0.493(±0.031) | 0.498(±0.051) | **0.519(±0.025)** | 0.541(±0.034) |

| | | DDC | | | | | | | |
|---|---|---|---|---|---|---|---|---|---|
| | | Heuristic | | | | Theoretical error guarantees | | | |
| Task | SO | TMV | TMR | TCR | SOR | IWV | DEV | IWA (ours) | TB |
| R → C | 0.555(±0.045) | 0.555(±0.013) | 0.555(±0.013) | 0.549(±0.008) | 0.552(±0.009) | 0.546(±0.013) | 0.546(±0.013) | **0.568(±0.025)** | 0.585(±0.076) |
| R → I | 0.374(±0.014) | 0.391(±0.003) | 0.391(±0.003) | 0.38(±0.014) | 0.398(±0.024) | 0.379(±0.009) | 0.379(±0.009) | **0.385(±0.008)** | 0.416(±0.014) |
| R → P | 0.709(±0.025) | 0.714(±0.0) | 0.714(±0.0) | 0.712(±0.004) | 0.709(±0.004) | 0.713(±0.006) | **0.718(±0.003)** | 0.715(±0.002) | 0.717(±0.004) |
| R → Q | 0.334(±0.002) | 0.331(±0.003) | 0.331(±0.003) | 0.333(±0.0) | 0.327(±0.018) | 0.333(±0.005) | **0.335(±0.002)** | 0.335(±0.003) | 0.337(±0.004) |
| R → S | 0.576(±0.022) | 0.571(±0.01) | 0.571(±0.01) | 0.578(±0.008) | 0.582(±0.005) | 0.582(±0.005) | **0.582(±0.005)** | 0.579(±0.007) | 0.585(±0.021) |
| Avg. | 0.51(±0.021) | 0.512(±0.005) | 0.512(±0.005) | 0.511(±0.007) | 0.514(±0.012) | 0.511(±0.008) | 0.512(±0.006) | **0.516(±0.009)** | 0.528(±0.024) |

| | | CMD | | | | | | | |
|---|---|---|---|---|---|---|---|---|---|
| | | Heuristic | | | | Theoretical error guarantees | | | |
| Task | SO | TMV | TMR | TCR | SOR | IWV | DEV | IWA (ours) | TB |
| R → C | 0.555(±0.042) | 0.598(±0.008) | 0.598(±0.008) | 0.59(±0.008) | 0.544(±0.021) | 0.552(±0.031) | 0.443(±0.067) | **0.593(±0.009)** | 0.617(±0.025) |
| R → I | 0.377(±0.009) | 0.368(±0.007) | 0.368(±0.007) | 0.371(±0.005) | 0.395(±0.007) | 0.379(±0.011) | 0.379(±0.007) | **0.379(±0.007)** | 0.389(±0.007) |
| R → P | 0.704(±0.02) | 0.741(±0.015) | 0.741(±0.015) | 0.733(±0.01) | 0.723(±0.02) | 0.732(±0.019) | 0.736(±0.013) | **0.74(±0.004)** | 0.734(±0.015) |
| R → Q | 0.332(±0.015) | 0.365(±0.005) | 0.365(±0.005) | 0.374(±0.003) | 0.361(±0.036) | 0.338(±0.059) | 0.303(±0.061) | **0.364(±0.007)** | 0.378(±0.004) |
| R → S | 0.578(±0.022) | 0.582(±0.01) | 0.582(±0.01) | 0.587(±0.005) | 0.589(±0.005) | 0.588(±0.019) | 0.591(±0.015) | **0.592(±0.003)** | 0.59(±0.003) |
| Avg. | 0.509(±0.022) | 0.531(±0.009) | 0.531(±0.009) | 0.531(±0.006) | 0.522(±0.018) | 0.518(±0.028) | 0.49(±0.033) | **0.533(±0.006)** | 0.542(±0.012) |

| | | MMDA | | | | | | | |
|---|---|---|---|---|---|---|---|---|---|
| | | Heuristic | | | | Theoretical error guarantees | | | |
| Task | SO | TMV | TMR | TCR | SOR | IWV | DEV | IWA (ours) | TB |
| R → C | 0.555(±0.042) | 0.587(±0.009) | 0.587(±0.009) | 0.571(±0.013) | 0.56(±0.019) | 0.544(±0.033) | 0.544(±0.033) | **0.585(±0.009)** | 0.585(±0.009) |
| R → I | 0.377(±0.009) | 0.379(±0.007) | 0.379(±0.007) | 0.376(±0.012) | 0.389(±0.008) | 0.389(±0.021) | **0.389(±0.021)** | 0.373(±0.003) | 0.415(±0.016) |
| R → P | 0.704(±0.02) | 0.714(±0.014) | 0.714(±0.014) | 0.714(±0.011) | 0.717(±0.014) | 0.713(±0.003) | 0.721(±0.015) | **0.723(±0.004)** | 0.735(±0.007) |
| R → Q | 0.332(±0.015) | 0.352(±0.023) | 0.352(±0.023) | 0.372(±0.006) | 0.335(±0.018) | 0.353(±0.025) | **0.38(±0.035)** | 0.362(±0.009) | 0.392(±0.018) |
| R → S | 0.578(±0.022) | 0.581(±0.011) | 0.581(±0.011) | 0.583(±0.013) | 0.592(±0.004) | 0.587(±0.01) | 0.587(±0.01) | **0.593(±0.009)** | 0.605(±0.008) |
| Avg. | 0.509(±0.022) | 0.522(±0.013) | 0.522(±0.013) | 0.523(±0.011) | 0.519(±0.013) | 0.517(±0.018) | 0.524(±0.023) | **0.527(±0.007)** | 0.546(±0.012) |

Table 20: Mean and standard deviation (after ±) of target classification accuracy on MiniDomainNet (Part 2) over 3 repetitions with different random initialization of model weights.

| CoDATS | | | | | | | | | |
|---|---|---|---|---|---|---|---|---|---|
| | | Heuristic | | | | Theoretical error guarantees | | | |
| Task | SO | TMV | TMR | TCR | SOR | IWV | DEV | IWA (ours) | TB |
| R → C | 0.538(±0.072) | 0.585(±0.025) | 0.585(±0.025) | 0.607(±0.028) | 0.585(±0.013) | 0.555(±0.031) | **0.626(±0.058)** | 0.577(±0.026) | 0.617(±0.058) |
| R → I | 0.365(±0.037) | 0.371(±0.024) | 0.371(±0.024) | 0.367(±0.016) | 0.38(±0.02) | **0.373(±0.041)** | 0.363(±0.056) | 0.371(±0.012) | 0.406(±0.021) |
| R → P | 0.689(±0.024) | 0.735(±0.01) | 0.735(±0.01) | 0.734(±0.015) | 0.726(±0.014) | 0.689(±0.043) | 0.689(±0.043) | **0.743(±0.016)** | 0.732(±0.016) |
| R → Q | 0.322(±0.022) | 0.356(±0.02) | 0.356(±0.02) | 0.358(±0.017) | 0.34(±0.017) | **0.384(±0.012)** | 0.348(±0.044) | 0.364(±0.012) | 0.417(±0.025) |
| R → S | 0.597(±0.004) | 0.619(±0.013) | 0.619(±0.013) | 0.616(±0.018) | 0.593(±0.01) | 0.592(±0.013) | 0.592(±0.013) | **0.622(±0.009)** | 0.627(±0.008) |
| Avg. | 0.502(±0.032) | 0.533(±0.018) | 0.533(±0.018) | 0.536(±0.019) | 0.525(±0.015) | 0.518(±0.028) | 0.524(±0.043) | **0.536(±0.015)** | 0.56(±0.026) |

| Deep-Coral | | | | | | | | | |
|---|---|---|---|---|---|---|---|---|---|
| | | Heuristic | | | | Theoretical error guarantees | | | |
| Task | SO | TMV | TMR | TCR | SOR | IWV | DEV | IWA (ours) | TB |
| R → C | 0.555(±0.045) | 0.585(±0.017) | 0.585(±0.017) | 0.582(±0.014) | 0.557(±0.0) | 0.536(±0.025) | 0.568(±0.013) | **0.59(±0.022)** | 0.601(±0.056) |
| R → I | 0.376(±0.012) | 0.37(±0.01) | 0.37(±0.01) | 0.368(±0.01) | 0.38(±0.005) | 0.368(±0.003) | **0.368(±0.003)** | 0.367(±0.012) | 0.386(±0.007) |
| R → P | 0.711(±0.021) | 0.726(±0.015) | 0.726(±0.015) | 0.723(±0.007) | 0.714(±0.007) | 0.722(±0.007) | 0.722(±0.012) | **0.738(±0.012)** | 0.734(±0.012) |
| R → Q | 0.316(±0.041) | 0.383(±0.003) | 0.383(±0.003) | 0.386(±0.004) | 0.365(±0.008) | 0.378(±0.011) | 0.375(±0.033) | **0.38(±0.002)** | 0.388(±0.002) |
| R → S | 0.569(±0.02) | 0.606(±0.009) | 0.606(±0.009) | 0.606(±0.005) | 0.581(±0.007) | 0.588(±0.008) | 0.588(±0.008) | **0.622(±0.001)** | 0.658(±0.035) |
| Avg. | 0.505(±0.028) | 0.534(±0.011) | 0.534(±0.011) | 0.533(±0.008) | 0.52(±0.005) | 0.518(±0.011) | 0.524(±0.014) | **0.539(±0.01)** | 0.554(±0.022) |

| CDAN | | | | | | | | | |
|---|---|---|---|---|---|---|---|---|---|
| | | Heuristic | | | | Theoretical error guarantees | | | |
| Task | SO | TMV | TMR | TCR | SOR | IWV | DEV | IWA (ours) | TB |
| R → C | 0.557(±0.008) | 0.587(±0.013) | 0.587(±0.013) | 0.593(±0.026) | 0.563(±0.009) | 0.552(±0.025) | 0.574(±0.05) | **0.598(±0.022)** | 0.615(±0.008) |
| R → I | 0.386(±0.01) | 0.359(±0.005) | 0.359(±0.005) | 0.357(±0.009) | 0.368(±0.005) | **0.363(±0.04)** | 0.335(±0.016) | 0.357(±0.005) | 0.386(±0.01) |
| R → P | 0.709(±0.022) | 0.732(±0.02) | 0.732(±0.02) | 0.734(±0.023) | 0.71(±0.006) | 0.72(±0.039) | 0.723(±0.036) | **0.731(±0.004)** | 0.716(±0.023) |
| R → Q | 0.342(±0.006) | 0.352(±0.024) | 0.352(±0.024) | 0.349(±0.02) | 0.347(±0.016) | 0.356(±0.034) | **0.38(±0.028)** | 0.364(±0.013) | 0.442(±0.114) |
| R → S | 0.576(±0.027) | 0.588(±0.01) | 0.588(±0.01) | 0.593(±0.005) | 0.596(±0.01) | 0.574(±0.022) | 0.568(±0.03) | **0.61(±0.008)** | 0.651(±0.056) |
| Avg. | 0.514(±0.015) | 0.524(±0.014) | 0.524(±0.014) | 0.525(±0.017) | 0.517(±0.009) | 0.513(±0.032) | 0.516(±0.032) | **0.532(±0.011)** | 0.562(±0.042) |

| DANN | | | | | | | | | |
|---|---|---|---|---|---|---|---|---|---|
| | | Heuristic | | | | Theoretical error guarantees | | | |
| Task | SO | TMV | TMR | TCR | SOR | IWV | DEV | IWA (ours) | TB |
| R → C | 0.519(±0.013) | 0.626(±0.021) | 0.626(±0.021) | 0.623(±0.036) | 0.557(±0.014) | 0.582(±0.022) | 0.56(±0.07) | **0.607(±0.008)** | 0.631(±0.038) |
| R → I | 0.376(±0.016) | 0.37(±0.013) | 0.37(±0.013) | 0.377(±0.013) | 0.389(±0.032) | 0.379(±0.036) | 0.379(±0.036) | **0.388(±0.019)** | 0.4(±0.025) |
| R → P | 0.705(±0.011) | 0.704(±0.013) | 0.704(±0.013) | 0.704(±0.011) | 0.696(±0.006) | 0.699(±0.006) | 0.697(±0.014) | **0.721(±0.008)** | 0.713(±0.008) |
| R → Q | 0.3(±0.043) | 0.377(±0.008) | 0.377(±0.008) | 0.38(±0.006) | 0.343(±0.004) | 0.351(±0.048) | 0.351(±0.048) | **0.374(±0.003)** | 0.392(±0.005) |
| R → S | 0.581(±0.014) | 0.598(±0.023) | 0.598(±0.023) | 0.607(±0.016) | 0.596(±0.01) | 0.586(±0.015) | 0.586(±0.015) | **0.617(±0.005)** | 0.657(±0.015) |
| Avg. | 0.496(±0.019) | 0.535(±0.016) | 0.535(±0.016) | 0.538(±0.016) | 0.516(±0.013) | 0.519(±0.025) | 0.515(±0.037) | **0.541(±0.009)** | 0.559(±0.018) |

| DSAN | | | | | | | | | |
|---|---|---|---|---|---|---|---|---|---|
| | | Heuristic | | | | Theoretical error guarantees | | | |
| Task | SO | TMV | TMR | TCR | SOR | IWV | DEV | IWA (ours) | TB |
| R → C | 0.555(±0.042) | 0.637(±0.021) | 0.637(±0.021) | 0.639(±0.022) | 0.566(±0.016) | 0.557(±0.059) | 0.571(±0.045) | **0.639(±0.008)** | 0.648(±0.014) |
| R → I | 0.377(±0.009) | 0.363(±0.01) | 0.363(±0.01) | 0.374(±0.015) | 0.406(±0.005) | 0.362(±0.034) | 0.362(±0.034) | **0.377(±0.016)** | 0.404(±0.005) |
| R → P | 0.705(±0.021) | 0.712(±0.009) | 0.712(±0.009) | 0.714(±0.005) | 0.72(±0.008) | 0.716(±0.003) | 0.719(±0.003) | **0.719(±0.007)** | 0.716(±0.008) |
| R → Q | 0.332(±0.012) | 0.364(±0.021) | 0.364(±0.021) | 0.38(±0.012) | 0.34(±0.016) | 0.342(±0.009) | **0.437(±0.083)** | 0.372(±0.011) | 0.478(±0.013) |
| R → S | 0.577(±0.026) | 0.594(±0.004) | 0.594(±0.004) | 0.607(±0.005) | 0.594(±0.003) | 0.585(±0.004) | 0.585(±0.004) | **0.625(±0.016)** | 0.639(±0.052) |
| Avg. | 0.509(±0.022) | 0.534(±0.013) | 0.534(±0.013) | 0.543(±0.012) | 0.525(±0.01) | 0.513(±0.022) | 0.535(±0.034) | **0.546(±0.012)** | 0.577(±0.018) |

### D.7.4 DETAILED TIME-SERIES RESULTS

Table 21: Mean and standard deviation (after ±) of target classification accuracy on Sleep-EDF (Part 1) over 3 repetitions with different random initialization of model weights.

| | | HoMM | | | | | | | |
|---|---|---|---|---|---|---|---|---|---|
| | | | Heuristic | | | | Theoretical error guarantees | | |
| Task | SO | TMV | TMR | TCR | SOR | IWV | DEV | IWA (ours) | TB |
| 0 → 11 | 0.565(±0.033) | 0.557(±0.066) | 0.557(±0.066) | 0.553(±0.068) | *0.72(±0.052)* | 0.694(±0.075) | 0.488(±0.09) | **0.719(±0.04)** | 0.654(±0.052) |
| 12 → 5 | 0.698(±0.081) | 0.771(±0.014) | 0.771(±0.014) | 0.768(±0.008) | *0.788(±0.022)* | 0.776(±0.026) | **0.776(±0.026)** | 0.762(±0.022) | 0.811(±0.013) |
| 16 → 1 | 0.664(±0.068) | *0.702(±0.02)* | 0.702(±0.02) | 0.698(±0.025) | 0.616(±0.017) | 0.628(±0.017) | 0.628(±0.017) | **0.682(±0.008)** | 0.729(±0.025) |
| 7 → 18 | 0.712(±0.024) | 0.734(±0.021) | 0.734(±0.021) | *0.736(±0.012)* | 0.678(±0.006) | 0.729(±0.025) | 0.734(±0.028) | **0.74(±0.008)** | 0.755(±0.02) |
| 9 → 14 | 0.741(±0.031) | 0.832(±0.008) | 0.832(±0.008) | *0.833(±0.01)* | 0.82(±0.024) | 0.802(±0.025) | 0.764(±0.07) | **0.832(±0.01)** | 0.814(±0.01) |
| Avg. | 0.676(±0.047) | 0.719(±0.026) | 0.719(±0.026) | 0.718(±0.025) | *0.724(±0.024)* | 0.726(±0.034) | 0.678(±0.046) | **0.747(±0.018)** | 0.753(±0.024) |

| | | AdvSKM | | | | | | | |
|---|---|---|---|---|---|---|---|---|---|
| | | | Heuristic | | | | Theoretical error guarantees | | |
| Task | SO | TMV | TMR | TCR | SOR | IWV | DEV | IWA (ours) | TB |
| 0 → 11 | 0.611(±0.005) | 0.59(±0.02) | 0.59(±0.02) | 0.589(±0.025) | *0.706(±0.043)* | 0.645(±0.038) | 0.595(±0.035) | **0.664(±0.05)** | 0.611(±0.043) |
| 12 → 5 | 0.723(±0.061) | 0.75(±0.02) | 0.75(±0.02) | 0.747(±0.011) | *0.777(±0.024)* | 0.753(±0.015) | 0.753(±0.015) | **0.754(±0.027)** | 0.777(±0.051) |
| 16 → 1 | 0.558(±0.035) | 0.667(±0.037) | 0.667(±0.037) | *0.671(±0.035)* | 0.619(±0.043) | 0.65(±0.076) | 0.65(±0.076) | **0.658(±0.03)** | 0.73(±0.033) |
| 7 → 18 | 0.651(±0.053) | *0.72(±0.022)* | 0.72(±0.022) | 0.72(±0.018) | 0.669(±0.018) | 0.706(±0.041) | 0.681(±0.023) | **0.707(±0.004)** | 0.723(±0.033) |
| 9 → 14 | 0.78(±0.027) | *0.835(±0.016)* | 0.835(±0.016) | 0.835(±0.016) | 0.819(±0.032) | 0.76(±0.059) | 0.78(±0.027) | **0.824(±0.014)** | 0.789(±0.016) |
| Avg. | 0.665(±0.036) | 0.712(±0.023) | 0.712(±0.023) | 0.712(±0.021) | *0.718(±0.032)* | 0.703(±0.046) | 0.692(±0.035) | **0.722(±0.025)** | 0.726(±0.035) |

| | | DIRT | | | | | | | |
|---|---|---|---|---|---|---|---|---|---|
| | | | Heuristic | | | | Theoretical error guarantees | | |
| Task | SO | TMV | TMR | TCR | SOR | IWV | DEV | IWA (ours) | TB |
| 0 → 11 | 0.582(±0.041) | 0.493(±0.106) | 0.493(±0.106) | 0.509(±0.094) | *0.625(±0.031)* | 0.447(±0.123) | 0.487(±0.188) | **0.54(±0.043)** | 0.582(±0.031) |
| 12 → 5 | 0.712(±0.08) | 0.835(±0.015) | 0.835(±0.015) | 0.832(±0.017) | *0.87(±0.019)* | 0.805(±0.024) | 0.789(±0.015) | **0.829(±0.02)** | 0.857(±0.019) |
| 16 → 1 | 0.526(±0.162) | *0.757(±0.025)* | 0.757(±0.025) | 0.755(±0.021) | 0.671(±0.071) | 0.618(±0.124) | 0.618(±0.124) | **0.746(±0.03)** | 0.797(±0.02) |
| 7 → 18 | 0.749(±0.041) | *0.777(±0.018)* | 0.777(±0.018) | 0.775(±0.005) | 0.717(±0.044) | 0.74(±0.069) | 0.719(±0.072) | **0.762(±0.0)** | 0.789(±0.02) |
| 9 → 14 | 0.711(±0.098) | 0.862(±0.016) | 0.862(±0.016) | *0.871(±0.014)* | 0.826(±0.027) | 0.786(±0.071) | 0.818(±0.045) | **0.868(±0.012)** | 0.874(±0.022) |
| Avg. | 0.656(±0.084) | 0.745(±0.036) | 0.745(±0.036) | *0.748(±0.03)* | 0.742(±0.039) | 0.679(±0.09) | 0.686(±0.107) | **0.749(±0.021)** | 0.78(±0.023) |

| | | DDC | | | | | | | |
|---|---|---|---|---|---|---|---|---|---|
| | | | Heuristic | | | | Theoretical error guarantees | | |
| Task | SO | TMV | TMR | TCR | SOR | IWV | DEV | IWA (ours) | TB |
| 0 → 11 | 0.546(±0.033) | 0.603(±0.029) | 0.603(±0.029) | 0.598(±0.027) | *0.658(±0.06)* | 0.629(±0.059) | 0.492(±0.032) | **0.645(±0.095)** | 0.641(±0.06) |
| 12 → 5 | 0.695(±0.075) | 0.751(±0.01) | 0.751(±0.01) | 0.749(±0.012) | *0.779(±0.029)* | 0.716(±0.046) | 0.716(±0.046) | **0.755(±0.015)** | 0.783(±0.013) |
| 16 → 1 | 0.551(±0.091) | *0.683(±0.013)* | 0.683(±0.013) | 0.672(±0.005) | 0.56(±0.028) | 0.618(±0.021) | 0.623(±0.012) | **0.666(±0.017)** | 0.728(±0.051) |
| 7 → 18 | 0.699(±0.022) | *0.729(±0.038)* | 0.729(±0.038) | 0.717(±0.018) | 0.674(±0.009) | 0.724(±0.029) | 0.712(±0.036) | **0.733(±0.013)** | 0.743(±0.033) |
| 9 → 14 | 0.737(±0.035) | *0.84(±0.014)* | 0.84(±0.014) | 0.827(±0.025) | 0.802(±0.014) | 0.784(±0.01) | 0.788(±0.042) | **0.823(±0.016)** | 0.792(±0.014) |
| Avg. | 0.646(±0.051) | *0.721(±0.021)* | 0.721(±0.021) | 0.712(±0.018) | 0.695(±0.028) | 0.694(±0.033) | 0.666(±0.034) | **0.724(±0.031)** | 0.737(±0.034) |

| | | CMD | | | | | | | |
|---|---|---|---|---|---|---|---|---|---|
| | | | Heuristic | | | | Theoretical error guarantees | | |
| Task | SO | TMV | TMR | TCR | SOR | IWV | DEV | IWA (ours) | TB |
| 0 → 11 | 0.536(±0.041) | 0.564(±0.01) | 0.564(±0.01) | 0.551(±0.027) | *0.655(±0.079)* | **0.607(±0.087)** | 0.505(±0.016) | 0.581(±0.009) | 0.611(±0.079) |
| 12 → 5 | 0.751(±0.032) | *0.805(±0.012)* | 0.805(±0.012) | 0.79(±0.016) | 0.798(±0.016) | 0.784(±0.03) | 0.629(±0.044) | **0.793(±0.014)** | 0.835(±0.02) |
| 16 → 1 | 0.557(±0.099) | *0.747(±0.011)* | 0.747(±0.011) | 0.701(±0.034) | 0.624(±0.018) | 0.678(±0.005) | 0.665(±0.018) | **0.69(±0.015)** | 0.749(±0.019) |
| 7 → 18 | 0.681(±0.047) | 0.715(±0.017) | 0.715(±0.017) | *0.727(±0.004)* | 0.669(±0.019) | **0.741(±0.005)** | 0.728(±0.025) | 0.738(±0.004) | 0.75(±0.026) |
| 9 → 14 | 0.74(±0.07) | *0.85(±0.008)* | 0.85(±0.008) | 0.848(±0.014) | 0.797(±0.021) | 0.772(±0.064) | 0.672(±0.229) | **0.841(±0.006)** | 0.841(±0.008) |
| Avg. | 0.653(±0.058) | *0.736(±0.012)* | 0.736(±0.012) | 0.723(±0.019) | 0.709(±0.031) | 0.716(±0.038) | 0.64(±0.066) | **0.729(±0.01)** | 0.757(±0.03) |

| | | MMDA | | | | | | | |
|---|---|---|---|---|---|---|---|---|---|
| | | | Heuristic | | | | Theoretical error guarantees | | |
| Task | SO | TMV | TMR | TCR | SOR | IWV | DEV | IWA (ours) | TB |
| 0 → 11 | 0.538(±0.032) | 0.52(±0.01) | 0.52(±0.01) | 0.523(±0.007) | *0.654(±0.025)* | 0.596(±0.086) | 0.405(±0.099) | **0.612(±0.037)** | 0.615(±0.025) |
| 12 → 5 | 0.749(±0.034) | 0.797(±0.016) | 0.797(±0.016) | 0.79(±0.025) | *0.798(±0.006)* | 0.698(±0.079) | 0.747(±0.076) | **0.793(±0.01)** | 0.823(±0.025) |
| 16 → 1 | 0.544(±0.116) | *0.705(±0.015)* | 0.705(±0.015) | 0.7(±0.022) | 0.645(±0.032) | **0.703(±0.02)** | 0.692(±0.008) | 0.7(±0.014) | 0.739(±0.061) |
| 7 → 18 | 0.678(±0.046) | *0.773(±0.01)* | 0.773(±0.01) | 0.762(±0.0) | 0.68(±0.051) | 0.729(±0.094) | 0.702(±0.082) | **0.772(±0.005)** | 0.797(±0.004) |
| 9 → 14 | 0.742(±0.071) | *0.84(±0.029)* | 0.84(±0.029) | 0.839(±0.027) | 0.793(±0.014) | 0.792(±0.073) | 0.755(±0.051) | **0.85(±0.012)** | 0.835(±0.012) |
| Avg. | 0.65(±0.06) | *0.727(±0.016)* | 0.727(±0.016) | 0.723(±0.016) | 0.714(±0.026) | 0.704(±0.07) | 0.66(±0.063) | **0.745(±0.016)** | 0.762(±0.025) |

Table 22: Mean and standard deviation (after ±) of target classification accuracy on Sleep-EDF (Part 2) over 3 repetitions with different random initialization of model weights.

**CoDATS**

| Task | SO | Heuristic | | | | Theoretical error guarantees | | | TB |
|---|---|---|---|---|---|---|---|---|---|
| | | TMV | TMR | TCR | SOR | IWV | DEV | IWA (ours) | |
| 0 → 11 | 0.568(±0.027) | 0.565(±0.118) | 0.565(±0.118) | 0.556(±0.107) | 0.625(±0.023) | 0.539(±0.155) | 0.564(±0.008) | **0.602(±0.027)** | 0.6(±0.023) |
| 12 → 5 | 0.682(±0.043) | 0.812(±0.014) | 0.812(±0.014) | 0.805(±0.014) | 0.827(±0.012) | 0.75(±0.024) | 0.728(±0.018) | **0.809(±0.007)** | 0.798(±0.012) |
| 16 → 1 | 0.574(±0.026) | 0.732(±0.016) | 0.732(±0.016) | 0.729(±0.007) | 0.642(±0.042) | 0.633(±0.066) | 0.676(±0.031) | **0.729(±0.004)** | 0.735(±0.082) |
| 7 → 18 | 0.759(±0.039) | 0.749(±0.023) | 0.749(±0.023) | 0.75(±0.02) | 0.715(±0.014) | 0.724(±0.008) | 0.717(±0.092) | **0.742(±0.01)** | 0.771(±0.02) |
| 9 → 14 | 0.779(±0.038) | 0.837(±0.013) | 0.837(±0.013) | 0.839(±0.008) | 0.805(±0.01) | 0.771(±0.026) | 0.763(±0.006) | **0.84(±0.014)** | 0.811(±0.014) |
| Avg. | 0.672(±0.035) | 0.739(±0.037) | 0.739(±0.037) | 0.736(±0.031) | 0.723(±0.02) | 0.683(±0.056) | 0.69(±0.031) | **0.744(±0.012)** | 0.743(±0.03) |

**Deep-Coral**

| Task | SO | Heuristic | | | | Theoretical error guarantees | | | TB |
|---|---|---|---|---|---|---|---|---|---|
| | | TMV | TMR | TCR | SOR | IWV | DEV | IWA (ours) | |
| 0 → 11 | 0.547(±0.032) | 0.586(±0.028) | 0.586(±0.028) | 0.6(±0.029) | 0.652(±0.058) | **0.629(±0.063)** | 0.569(±0.11) | 0.592(±0.028) | 0.637(±0.058) |
| 12 → 5 | 0.694(±0.084) | 0.75(±0.02) | 0.75(±0.02) | 0.745(±0.016) | 0.781(±0.028) | 0.758(±0.036) | **0.758(±0.036)** | 0.755(±0.018) | 0.776(±0.028) |
| 16 → 1 | 0.545(±0.047) | 0.68(±0.015) | 0.68(±0.015) | 0.671(±0.014) | 0.564(±0.034) | 0.605(±0.018) | 0.553(±0.104) | **0.662(±0.017)** | 0.728(±0.036) |
| 7 → 18 | 0.698(±0.022) | 0.73(±0.042) | 0.73(±0.042) | 0.716(±0.02) | 0.673(±0.006) | 0.723(±0.027) | 0.711(±0.034) | **0.733(±0.013)** | 0.741(±0.016) |
| 9 → 14 | 0.733(±0.047) | 0.836(±0.016) | 0.836(±0.016) | 0.826(±0.02) | 0.802(±0.018) | 0.786(±0.014) | 0.784(±0.043) | **0.82(±0.02)** | 0.792(±0.016) |
| Avg. | 0.643(±0.056) | 0.717(±0.024) | 0.717(±0.024) | 0.712(±0.018) | 0.694(±0.029) | 0.7(±0.031) | 0.675(±0.065) | **0.713(±0.019)** | 0.735(±0.031) |

**CDAN**

| Task | SO | Heuristic | | | | Theoretical error guarantees | | | TB |
|---|---|---|---|---|---|---|---|---|---|
| | | TMV | TMR | TCR | SOR | IWV | DEV | IWA (ours) | |
| 0 → 11 | 0.553(±0.024) | 0.499(±0.065) | 0.499(±0.065) | 0.507(±0.079) | 0.643(±0.053) | 0.529(±0.111) | 0.342(±0.024) | **0.565(±0.078)** | 0.553(±0.053) |
| 12 → 5 | 0.658(±0.101) | 0.844(±0.01) | 0.844(±0.01) | 0.84(±0.047) | 0.829(±0.04) | 0.792(±0.06) | 0.671(±0.055) | **0.835(±0.002)** | 0.85(±0.016) |
| 16 → 1 | 0.699(±0.077) | 0.729(±0.023) | 0.729(±0.023) | 0.726(±0.028) | 0.659(±0.05) | 0.626(±0.117) | 0.682(±0.029) | **0.706(±0.017)** | 0.764(±0.084) |
| 7 → 18 | 0.647(±0.045) | 0.758(±0.017) | 0.758(±0.017) | 0.76(±0.016) | 0.704(±0.005) | 0.73(±0.004) | 0.754(±0.041) | **0.773(±0.012)** | 0.794(±0.035) |
| 9 → 14 | 0.703(±0.041) | 0.865(±0.018) | 0.865(±0.018) | 0.862(±0.028) | 0.806(±0.032) | 0.81(±0.053) | 0.759(±0.041) | **0.862(±0.018)** | 0.832(±0.018) |
| Avg. | 0.652(±0.058) | 0.739(±0.027) | 0.739(±0.027) | 0.739(±0.032) | 0.728(±0.03) | 0.697(±0.069) | 0.642(±0.038) | **0.748(±0.025)** | 0.759(±0.041) |

**DANN**

| Task | SO | Heuristic | | | | Theoretical error guarantees | | | TB |
|---|---|---|---|---|---|---|---|---|---|
| | | TMV | TMR | TCR | SOR | IWV | DEV | IWA (ours) | |
| 0 → 11 | 0.551(±0.018) | 0.602(±0.058) | 0.602(±0.058) | 0.602(±0.052) | 0.672(±0.008) | 0.594(±0.081) | 0.495(±0.114) | **0.622(±0.04)** | 0.574(±0.008) |
| 12 → 5 | 0.645(±0.075) | 0.801(±0.014) | 0.801(±0.014) | 0.792(±0.014) | 0.798(±0.018) | 0.686(±0.051) | 0.686(±0.051) | **0.784(±0.02)** | 0.819(±0.054) |
| 16 → 1 | 0.57(±0.136) | 0.676(±0.002) | 0.676(±0.002) | 0.67(±0.008) | 0.593(±0.025) | 0.651(±0.055) | 0.651(±0.055) | **0.661(±0.008)** | 0.747(±0.049) |
| 7 → 18 | 0.695(±0.037) | 0.725(±0.006) | 0.725(±0.006) | 0.725(±0.014) | 0.686(±0.013) | 0.703(±0.034) | 0.703(±0.034) | **0.732(±0.011)** | 0.738(±0.018) |
| 9 → 14 | 0.746(±0.021) | 0.814(±0.002) | 0.814(±0.002) | 0.815(±0.005) | 0.822(±0.012) | 0.799(±0.039) | 0.686(±0.085) | **0.822(±0.01)** | 0.81(±0.01) |
| Avg. | 0.641(±0.057) | 0.723(±0.016) | 0.723(±0.016) | 0.721(±0.02) | 0.714(±0.015) | 0.687(±0.052) | 0.644(±0.068) | **0.724(±0.018)** | 0.738(±0.028) |

**DSAN**

| Task | SO | Heuristic | | | | Theoretical error guarantees | | | TB |
|---|---|---|---|---|---|---|---|---|---|
| | | TMV | TMR | TCR | SOR | IWV | DEV | IWA (ours) | |
| 0 → 11 | 0.547(±0.012) | 0.539(±0.035) | 0.539(±0.035) | 0.527(±0.063) | 0.668(±0.024) | 0.59(±0.055) | 0.311(±0.026) | **0.629(±0.051)** | 0.603(±0.024) |
| 12 → 5 | 0.717(±0.086) | 0.84(±0.02) | 0.84(±0.02) | 0.841(±0.022) | 0.837(±0.026) | 0.822(±0.028) | 0.699(±0.061) | **0.842(±0.018)** | 0.855(±0.007) |
| 16 → 1 | 0.697(±0.06) | 0.755(±0.031) | 0.755(±0.031) | 0.722(±0.02) | 0.612(±0.026) | 0.617(±0.118) | 0.617(±0.118) | **0.721(±0.014)** | 0.742(±0.031) |
| 7 → 18 | 0.684(±0.048) | 0.734(±0.029) | 0.734(±0.029) | 0.741(±0.011) | 0.71(±0.067) | 0.749(±0.02) | 0.745(±0.025) | **0.764(±0.002)** | 0.768(±0.059) |
| 9 → 14 | 0.62(±0.04) | 0.832(±0.024) | 0.832(±0.024) | 0.827(±0.018) | 0.815(±0.018) | 0.784(±0.046) | 0.576(±0.154) | **0.828(±0.021)** | 0.823(±0.024) |
| Avg. | 0.653(±0.049) | 0.74(±0.028) | 0.74(±0.028) | 0.732(±0.027) | 0.728(±0.032) | 0.712(±0.053) | 0.589(±0.077) | **0.757(±0.021)** | 0.758(±0.029) |

Table 23: Mean and standard deviation (after $\pm$) of target classification accuracy on UCI-HAR (Part 1) over 3 repetitions with different random initialization of model weights.

| HoMM | | | | | | | | | |
|---|---|---|---|---|---|---|---|---|---|
| | | Heuristic | | | | Theoretical error guarantees | | | |
| Task | SO | TMV | TMR | TCR | SOR | IWV | DEV | IWA (ours) | TB |
| 12 → 16 | 0.674(±0.012) | 0.667(±0.036) | 0.667(±0.036) | 0.667(±0.036) | 0.688(±0.0) | 0.674(±0.012) | 0.674(±0.012) | **0.688(±0.0)** | 0.715(±0.012) |
| 2 → 11 | 1.0(±0.0) | 1.0(±0.0) | 1.0(±0.0) | 1.0(±0.0) | 0.938(±0.031) | 0.99(±0.018) | 1.0(±0.0) | **1.0(±0.0)** | 1.0(±0.0) |
| 6 → 23 | 0.854(±0.036) | 0.896(±0.0) | 0.896(±0.0) | 0.896(±0.0) | 0.889(±0.012) | 0.854(±0.036) | 0.854(±0.036) | **0.903(±0.012)** | 0.938(±0.0) |
| 7 → 13 | 0.889(±0.052) | 0.91(±0.024) | 0.91(±0.024) | 0.91(±0.024) | 0.889(±0.052) | 0.889(±0.052) | 0.889(±0.052) | **0.91(±0.024)** | 0.944(±0.024) |
| 9 → 18 | 0.493(±0.032) | 0.618(±0.105) | 0.618(±0.105) | 0.618(±0.105) | 0.514(±0.188) | **0.639(±0.043)** | 0.583(±0.116) | 0.632(±0.079) | 0.715(±0.024) |
| Avg. | 0.782(±0.026) | 0.818(±0.033) | 0.818(±0.033) | 0.818(±0.033) | 0.783(±0.057) | 0.809(±0.032) | 0.8(±0.043) | **0.826(±0.023)** | 0.862(±0.012) |

| AdvSKM | | | | | | | | | |
|---|---|---|---|---|---|---|---|---|---|
| | | Heuristic | | | | Theoretical error guarantees | | | |
| Task | SO | TMV | TMR | TCR | SOR | IWV | DEV | IWA (ours) | TB |
| 12 → 16 | 0.59(±0.103) | 0.681(±0.012) | 0.681(±0.012) | 0.681(±0.012) | 0.708(±0.021) | 0.59(±0.103) | 0.59(±0.103) | **0.688(±0.0)** | 0.722(±0.012) |
| 2 → 11 | 0.979(±0.018) | 1.0(±0.0) | 1.0(±0.0) | 1.0(±0.0) | 0.854(±0.065) | 0.896(±0.1) | 0.917(±0.118) | **1.0(±0.0)** | 1.0(±0.0) |
| 6 → 23 | 0.826(±0.084) | 0.896(±0.0) | 0.896(±0.0) | 0.896(±0.0) | 0.889(±0.012) | 0.826(±0.084) | 0.826(±0.084) | **0.896(±0.0)** | 0.896(±0.0) |
| 7 → 13 | 0.833(±0.036) | 0.91(±0.024) | 0.91(±0.024) | 0.917(±0.036) | 0.896(±0.036) | 0.833(±0.036) | 0.833(±0.036) | **0.903(±0.024)** | 0.924(±0.012) |
| 9 → 18 | 0.389(±0.151) | 0.514(±0.079) | 0.514(±0.079) | 0.556(±0.064) | 0.493(±0.064) | 0.389(±0.151) | 0.389(±0.151) | **0.514(±0.024)** | 0.597(±0.052) |
| Avg. | 0.724(±0.078) | 0.8(±0.023) | 0.8(±0.023) | 0.81(±0.022) | 0.768(±0.04) | 0.707(±0.095) | 0.711(±0.098) | **0.8(±0.01)** | 0.828(±0.015) |

| DIRT | | | | | | | | | |
|---|---|---|---|---|---|---|---|---|---|
| | | Heuristic | | | | Theoretical error guarantees | | | |
| Task | SO | TMV | TMR | TCR | SOR | IWV | DEV | IWA (ours) | TB |
| 12 → 16 | 0.667(±0.075) | 0.778(±0.012) | 0.778(±0.012) | 0.75(±0.0) | 0.736(±0.012) | 0.667(±0.075) | 0.667(±0.075) | **0.764(±0.032)** | 0.819(±0.064) |
| 2 → 11 | 1.0(±0.0) | 1.0(±0.0) | 1.0(±0.0) | 1.0(±0.0) | 0.854(±0.079) | 1.0(±0.0) | 1.0(±0.0) | **1.0(±0.0)** | 1.0(±0.0) |
| 6 → 23 | 0.903(±0.084) | 0.938(±0.0) | 0.938(±0.0) | 0.938(±0.0) | 0.847(±0.06) | 0.847(±0.06) | 0.847(±0.06) | **0.938(±0.0)** | 0.938(±0.0) |
| 7 → 13 | 0.833(±0.0) | 0.958(±0.0) | 0.958(±0.0) | 0.958(±0.0) | 0.924(±0.06) | 0.854(±0.036) | 0.861(±0.032) | **0.958(±0.0)** | 0.958(±0.0) |
| 9 → 18 | 0.514(±0.024) | 0.861(±0.122) | 0.861(±0.122) | 0.806(±0.032) | 0.417(±0.075) | 0.667(±0.225) | 0.667(±0.225) | **0.84(±0.048)** | 0.986(±0.012) |
| Avg. | 0.783(±0.037) | 0.907(±0.027) | 0.907(±0.027) | 0.89(±0.006) | 0.756(±0.057) | 0.807(±0.079) | 0.808(±0.078) | **0.9(±0.016)** | 0.94(±0.015) |

| DDC | | | | | | | | | |
|---|---|---|---|---|---|---|---|---|---|
| | | Heuristic | | | | Theoretical error guarantees | | | |
| Task | SO | TMV | TMR | TCR | SOR | IWV | DEV | IWA (ours) | TB |
| 12 → 16 | 0.681(±0.024) | 0.681(±0.012) | 0.681(±0.012) | 0.674(±0.012) | 0.674(±0.012) | 0.681(±0.024) | 0.604(±0.108) | **0.681(±0.012)** | 0.715(±0.073) |
| 2 → 11 | 1.0(±0.0) | 1.0(±0.0) | 1.0(±0.0) | 1.0(±0.0) | 0.74(±0.213) | 0.74(±0.213) | 0.802(±0.266) | **1.0(±0.0)** | 1.0(±0.0) |
| 6 → 23 | 0.889(±0.012) | 0.896(±0.0) | 0.896(±0.0) | 0.896(±0.0) | 0.882(±0.012) | 0.889(±0.012) | 0.896(±0.021) | **0.896(±0.0)** | 0.889(±0.0) |
| 7 → 13 | 0.868(±0.012) | 0.896(±0.036) | 0.896(±0.036) | 0.91(±0.032) | 0.903(±0.024) | 0.868(±0.012) | 0.868(±0.012) | **0.903(±0.024)** | 0.917(±0.021) |
| 9 → 18 | 0.514(±0.043) | 0.562(±0.091) | 0.562(±0.091) | 0.569(±0.115) | 0.583(±0.165) | 0.444(±0.103) | 0.5(±0.127) | **0.542(±0.091)** | 0.59(±0.189) |
| Avg. | 0.79(±0.018) | 0.807(±0.028) | 0.807(±0.028) | 0.81(±0.032) | 0.756(±0.085) | 0.724(±0.073) | 0.734(±0.107) | **0.804(±0.025)** | 0.822(±0.057) |

| CMD | | | | | | | | | |
|---|---|---|---|---|---|---|---|---|---|
| | | Heuristic | | | | Theoretical error guarantees | | | |
| Task | SO | TMV | TMR | TCR | SOR | IWV | DEV | IWA (ours) | TB |
| 12 → 16 | 0.694(±0.012) | 0.729(±0.0) | 0.729(±0.0) | 0.722(±0.012) | 0.688(±0.0) | 0.694(±0.012) | 0.694(±0.012) | **0.729(±0.0)** | 0.75(±0.036) |
| 2 → 11 | 1.0(±0.0) | 1.0(±0.0) | 1.0(±0.0) | 1.0(±0.0) | 0.771(±0.036) | 0.812(±0.272) | 0.812(±0.272) | **1.0(±0.0)** | 1.0(±0.0) |
| 6 → 23 | 0.896(±0.042) | 0.938(±0.0) | 0.938(±0.0) | 0.931(±0.012) | 0.889(±0.012) | 0.903(±0.032) | 0.896(±0.042) | **0.938(±0.0)** | 0.938(±0.0) |
| 7 → 13 | 0.826(±0.073) | 0.938(±0.0) | 0.938(±0.0) | 0.938(±0.0) | 0.91(±0.024) | 0.826(±0.073) | 0.826(±0.073) | **0.938(±0.0)** | 0.944(±0.012) |
| 9 → 18 | 0.521(±0.095) | 0.639(±0.043) | 0.639(±0.043) | 0.604(±0.055) | 0.396(±0.11) | 0.785(±0.146) | **0.833(±0.163)** | 0.604(±0.075) | 0.882(±0.079) |
| Avg. | 0.788(±0.044) | 0.849(±0.009) | 0.849(±0.009) | 0.839(±0.016) | 0.731(±0.036) | 0.804(±0.107) | 0.812(±0.112) | **0.842(±0.015)** | 0.903(±0.025) |

| MMDA | | | | | | | | | |
|---|---|---|---|---|---|---|---|---|---|
| | | Heuristic | | | | Theoretical error guarantees | | | |
| Task | SO | TMV | TMR | TCR | SOR | IWV | DEV | IWA (ours) | TB |
| 12 → 16 | 0.674(±0.012) | 0.694(±0.012) | 0.694(±0.012) | 0.688(±0.0) | 0.681(±0.012) | 0.674(±0.012) | 0.674(±0.012) | **0.688(±0.0)** | 0.757(±0.043) |
| 2 → 11 | 1.0(±0.0) | 1.0(±0.0) | 1.0(±0.0) | 1.0(±0.0) | 0.844(±0.125) | 0.927(±0.065) | 0.927(±0.065) | **1.0(±0.0)** | 1.0(±0.0) |
| 6 → 23 | 0.847(±0.032) | 0.896(±0.0) | 0.896(±0.0) | 0.896(±0.0) | 0.847(±0.067) | 0.847(±0.032) | 0.847(±0.032) | **0.896(±0.0)** | 0.91(±0.024) |
| 7 → 13 | 0.882(±0.043) | 0.903(±0.012) | 0.903(±0.012) | 0.917(±0.021) | 0.924(±0.024) | 0.882(±0.043) | 0.882(±0.043) | **0.931(±0.024)** | 0.944(±0.012) |
| 9 → 18 | 0.521(±0.072) | 0.569(±0.139) | 0.569(±0.139) | 0.5(±0.104) | 0.5(±0.095) | **0.535(±0.064)** | 0.507(±0.024) | 0.521(±0.091) | 0.688(±0.075) |
| Avg. | 0.785(±0.032) | 0.812(±0.033) | 0.812(±0.033) | 0.8(±0.025) | 0.759(±0.065) | 0.773(±0.043) | 0.767(±0.035) | **0.807(±0.023)** | 0.86(±0.031) |

Table 24: Mean and standard deviation (after $\pm$) of target classification accuracy on UCI-HAR (Part 2) over 3 repetitions with different random initialization of model weights.

**CoDATS**

| Task | SO | Heuristic | | | | Theoretical error guarantees | | | TB |
|---|---|---|---|---|---|---|---|---|---|
| | | TMV | TMR | TCR | SOR | IWV | DEV | IWA (ours) | |
| $12 \rightarrow 16$ | $0.681(\pm 0.024)$ | $0.701(\pm 0.024)$ | $0.701(\pm 0.024)$ | $0.688(\pm 0.0)$ | $0.694(\pm 0.024)$ | $0.681(\pm 0.024)$ | $0.694(\pm 0.024)$ | $\mathbf{0.694}(\pm \mathbf{0.012})$ | $0.708(\pm 0.021)$ |
| $2 \rightarrow 11$ | $1.0(\pm 0.0)$ | $1.0(\pm 0.0)$ | $1.0(\pm 0.0)$ | $1.0(\pm 0.0)$ | $0.865(\pm 0.11)$ | $0.865(\pm 0.1)$ | $0.865(\pm 0.1)$ | $\mathbf{1.0}(\pm \mathbf{0.0})$ | $1.0(\pm 0.0)$ |
| $6 \rightarrow 23$ | $0.84(\pm 0.032)$ | $0.889(\pm 0.032)$ | $0.889(\pm 0.032)$ | $0.889(\pm 0.032)$ | $0.882(\pm 0.064)$ | $0.84(\pm 0.048)$ | $0.896(\pm 0.083)$ | $\mathbf{0.924}(\pm \mathbf{0.012})$ | $0.951(\pm 0.043)$ |
| $7 \rightarrow 13$ | $0.75(\pm 0.217)$ | $0.944(\pm 0.012)$ | $0.944(\pm 0.012)$ | $0.944(\pm 0.012)$ | $0.917(\pm 0.036)$ | $0.854(\pm 0.036)$ | $0.75(\pm 0.217)$ | $\mathbf{0.938}(\pm \mathbf{0.0})$ | $0.951(\pm 0.012)$ |
| $9 \rightarrow 18$ | $0.528(\pm 0.032)$ | $0.625(\pm 0.062)$ | $0.625(\pm 0.062)$ | $0.639(\pm 0.043)$ | $0.569(\pm 0.305)$ | $0.764(\pm 0.12)$ | $\mathbf{0.764}(\pm \mathbf{0.12})$ | $0.674(\pm 0.115)$ | $0.826(\pm 0.012)$ |
| Avg. | $0.76(\pm 0.061)$ | $0.832(\pm 0.026)$ | $0.832(\pm 0.026)$ | $0.832(\pm 0.017)$ | $0.785(\pm 0.108)$ | $0.801(\pm 0.066)$ | $0.794(\pm 0.109)$ | $\mathbf{0.846}(\pm \mathbf{0.028})$ | $0.888(\pm 0.018)$ |

**Deep-Coral**

| Task | SO | Heuristic | | | | Theoretical error guarantees | | | TB |
|---|---|---|---|---|---|---|---|---|---|
| | | TMV | TMR | TCR | SOR | IWV | DEV | IWA (ours) | |
| $12 \rightarrow 16$ | $0.681(\pm 0.012)$ | $0.681(\pm 0.012)$ | $0.681(\pm 0.012)$ | $0.674(\pm 0.012)$ | $0.681(\pm 0.012)$ | $0.681(\pm 0.012)$ | $0.639(\pm 0.084)$ | $\mathbf{0.688}(\pm \mathbf{0.0})$ | $0.688(\pm 0.0)$ |
| $2 \rightarrow 11$ | $0.969(\pm 0.054)$ | $1.0(\pm 0.0)$ | $1.0(\pm 0.0)$ | $1.0(\pm 0.0)$ | $0.875(\pm 0.094)$ | $0.875(\pm 0.083)$ | $0.885(\pm 0.095)$ | $\mathbf{1.0}(\pm \mathbf{0.0})$ | $1.0(\pm 0.0)$ |
| $6 \rightarrow 23$ | $0.826(\pm 0.084)$ | $0.882(\pm 0.012)$ | $0.882(\pm 0.012)$ | $0.889(\pm 0.012)$ | $0.896(\pm 0.0)$ | $0.826(\pm 0.084)$ | $0.833(\pm 0.091)$ | $\mathbf{0.896}(\pm \mathbf{0.0})$ | $0.903(\pm 0.043)$ |
| $7 \rightarrow 13$ | $0.917(\pm 0.036)$ | $0.91(\pm 0.032)$ | $0.91(\pm 0.032)$ | $0.91(\pm 0.032)$ | $0.896(\pm 0.021)$ | $0.917(\pm 0.036)$ | $\mathbf{0.917}(\pm \mathbf{0.036})$ | $0.903(\pm 0.024)$ | $0.931(\pm 0.012)$ |
| $9 \rightarrow 18$ | $0.556(\pm 0.087)$ | $0.528(\pm 0.043)$ | $0.528(\pm 0.043)$ | $0.569(\pm 0.115)$ | $0.507(\pm 0.272)$ | $0.542(\pm 0.144)$ | $\mathbf{0.59}(\pm \mathbf{0.032})$ | $0.556(\pm 0.064)$ | $0.59(\pm 0.221)$ |
| Avg. | $0.79(\pm 0.055)$ | $0.8(\pm 0.02)$ | $0.8(\pm 0.02)$ | $0.808(\pm 0.034)$ | $0.771(\pm 0.08)$ | $0.768(\pm 0.072)$ | $0.773(\pm 0.068)$ | $\mathbf{0.808}(\pm \mathbf{0.018})$ | $0.822(\pm 0.055)$ |

**CDAN**

| Task | SO | Heuristic | | | | Theoretical error guarantees | | | TB |
|---|---|---|---|---|---|---|---|---|---|
| | | TMV | TMR | TCR | SOR | IWV | DEV | IWA (ours) | |
| $12 \rightarrow 16$ | $0.674(\pm 0.043)$ | $0.722(\pm 0.012)$ | $0.722(\pm 0.012)$ | $0.722(\pm 0.012)$ | $0.694(\pm 0.012)$ | $0.674(\pm 0.043)$ | $0.646(\pm 0.055)$ | $\mathbf{0.715}(\pm \mathbf{0.024})$ | $0.729(\pm 0.036)$ |
| $2 \rightarrow 11$ | $0.99(\pm 0.018)$ | $1.0(\pm 0.0)$ | $1.0(\pm 0.0)$ | $1.0(\pm 0.0)$ | $0.865(\pm 0.065)$ | $0.792(\pm 0.28)$ | $0.802(\pm 0.29)$ | $\mathbf{1.0}(\pm \mathbf{0.0})$ | $1.0(\pm 0.0)$ |
| $6 \rightarrow 23$ | $0.854(\pm 0.042)$ | $0.924(\pm 0.024)$ | $0.924(\pm 0.024)$ | $0.91(\pm 0.024)$ | $0.91(\pm 0.024)$ | $0.847(\pm 0.032)$ | $0.708(\pm 0.295)$ | $\mathbf{0.924}(\pm \mathbf{0.024})$ | $0.924(\pm 0.024)$ |
| $7 \rightarrow 13$ | $0.826(\pm 0.139)$ | $0.951(\pm 0.012)$ | $0.951(\pm 0.012)$ | $0.951(\pm 0.012)$ | $0.944(\pm 0.012)$ | $0.896(\pm 0.021)$ | $0.812(\pm 0.108)$ | $\mathbf{0.951}(\pm \mathbf{0.012})$ | $0.958(\pm 0.0)$ |
| $9 \rightarrow 18$ | $0.438(\pm 0.055)$ | $0.618(\pm 0.064)$ | $0.618(\pm 0.064)$ | $0.618(\pm 0.064)$ | $0.597(\pm 0.098)$ | $\mathbf{0.694}(\pm \mathbf{0.126})$ | $0.465(\pm 0.087)$ | $0.639(\pm 0.048)$ | $0.729(\pm 0.083)$ |
| Avg. | $0.756(\pm 0.059)$ | $0.843(\pm 0.022)$ | $0.843(\pm 0.022)$ | $0.84(\pm 0.022)$ | $0.802(\pm 0.042)$ | $0.781(\pm 0.1)$ | $0.687(\pm 0.167)$ | $\mathbf{0.846}(\pm \mathbf{0.022})$ | $0.868(\pm 0.029)$ |

**DANN**

| Task | SO | Heuristic | | | | Theoretical error guarantees | | | TB |
|---|---|---|---|---|---|---|---|---|---|
| | | TMV | TMR | TCR | SOR | IWV | DEV | IWA (ours) | |
| $12 \rightarrow 16$ | $0.688(\pm 0.021)$ | $0.701(\pm 0.024)$ | $0.701(\pm 0.024)$ | $0.701(\pm 0.024)$ | $0.715(\pm 0.012)$ | $0.701(\pm 0.012)$ | $0.701(\pm 0.012)$ | $\mathbf{0.729}(\pm \mathbf{0.0})$ | $0.722(\pm 0.0)$ |
| $2 \rightarrow 11$ | $1.0(\pm 0.0)$ | $1.0(\pm 0.0)$ | $1.0(\pm 0.0)$ | $1.0(\pm 0.0)$ | $1.0(\pm 0.0)$ | $0.76(\pm 0.28)$ | $0.729(\pm 0.141)$ | $\mathbf{1.0}(\pm \mathbf{0.0})$ | $1.0(\pm 0.0)$ |
| $6 \rightarrow 23$ | $0.882(\pm 0.024)$ | $0.917(\pm 0.021)$ | $0.917(\pm 0.021)$ | $0.931(\pm 0.012)$ | $0.889(\pm 0.032)$ | $0.868(\pm 0.024)$ | $0.868(\pm 0.024)$ | $\mathbf{0.931}(\pm \mathbf{0.012})$ | $0.958(\pm 0.036)$ |
| $7 \rightarrow 13$ | $0.806(\pm 0.127)$ | $0.944(\pm 0.012)$ | $0.944(\pm 0.012)$ | $0.944(\pm 0.012)$ | $0.938(\pm 0.021)$ | $0.903(\pm 0.024)$ | $0.944(\pm 0.048)$ | $\mathbf{0.944}(\pm \mathbf{0.012})$ | $0.965(\pm 0.012)$ |
| $9 \rightarrow 18$ | $0.403(\pm 0.12)$ | $0.715(\pm 0.012)$ | $0.715(\pm 0.012)$ | $0.701(\pm 0.067)$ | $0.729(\pm 0.127)$ | $0.583(\pm 0.11)$ | $0.583(\pm 0.11)$ | $\mathbf{0.639}(\pm \mathbf{0.103})$ | $0.674(\pm 0.127)$ |
| Avg. | $0.756(\pm 0.058)$ | $0.856(\pm 0.014)$ | $0.856(\pm 0.014)$ | $0.856(\pm 0.023)$ | $0.8(\pm 0.066)$ | $0.763(\pm 0.064)$ | $0.78(\pm 0.08)$ | $\mathbf{0.849}(\pm \mathbf{0.025})$ | $0.864(\pm 0.035)$ |

**DSAN**

| Task | SO | Heuristic | | | | Theoretical error guarantees | | | TB |
|---|---|---|---|---|---|---|---|---|---|
| | | TMV | TMR | TCR | SOR | IWV | DEV | IWA (ours) | |
| $12 \rightarrow 16$ | $0.681(\pm 0.012)$ | $0.688(\pm 0.021)$ | $0.688(\pm 0.021)$ | $0.715(\pm 0.024)$ | $0.694(\pm 0.012)$ | $0.681(\pm 0.012)$ | $0.556(\pm 0.229)$ | $\mathbf{0.715}(\pm \mathbf{0.024})$ | $0.736(\pm 0.012)$ |
| $2 \rightarrow 11$ | $1.0(\pm 0.0)$ | $1.0(\pm 0.0)$ | $1.0(\pm 0.0)$ | $1.0(\pm 0.0)$ | $0.969(\pm 0.0)$ | $1.0(\pm 0.0)$ | $1.0(\pm 0.0)$ | $\mathbf{1.0}(\pm \mathbf{0.0})$ | $1.0(\pm 0.0)$ |
| $6 \rightarrow 23$ | $0.764(\pm 0.094)$ | $0.938(\pm 0.0)$ | $0.938(\pm 0.0)$ | $0.917(\pm 0.021)$ | $0.847(\pm 0.052)$ | $0.785(\pm 0.103)$ | $0.785(\pm 0.103)$ | $\mathbf{0.931}(\pm \mathbf{0.012})$ | $0.938(\pm 0.0)$ |
| $7 \rightarrow 13$ | $0.854(\pm 0.036)$ | $0.951(\pm 0.012)$ | $0.951(\pm 0.012)$ | $0.951(\pm 0.012)$ | $0.854(\pm 0.036)$ | $0.854(\pm 0.036)$ | $0.854(\pm 0.036)$ | $\mathbf{0.951}(\pm \mathbf{0.012})$ | $0.958(\pm 0.0)$ |
| $9 \rightarrow 18$ | $0.514(\pm 0.032)$ | $0.639(\pm 0.079)$ | $0.639(\pm 0.079)$ | $0.688(\pm 0.091)$ | $0.382(\pm 0.012)$ | $0.556(\pm 0.067)$ | $0.528(\pm 0.067)$ | $\mathbf{0.694}(\pm \mathbf{0.032})$ | $0.743(\pm 0.087)$ |
| Avg. | $0.762(\pm 0.035)$ | $0.843(\pm 0.022)$ | $0.843(\pm 0.022)$ | $0.854(\pm 0.03)$ | $0.749(\pm 0.023)$ | $0.775(\pm 0.044)$ | $0.744(\pm 0.087)$ | $\mathbf{0.858}(\pm \mathbf{0.016})$ | $0.875(\pm 0.02)$ |

Table 25: Mean and standard deviation (after $\pm$) of target classification accuracy on HHAR (Part 1) over 3 repetitions with different random initialization of model weights.

**HoMM**

| Task | SO | Heuristic | | | | Theoretical error guarantees | | | TB |
|---|---|---|---|---|---|---|---|---|---|
| | | TMV | TMR | TCR | SOR | IWV | DEV | IWA (ours) | |
| 0→6 | 0.719(±0.041) | 0.737(±0.019) | 0.737(±0.019) | 0.735(±0.024) | 0.635(±0.122) | 0.704(±0.011) | 0.703(±0.013) | **0.732(±0.01)** | 0.733(±0.019) |
| 1→6 | 0.788(±0.11) | 0.871(±0.011) | 0.871(±0.011) | 0.869(±0.013) | 0.788(±0.163) | 0.749(±0.06) | 0.785(±0.106) | **0.879(±0.011)** | 0.914(±0.01) |
| 2→7 | 0.528(±0.103) | 0.457(±0.007) | 0.457(±0.007) | 0.461(±0.007) | 0.469(±0.051) | 0.497(±0.105) | **0.516(±0.087)** | 0.455(±0.015) | 0.546(±0.071) |
| 3→8 | 0.797(±0.007) | 0.818(±0.002) | 0.818(±0.002) | 0.816(±0.007) | 0.812(±0.014) | 0.805(±0.01) | 0.805(±0.01) | **0.818(±0.005)** | 0.831(±0.022) |
| 4→5 | 0.861(±0.022) | 0.914(±0.007) | 0.914(±0.007) | 0.911(±0.005) | 0.798(±0.164) | 0.844(±0.016) | 0.855(±0.01) | **0.911(±0.005)** | 0.94(±0.04) |
| Avg. | 0.739(±0.057) | 0.759(±0.009) | 0.759(±0.009) | 0.759(±0.011) | 0.7(±0.103) | 0.72(±0.041) | 0.733(±0.045) | **0.759(±0.009)** | 0.793(±0.032) |

**AdvSKM**

| Task | SO | Heuristic | | | | Theoretical error guarantees | | | TB |
|---|---|---|---|---|---|---|---|---|---|
| | | TMV | TMR | TCR | SOR | IWV | DEV | IWA (ours) | |
| 0→6 | 0.661(±0.035) | 0.722(±0.009) | 0.722(±0.009) | 0.725(±0.007) | 0.629(±0.076) | 0.699(±0.013) | 0.699(±0.013) | **0.718(±0.009)** | 0.731(±0.013) |
| 1→6 | 0.821(±0.046) | 0.849(±0.002) | 0.849(±0.002) | 0.851(±0.006) | 0.604(±0.045) | 0.806(±0.026) | 0.806(±0.026) | **0.857(±0.006)** | 0.838(±0.018) |
| 2→7 | 0.455(±0.07) | 0.473(±0.031) | 0.473(±0.031) | 0.484(±0.03) | 0.497(±0.067) | 0.497(±0.067) | 0.497(±0.067) | 0.49(±0.014) | 0.574(±0.112) |
| 3→8 | 0.79(±0.011) | 0.799(±0.008) | 0.799(±0.008) | 0.803(±0.005) | 0.805(±0.01) | 0.799(±0.005) | 0.788(±0.023) | **0.81(±0.002)** | 0.818(±0.012) |
| 4→5 | 0.862(±0.055) | 0.866(±0.013) | 0.866(±0.013) | 0.875(±0.008) | 0.809(±0.094) | 0.85(±0.025) | 0.85(±0.025) | **0.884(±0.005)** | 0.895(±0.014) |
| Avg. | 0.718(±0.044) | 0.742(±0.013) | 0.742(±0.013) | 0.748(±0.011) | 0.676(±0.058) | 0.73(±0.027) | 0.728(±0.031) | **0.752(±0.007)** | 0.771(±0.034) |

**DIRT**

| Task | SO | Heuristic | | | | Theoretical error guarantees | | | TB |
|---|---|---|---|---|---|---|---|---|---|
| | | TMV | TMR | TCR | SOR | IWV | DEV | IWA (ours) | |
| 0→6 | 0.708(±0.011) | 0.571(±0.079) | 0.571(±0.079) | 0.633(±0.137) | 0.629(±0.085) | 0.708(±0.011) | 0.708(±0.011) | **0.779(±0.011)** | 0.739(±0.011) |
| 1→6 | 0.756(±0.058) | 0.938(±0.004) | 0.938(±0.004) | 0.938(±0.004) | 0.904(±0.058) | 0.814(±0.056) | 0.756(±0.058) | **0.938(±0.0)** | 0.942(±0.004) |
| 2→7 | 0.531(±0.057) | 0.622(±0.091) | 0.622(±0.091) | 0.621(±0.049) | 0.658(±0.03) | 0.548(±0.049) | 0.548(±0.049) | 0.53(±0.025) | 0.688(±0.004) |
| 3→8 | 0.807(±0.008) | 0.846(±0.022) | 0.846(±0.022) | 0.837(±0.015) | 0.857(±0.087) | 0.807(±0.008) | 0.807(±0.008) | **0.848(±0.007)** | 0.911(±0.07) |
| 4→5 | 0.839(±0.016) | 0.984(±0.004) | 0.984(±0.004) | 0.984(±0.004) | 0.93(±0.061) | 0.839(±0.016) | 0.878(±0.074) | **0.984(±0.004)** | 0.984(±0.002) |
| Avg. | 0.728(±0.03) | 0.792(±0.04) | 0.792(±0.04) | 0.803(±0.05) | 0.796(±0.064) | 0.743(±0.028) | 0.739(±0.04) | **0.816(±0.009)** | 0.853(±0.018) |

**DDC**

| Task | SO | Heuristic | | | | Theoretical error guarantees | | | TB |
|---|---|---|---|---|---|---|---|---|---|
| | | TMV | TMR | TCR | SOR | IWV | DEV | IWA (ours) | |
| 0→6 | 0.575(±0.083) | 0.693(±0.017) | 0.693(±0.017) | 0.672(±0.009) | 0.632(±0.038) | 0.646(±0.015) | 0.6(±0.084) | **0.697(±0.013)** | 0.651(±0.013) |
| 1→6 | 0.875(±0.014) | 0.888(±0.007) | 0.888(±0.007) | 0.894(±0.013) | 0.886(±0.019) | 0.856(±0.067) | 0.856(±0.067) | **0.882(±0.017)** | 0.899(±0.016) |
| 2→7 | 0.487(±0.016) | 0.457(±0.03) | 0.457(±0.03) | 0.455(±0.029) | 0.497(±0.028) | 0.438(±0.054) | 0.438(±0.054) | **0.439(±0.041)** | 0.533(±0.107) |
| 3→8 | 0.815(±0.019) | 0.805(±0.014) | 0.805(±0.014) | 0.807(±0.016) | 0.797(±0.022) | 0.827(±0.025) | **0.831(±0.02)** | 0.818(±0.008) | 0.822(±0.02) |
| 4→5 | 0.831(±0.048) | 0.908(±0.011) | 0.908(±0.011) | 0.911(±0.01) | 0.772(±0.085) | 0.792(±0.02) | 0.801(±0.012) | **0.905(±0.008)** | 0.888(±0.01) |
| Avg. | 0.716(±0.036) | 0.75(±0.016) | 0.75(±0.016) | 0.748(±0.015) | 0.717(±0.038) | 0.711(±0.036) | 0.705(±0.047) | **0.748(±0.017)** | 0.758(±0.033) |

**CMD**

| Task | SO | Heuristic | | | | Theoretical error guarantees | | | TB |
|---|---|---|---|---|---|---|---|---|---|
| | | TMV | TMR | TCR | SOR | IWV | DEV | IWA (ours) | |
| 0→6 | 0.703(±0.04) | 0.643(±0.046) | 0.643(±0.046) | 0.679(±0.054) | 0.672(±0.103) | 0.743(±0.021) | **0.743(±0.021)** | 0.693(±0.012) | 0.724(±0.021) |
| 1→6 | 0.861(±0.035) | 0.915(±0.009) | 0.915(±0.009) | 0.912(±0.0) | 0.826(±0.103) | 0.899(±0.03) | 0.843(±0.156) | **0.907(±0.006)** | 0.925(±0.004) |
| 2→7 | 0.573(±0.018) | 0.494(±0.011) | 0.494(±0.011) | 0.499(±0.023) | 0.51(±0.066) | **0.577(±0.011)** | 0.557(±0.039) | 0.482(±0.008) | 0.603(±0.016) |
| 3→8 | 0.799(±0.016) | 0.816(±0.0) | 0.816(±0.0) | 0.816(±0.0) | 0.797(±0.004) | 0.799(±0.016) | 0.738(±0.109) | **0.811(±0.011)** | 0.822(±0.01) |
| 4→5 | 0.806(±0.024) | 0.952(±0.016) | 0.952(±0.016) | 0.93(±0.007) | 0.879(±0.054) | 0.854(±0.063) | 0.332(±0.051) | **0.939(±0.002)** | 0.961(±0.024) |
| Avg. | 0.748(±0.026) | 0.764(±0.016) | 0.764(±0.016) | 0.767(±0.017) | 0.737(±0.066) | **0.775(±0.028)** | 0.643(±0.075) | 0.766(±0.008) | 0.807(±0.015) |

**MMDA**

| Task | SO | Heuristic | | | | Theoretical error guarantees | | | TB |
|---|---|---|---|---|---|---|---|---|---|
| | | TMV | TMR | TCR | SOR | IWV | DEV | IWA (ours) | |
| 0→6 | 0.706(±0.042) | 0.737(±0.021) | 0.737(±0.021) | 0.737(±0.018) | 0.704(±0.037) | 0.699(±0.055) | 0.725(±0.018) | **0.746(±0.008)** | 0.732(±0.008) |
| 1→6 | 0.792(±0.098) | 0.91(±0.002) | 0.91(±0.002) | 0.91(±0.002) | 0.742(±0.147) | 0.732(±0.009) | 0.722(±0.009) | **0.897(±0.002)** | 0.914(±0.028) |
| 2→7 | 0.552(±0.073) | 0.481(±0.003) | 0.481(±0.003) | 0.482(±0.004) | 0.515(±0.058) | 0.496(±0.027) | **0.543(±0.055)** | 0.488(±0.01) | 0.552(±0.073) |
| 3→8 | 0.79(±0.016) | 0.862(±0.026) | 0.862(±0.026) | 0.841(±0.01) | 0.794(±0.022) | 0.802(±0.018) | 0.802(±0.018) | **0.839(±0.012)** | 0.932(±0.005) |
| 4→5 | 0.852(±0.036) | 0.917(±0.016) | 0.917(±0.016) | 0.931(±0.002) | 0.733(±0.061) | 0.865(±0.055) | 0.865(±0.055) | **0.928(±0.006)** | 0.947(±0.022) |
| Avg. | 0.738(±0.053) | 0.781(±0.014) | 0.781(±0.014) | 0.78(±0.007) | 0.698(±0.065) | 0.719(±0.033) | 0.731(±0.031) | **0.78(±0.008)** | 0.815(±0.027) |

Table 26: Mean and standard deviation (after ±) of target classification accuracy on HHAR (Part 2) over 3 repetitions with different random initialization of model weights.

| | | CoDATS | | | | | | | |
| | | Heuristic | | | | Theoretical error guarantees | | | |
| Task | SO | TMV | TMR | TCR | SOR | IWV | DEV | IWA (ours) | TB |
|---|---|---|---|---|---|---|---|---|---|
| 0 → 6 | 0.64(±0.096) | 0.497(±0.005) | 0.497(±0.005) | 0.499(±0.006) | *0.582(±0.122)* | 0.689(±0.069) | 0.606(±0.106) | **0.718(±0.017)** | 0.735(±0.01) |
| 1 → 6 | 0.775(±0.091) | *0.946(±0.0)* | 0.946(±0.0) | 0.946(±0.0) | 0.683(±0.076) | 0.896(±0.069) | 0.896(±0.069) | **0.939(±0.006)** | 0.947(±0.009) |
| 2 → 7 | 0.527(±0.076) | 0.473(±0.012) | 0.473(±0.012) | 0.475(±0.01) | *0.534(±0.052)* | 0.472(±0.019) | **0.555(±0.069)** | 0.472(±0.005) | 0.558(±0.041) |
| 3 → 8 | 0.783(±0.015) | *0.971(±0.018)* | 0.971(±0.018) | 0.969(±0.018) | 0.97(±0.025) | 0.789(±0.01) | 0.789(±0.01) | **0.96(±0.03)** | 0.987(±0.002) |
| 4 → 5 | 0.827(±0.039) | 0.971(±0.013) | 0.971(±0.013) | *0.978(±0.002)* | 0.841(±0.1) | 0.849(±0.074) | 0.849(±0.074) | **0.973(±0.004)** | 0.979(±0.006) |
| Avg. | 0.71(±0.063) | 0.772(±0.009) | 0.772(±0.009) | *0.773(±0.007)* | 0.722(±0.075) | 0.739(±0.048) | 0.739(±0.066) | **0.812(±0.012)** | 0.841(±0.014) |

| | | Deep-Coral | | | | | | | |
| | | Heuristic | | | | Theoretical error guarantees | | | |
| Task | SO | TMV | TMR | TCR | SOR | IWV | DEV | IWA (ours) | TB |
|---|---|---|---|---|---|---|---|---|---|
| 0 → 6 | 0.692(±0.026) | 0.731(±0.028) | 0.731(±0.028) | *0.736(±0.024)* | 0.632(±0.024) | 0.703(±0.038) | 0.7(±0.038) | **0.728(±0.019)** | 0.703(±0.024) |
| 1 → 6 | 0.862(±0.018) | 0.893(±0.009) | 0.893(±0.009) | *0.896(±0.004)* | 0.822(±0.104) | 0.864(±0.017) | 0.864(±0.017) | **0.886(±0.009)** | 0.911(±0.017) |
| 2 → 7 | 0.509(±0.097) | 0.454(±0.02) | 0.454(±0.02) | 0.457(±0.025) | 0.51(±0.057) | 0.499(±0.079) | **0.539(±0.076)** | 0.46(±0.012) | 0.565(±0.117) |
| 3 → 8 | 0.798(±0.016) | 0.801(±0.004) | 0.801(±0.004) | *0.803(±0.006)* | 0.793(±0.024) | 0.799(±0.013) | 0.802(±0.016) | **0.812(±0.0)** | 0.822(±0.01) |
| 4 → 5 | 0.862(±0.036) | 0.932(±0.025) | 0.932(±0.025) | *0.938(±0.022)* | 0.648(±0.108) | 0.906(±0.027) | 0.887(±0.035) | **0.936(±0.016)** | 0.96(±0.01) |
| Avg. | 0.745(±0.039) | 0.762(±0.017) | 0.762(±0.017) | *0.766(±0.016)* | 0.681(±0.063) | 0.754(±0.035) | 0.758(±0.036) | **0.764(±0.011)** | 0.792(±0.036) |

| | | CDAN | | | | | | | |
| | | Heuristic | | | | Theoretical error guarantees | | | |
| Task | SO | TMV | TMR | TCR | SOR | IWV | DEV | IWA (ours) | TB |
|---|---|---|---|---|---|---|---|---|---|
| 0 → 6 | 0.717(±0.004) | 0.483(±0.008) | 0.483(±0.008) | 0.481(±0.006) | *0.699(±0.021)* | 0.692(±0.083) | 0.692(±0.083) | **0.76(±0.016)** | 0.718(±0.016) |
| 1 → 6 | 0.742(±0.09) | *0.932(±0.006)* | 0.932(±0.006) | 0.932(±0.01) | 0.906(±0.021) | 0.942(±0.011) | **0.944(±0.013)** | 0.933(±0.0) | 0.946(±0.0) |
| 2 → 7 | 0.554(±0.031) | 0.522(±0.05) | 0.522(±0.05) | 0.56(±0.059) | *0.58(±0.077)* | 0.561(±0.068) | **0.561(±0.068)** | 0.531(±0.066) | 0.624(±0.014) |
| 3 → 8 | 0.77(±0.069) | *0.872(±0.09)* | 0.872(±0.09) | 0.871(±0.091) | 0.812(±0.007) | 0.801(±0.004) | 0.801(±0.004) | **0.874(±0.076)** | 0.987(±0.006) |
| 4 → 5 | 0.859(±0.014) | 0.978(±0.005) | 0.978(±0.005) | *0.979(±0.002)* | 0.828(±0.106) | 0.875(±0.087) | 0.875(±0.087) | **0.98(±0.0)** | 0.982(±0.002) |
| Avg. | 0.728(±0.042) | 0.758(±0.032) | 0.758(±0.032) | 0.764(±0.034) | *0.765(±0.046)* | 0.774(±0.051) | 0.775(±0.051) | **0.816(±0.031)** | 0.851(±0.008) |

| | | DANN | | | | | | | |
| | | Heuristic | | | | Theoretical error guarantees | | | |
| Task | SO | TMV | TMR | TCR | SOR | IWV | DEV | IWA (ours) | TB |
|---|---|---|---|---|---|---|---|---|---|
| 0 → 6 | 0.704(±0.037) | 0.489(±0.006) | 0.489(±0.006) | 0.49(±0.006) | *0.585(±0.092)* | 0.724(±0.016) | **0.724(±0.016)** | 0.722(±0.071) | 0.711(±0.016) |
| 1 → 6 | 0.833(±0.036) | *0.94(±0.002)* | 0.94(±0.002) | 0.938(±0.004) | 0.868(±0.142) | 0.929(±0.004) | 0.931(±0.006) | **0.936(±0.002)** | 0.939(±0.002) |
| 2 → 7 | 0.591(±0.023) | 0.496(±0.004) | 0.496(±0.004) | 0.493(±0.011) | *0.557(±0.061)* | **0.618(±0.029)** | 0.592(±0.022) | 0.49(±0.003) | 0.635(±0.025) |
| 3 → 8 | 0.809(±0.02) | *0.966(±0.015)* | 0.966(±0.015) | 0.966(±0.015) | 0.823(±0.069) | 0.796(±0.03) | 0.796(±0.03) | **0.964(±0.006)** | 0.983(±0.002) |
| 4 → 5 | 0.846(±0.022) | *0.98(±0.0)* | 0.98(±0.0) | 0.98(±0.0) | 0.779(±0.138) | 0.922(±0.078) | 0.922(±0.078) | **0.98(±0.0)** | 0.98(±0.0) |
| Avg. | 0.757(±0.027) | *0.774(±0.006)* | 0.774(±0.006) | 0.773(±0.007) | 0.722(±0.1) | 0.798(±0.031) | 0.793(±0.031) | **0.818(±0.016)** | 0.85(±0.009) |

| | | DSAN | | | | | | | |
| | | Heuristic | | | | Theoretical error guarantees | | | |
| Task | SO | TMV | TMR | TCR | SOR | IWV | DEV | IWA (ours) | TB |
|---|---|---|---|---|---|---|---|---|---|
| 0 → 6 | 0.653(±0.033) | 0.607(±0.057) | 0.607(±0.057) | *0.635(±0.119)* | 0.597(±0.075) | 0.572(±0.103) | 0.471(±0.259) | **0.747(±0.027)** | 0.664(±0.027) |
| 1 → 6 | 0.806(±0.083) | 0.922(±0.002) | 0.922(±0.002) | *0.925(±0.004)* | 0.821(±0.105) | 0.908(±0.065) | 0.908(±0.065) | **0.929(±0.0)** | 0.94(±0.005) |
| 2 → 7 | 0.49(±0.053) | *0.496(±0.004)* | 0.496(±0.004) | 0.494(±0.003) | 0.485(±0.005) | 0.482(±0.012) | 0.49(±0.003) | **0.496(±0.0)** | 0.586(±0.083) |
| 3 → 8 | 0.797(±0.016) | *0.979(±0.002)* | 0.979(±0.002) | 0.977(±0.004) | 0.855(±0.097) | 0.816(±0.01) | 0.639(±0.454) | **0.971(±0.005)** | 0.982(±0.01) |
| 4 → 5 | 0.861(±0.043) | *0.98(±0.008)* | 0.98(±0.008) | 0.979(±0.006) | 0.863(±0.085) | 0.928(±0.08) | 0.471(±0.438) | **0.98(±0.0)** | 0.982(±0.005) |
| Avg. | 0.721(±0.046) | 0.797(±0.015) | 0.797(±0.015) | *0.802(±0.027)* | 0.724(±0.073) | 0.741(±0.054) | 0.596(±0.244) | **0.825(±0.006)** | 0.831(±0.026) |

Table 27: Mean and standard deviation (after ±) of target classification accuracy on WISDM (Part 1) over 3 repetitions with different random initialization of model weights.

**HoMM**

| Task | SO | Heuristic | | | | Theoretical error guarantees | | | TB |
|---|---|---|---|---|---|---|---|---|---|
| | | TMV | TMR | TCR | SOR | IWV | DEV | IWA (ours) | TB |
| 18 → 23 | 0.733(±0.058) | 0.744(±0.038) | 0.744(±0.038) | *0.756(±0.051)* | 0.733(±0.033) | 0.733(±0.058) | **0.733(±0.058)** | 0.711(±0.019) | 0.767(±0.033) |
| 20 → 30 | 0.853(±0.022) | 0.814(±0.022) | 0.814(±0.022) | 0.801(±0.011) | *0.827(±0.0)* | 0.853(±0.022) | **0.853(±0.022)** | 0.814(±0.011) | 0.853(±0.022) |
| 35 → 31 | 0.579(±0.055) | *0.77(±0.014)* | 0.77(±0.014) | 0.77(±0.014) | 0.722(±0.05) | 0.579(±0.055) | 0.579(±0.055) | **0.746(±0.027)** | 0.77(±0.014) |
| 6 → 19 | 0.889(±0.009) | 0.864(±0.052) | 0.864(±0.052) | 0.823(±0.035) | *0.884(±0.017)* | 0.889(±0.009) | **0.889(±0.009)** | 0.823(±0.035) | 0.894(±0.026) |
| 7 → 18 | 0.711(±0.066) | 0.497(±0.011) | 0.497(±0.011) | 0.547(±0.068) | *0.711(±0.029)* | **0.711(±0.066)** | 0.648(±0.123) | 0.547(±0.0) | 0.711(±0.066) |
| Avg. | 0.753(±0.042) | 0.738(±0.028) | 0.738(±0.028) | 0.739(±0.036) | *0.775(±0.026)* | **0.753(±0.042)** | 0.74(±0.053) | 0.728(±0.019) | 0.799(±0.032) |

**AdvSKM**

| Task | SO | Heuristic | | | | Theoretical error guarantees | | | TB |
|---|---|---|---|---|---|---|---|---|---|
| | | TMV | TMR | TCR | SOR | IWV | DEV | IWA (ours) | TB |
| 18 → 23 | 0.711(±0.019) | 0.744(±0.038) | 0.744(±0.038) | 0.744(±0.038) | *0.767(±0.033)* | 0.711(±0.019) | 0.711(±0.019) | **0.744(±0.019)** | 0.778(±0.096) |
| 20 → 30 | 0.872(±0.029) | *0.885(±0.033)* | 0.885(±0.033) | 0.885(±0.033) | 0.872(±0.059) | 0.872(±0.029) | **0.872(±0.029)** | 0.853(±0.029) | 0.885(±0.033) |
| 35 → 31 | 0.619(±0.041) | *0.698(±0.027)* | 0.698(±0.027) | 0.69(±0.082) | 0.675(±0.096) | 0.619(±0.041) | 0.619(±0.041) | **0.698(±0.027)** | 0.714(±0.024) |
| 6 → 19 | 0.818(±0.131) | *0.894(±0.0)* | 0.894(±0.0) | 0.869(±0.044) | 0.833(±0.052) | 0.818(±0.131) | 0.818(±0.131) | **0.874(±0.009)** | 0.894(±0.0) |
| 7 → 18 | 0.717(±0.05) | 0.686(±0.058) | 0.686(±0.058) | *0.704(±0.039)* | 0.566(±0.068) | 0.717(±0.05) | 0.717(±0.05) | **0.717(±0.019)** | 0.736(±0.086) |
| Avg. | 0.747(±0.054) | *0.781(±0.031)* | 0.781(±0.031) | 0.779(±0.047) | 0.742(±0.062) | 0.747(±0.054) | 0.747(±0.054) | **0.777(±0.021)** | 0.801(±0.048) |

**DIRT**

| Task | SO | Heuristic | | | | Theoretical error guarantees | | | TB |
|---|---|---|---|---|---|---|---|---|---|
| | | TMV | TMR | TCR | SOR | IWV | DEV | IWA (ours) | TB |
| 18 → 23 | 0.711(±0.038) | *0.733(±0.033)* | 0.733(±0.033) | 0.733(±0.0) | 0.711(±0.019) | 0.711(±0.038) | 0.711(±0.038) | **0.733(±0.0)** | 0.744(±0.019) |
| 20 → 30 | 0.872(±0.011) | 0.846(±0.0) | 0.846(±0.0) | 0.846(±0.0) | *0.891(±0.04)* | **0.872(±0.011)** | 0.859(±0.029) | 0.84(±0.011) | 0.917(±0.011) |
| 35 → 31 | 0.492(±0.036) | *0.746(±0.036)* | 0.746(±0.036) | 0.746(±0.036) | 0.603(±0.107) | 0.492(±0.036) | **0.77(±0.069)** | 0.746(±0.014) | 0.778(±0.027) |
| 6 → 19 | 0.879(±0.026) | 0.833(±0.052) | 0.833(±0.052) | 0.833(±0.052) | *0.869(±0.023)* | 0.879(±0.026) | **0.879(±0.026)** | 0.838(±0.049) | 0.889(±0.009) |
| 7 → 18 | 0.736(±0.082) | *0.824(±0.011)* | 0.824(±0.011) | 0.824(±0.011) | 0.704(±0.093) | 0.736(±0.082) | 0.767(±0.109) | **0.83(±0.0)** | 0.83(±0.0) |
| Avg. | 0.738(±0.039) | *0.797(±0.027)* | 0.797(±0.027) | 0.797(±0.02) | 0.756(±0.057) | 0.738(±0.039) | 0.797(±0.054) | **0.798(±0.015)** | 0.832(±0.013) |

**DDC**

| Task | SO | Heuristic | | | | Theoretical error guarantees | | | TB |
|---|---|---|---|---|---|---|---|---|---|
| | | TMV | TMR | TCR | SOR | IWV | DEV | IWA (ours) | TB |
| 18 → 23 | 0.7(±0.067) | 0.733(±0.033) | 0.733(±0.033) | *0.767(±0.033)* | 0.7(±0.0) | 0.7(±0.067) | 0.7(±0.067) | **0.744(±0.019)** | 0.767(±0.0) |
| 20 → 30 | 0.833(±0.022) | *0.865(±0.0)* | 0.865(±0.0) | 0.859(±0.032) | 0.833(±0.029) | 0.833(±0.022) | 0.833(±0.022) | **0.84(±0.011)** | 0.885(±0.019) |
| 35 → 31 | 0.54(±0.036) | 0.722(±0.06) | 0.722(±0.06) | *0.746(±0.069)* | 0.635(±0.096) | 0.54(±0.036) | 0.54(±0.036) | **0.746(±0.027)** | 0.73(±0.027) |
| 6 → 19 | 0.884(±0.017) | *0.884(±0.017)* | 0.884(±0.017) | 0.859(±0.032) | 0.869(±0.044) | 0.884(±0.017) | **0.884(±0.017)** | 0.869(±0.023) | 0.884(±0.017) |
| 7 → 18 | 0.748(±0.093) | 0.692(±0.039) | 0.692(±0.039) | *0.704(±0.029)* | 0.648(±0.079) | 0.748(±0.093) | **0.748(±0.093)** | 0.711(±0.011) | 0.748(±0.093) |
| Avg. | 0.741(±0.047) | 0.779(±0.03) | 0.779(±0.03) | *0.787(±0.035)* | 0.737(±0.05) | 0.741(±0.047) | 0.741(±0.047) | **0.782(±0.018)** | 0.803(±0.031) |

**CMD**

| Task | SO | Heuristic | | | | Theoretical error guarantees | | | TB |
|---|---|---|---|---|---|---|---|---|---|
| | | TMV | TMR | TCR | SOR | IWV | DEV | IWA (ours) | TB |
| 18 → 23 | 0.711(±0.038) | 0.678(±0.019) | 0.678(±0.019) | 0.644(±0.069) | *0.689(±0.107)* | 0.689(±0.107) | 0.622(±0.069) | **0.7(±0.0)** | 0.756(±0.038) |
| 20 → 30 | 0.853(±0.011) | 0.859(±0.022) | 0.859(±0.022) | *0.897(±0.029)* | 0.885(±0.051) | 0.853(±0.011) | 0.69(±0.024) | **0.904(±0.0)** | 0.91(±0.022) |
| 35 → 31 | 0.683(±0.014) | *0.722(±0.014)* | 0.722(±0.014) | 0.722(±0.014) | 0.659(±0.014) | 0.683(±0.014) | 0.69(±0.024) | **0.722(±0.027)** | 0.77(±0.014) |
| 6 → 19 | 0.742(±0.084) | 0.798(±0.009) | 0.798(±0.009) | 0.803(±0.015) | *0.833(±0.12)* | 0.783(±0.118) | 0.682(±0.124) | **0.798(±0.035)** | 0.813(±0.12) |
| 7 → 18 | 0.56(±0.044) | *0.767(±0.058)* | 0.767(±0.058) | 0.767(±0.058) | 0.572(±0.061) | 0.56(±0.044) | 0.579(±0.066) | **0.742(±0.029)** | 0.736(±0.058) |
| Avg. | 0.71(±0.038) | 0.765(±0.024) | 0.765(±0.024) | *0.767(±0.037)* | 0.728(±0.071) | 0.713(±0.059) | 0.686(±0.059) | **0.773(±0.018)** | 0.797(±0.05) |

**MMDA**

| Task | SO | Heuristic | | | | Theoretical error guarantees | | | TB |
|---|---|---|---|---|---|---|---|---|---|
| | | TMV | TMR | TCR | SOR | IWV | DEV | IWA (ours) | TB |
| 18 → 23 | 0.756(±0.051) | *0.767(±0.033)* | 0.767(±0.033) | 0.767(±0.033) | 0.756(±0.019) | 0.756(±0.051) | 0.756(±0.051) | **0.8(±0.0)** | 0.889(±0.019) |
| 20 → 30 | 0.872(±0.011) | *0.878(±0.029)* | 0.878(±0.029) | 0.872(±0.029) | 0.814(±0.029) | 0.872(±0.011) | 0.872(±0.011) | **0.878(±0.029)** | 0.891(±0.04) |
| 35 → 31 | 0.571(±0.024) | *0.786(±0.0)* | 0.786(±0.0) | 0.754(±0.036) | 0.683(±0.107) | 0.571(±0.024) | 0.571(±0.024) | **0.77(±0.014)** | 0.746(±0.0) |
| 6 → 19 | 0.879(±0.015) | 0.843(±0.087) | 0.843(±0.087) | 0.747(±0.009) | *0.864(±0.04)* | 0.879(±0.015) | **0.879(±0.015)** | 0.854(±0.009) | 0.909(±0.015) |
| 7 → 18 | 0.717(±0.05) | 0.585(±0.019) | 0.585(±0.019) | 0.585(±0.019) | *0.654(±0.143)* | **0.717(±0.05)** | 0.673(±0.022) | 0.648(±0.011) | 0.723(±0.029) |
| Avg. | 0.759(±0.03) | *0.772(±0.034)* | 0.772(±0.034) | 0.745(±0.025) | 0.754(±0.068) | 0.759(±0.03) | 0.75(±0.025) | **0.79(±0.013)** | 0.832(±0.021) |

Table 28: Mean and standard deviation (after ±) of target classification accuracy on WISDM (Part 2) over 3 repetitions with different random initialization of model weights.

**CoDATS**

| Task | SO | Heuristic | | | | Theoretical error guarantees | | | TB |
|---|---|---|---|---|---|---|---|---|---|
| | | TMV | TMR | TCR | SOR | IWV | DEV | IWA (ours) | |
| 18 → 23 | 0.689(±0.051) | *0.722(±0.069)* | 0.722(±0.069) | *0.733(±0.058)* | 0.667(±0.033) | 0.678(±0.077) | 0.7(±0.067) | **0.756**(±**0.019**) | 0.756(±0.019) |
| 20 → 30 | 0.846(±0.069) | *0.891(±0.029)* | 0.891(±0.029) | 0.885(±0.038) | 0.776(±0.022) | 0.846(±0.069) | 0.846(±0.069) | **0.904**(±**0.019**) | 0.904(±0.051) |
| 35 → 31 | 0.619(±0.095) | *0.738(±0.048)* | 0.738(±0.048) | 0.722(±0.014) | 0.643(±0.124) | 0.619(±0.095) | **0.73**(±**0.036**) | 0.714(±0.063) | 0.786(±0.024) |
| 6 → 19 | 0.727(±0.069) | *0.899(±0.038)* | 0.899(±0.038) | 0.864(±0.069) | 0.742(±0.076) | 0.727(±0.069) | 0.727(±0.069) | **0.899**(±**0.032**) | 0.924(±0.08) |
| 7 → 18 | 0.673(±0.071) | 0.535(±0.076) | 0.535(±0.076) | 0.553(±0.066) | *0.585(±0.098)* | 0.673(±0.071) | **0.673**(±**0.071**) | 0.547(±0.057) | 0.692(±0.142) |
| Avg. | 0.711(±0.071) | *0.757(±0.052)* | 0.757(±0.052) | 0.751(±0.049) | 0.682(±0.071) | 0.709(±0.076) | 0.735(±0.063) | **0.764**(±**0.038**) | 0.812(±0.063) |

**Deep-Coral**

| Task | SO | Heuristic | | | | Theoretical error guarantees | | | TB |
|---|---|---|---|---|---|---|---|---|---|
| | | TMV | TMR | TCR | SOR | IWV | DEV | IWA (ours) | |
| 18 → 23 | 0.656(±0.077) | *0.689(±0.019)* | 0.689(±0.019) | 0.689(±0.019) | 0.667(±0.058) | 0.656(±0.077) | 0.656(±0.077) | **0.689**(±**0.069**) | 0.711(±0.051) |
| 20 → 30 | 0.827(±0.033) | *0.891(±0.029)* | 0.891(±0.029) | 0.872(±0.044) | 0.84(±0.04) | 0.827(±0.033) | 0.827(±0.033) | **0.885**(±**0.033**) | 0.904(±0.038) |
| 35 → 31 | 0.619(±0.086) | *0.69(±0.071)* | 0.69(±0.071) | 0.667(±0.082) | 0.532(±0.122) | 0.619(±0.086) | 0.619(±0.086) | **0.69**(±**0.048**) | 0.706(±0.06) |
| 6 → 19 | 0.722(±0.044) | *0.732(±0.053)* | 0.732(±0.053) | 0.707(±0.035) | 0.717(±0.068) | 0.722(±0.044) | 0.722(±0.044) | **0.722**(±**0.009**) | 0.788(±0.139) |
| 7 → 18 | 0.648(±0.047) | 0.61(±0.054) | 0.61(±0.054) | *0.629(±0.071)* | 0.566(±0.082) | **0.648**(±**0.047**) | 0.528(±0.218) | 0.629(±0.039) | 0.654(±0.029) |
| Avg. | 0.694(±0.057) | *0.723(±0.046)* | 0.723(±0.046) | 0.713(±0.051) | 0.664(±0.074) | 0.694(±0.057) | 0.67(±0.091) | **0.723**(±**0.04**) | 0.753(±0.063) |

**CDAN**

| Task | SO | Heuristic | | | | Theoretical error guarantees | | | TB |
|---|---|---|---|---|---|---|---|---|---|
| | | TMV | TMR | TCR | SOR | IWV | DEV | IWA (ours) | |
| 18 → 23 | 0.767(±0.033) | 0.678(±0.038) | 0.678(±0.038) | 0.722(±0.038) | *0.756(±0.019)* | 0.767(±0.033) | **0.767**(±**0.033**) | 0.7(±0.0) | 0.767(±0.033) |
| 20 → 30 | 0.872(±0.029) | 0.853(±0.029) | 0.853(±0.029) | 0.84(±0.011) | *0.872(±0.04)* | **0.872**(±**0.029**) | 0.833(±0.04) | 0.84(±0.029) | 0.878(±0.044) |
| 35 → 31 | 0.563(±0.05) | 0.738(±0.041) | 0.738(±0.041) | *0.746(±0.036)* | 0.651(±0.144) | 0.563(±0.05) | 0.722(±0.172) | **0.722**(±**0.014**) | 0.77(±0.06) |
| 6 → 19 | 0.869(±0.044) | 0.833(±0.052) | 0.833(±0.052) | *0.838(±0.049)* | 0.838(±0.049) | **0.869**(±**0.044**) | 0.717(±0.306) | 0.848(±0.045) | 0.869(±0.044) |
| 7 → 18 | 0.73(±0.093) | 0.711(±0.115) | 0.711(±0.115) | *0.761(±0.039)* | 0.723(±0.058) | **0.73**(±**0.093**) | 0.711(±0.126) | 0.717(±0.068) | 0.767(±0.076) |
| Avg. | 0.76(±0.05) | 0.762(±0.055) | 0.762(±0.055) | *0.781(±0.035)* | 0.768(±0.062) | 0.76(±0.05) | 0.75(±0.135) | **0.765**(±**0.031**) | 0.81(±0.052) |

**DANN**

| Task | SO | Heuristic | | | | Theoretical error guarantees | | | TB |
|---|---|---|---|---|---|---|---|---|---|
| | | TMV | TMR | TCR | SOR | IWV | DEV | IWA (ours) | |
| 18 → 23 | 0.689(±0.107) | *0.778(±0.069)* | 0.778(±0.069) | 0.778(±0.069) | 0.756(±0.038) | 0.667(±0.088) | 0.656(±0.077) | **0.767**(±**0.0**) | 0.778(±0.019) |
| 20 → 30 | 0.795(±0.068) | 0.846(±0.0) | 0.846(±0.0) | 0.859(±0.022) | *0.865(±0.033)* | 0.795(±0.068) | 0.795(±0.068) | **0.846**(±**0.019**) | 0.885(±0.051) |
| 35 → 31 | 0.651(±0.107) | *0.77(±0.014)* | 0.77(±0.014) | 0.77(±0.027) | 0.659(±0.159) | 0.651(±0.107) | 0.651(±0.107) | **0.77**(±**0.027**) | 0.746(±0.027) |
| 6 → 19 | 0.788(±0.124) | 0.914(±0.046) | 0.914(±0.046) | *0.924(±0.052)* | 0.854(±0.126) | 0.788(±0.124) | 0.788(±0.124) | **0.899**(±**0.061**) | 0.919(±0.052) |
| 7 → 18 | 0.698(±0.033) | *0.704(±0.029)* | 0.704(±0.029) | 0.648(±0.029) | 0.591(±0.104) | **0.698**(±**0.033**) | 0.623(±0.191) | 0.61(±0.054) | 0.698(±0.029) |
| Avg. | 0.724(±0.088) | *0.802(±0.032)* | 0.802(±0.032) | 0.796(±0.04) | 0.745(±0.092) | 0.72(±0.084) | 0.702(±0.113) | **0.778**(±**0.032**) | 0.805(±0.036) |

**DSAN**

| Task | SO | Heuristic | | | | Theoretical error guarantees | | | TB |
|---|---|---|---|---|---|---|---|---|---|
| | | TMV | TMR | TCR | SOR | IWV | DEV | IWA (ours) | |
| 18 → 23 | 0.756(±0.051) | 0.767(±0.033) | 0.767(±0.033) | *0.778(±0.038)* | 0.711(±0.051) | 0.756(±0.051) | **0.756**(±**0.051**) | 0.744(±0.019) | 0.8(±0.0) |
| 20 → 30 | 0.872(±0.011) | *0.859(±0.022)* | 0.859(±0.022) | 0.846(±0.0) | 0.84(±0.011) | **0.872**(±**0.011**) | 0.731(±0.233) | 0.865(±0.0) | 0.872(±0.011) |
| 35 → 31 | 0.571(±0.024) | *0.786(±0.0)* | 0.786(±0.0) | 0.778(±0.014) | 0.698(±0.151) | 0.571(±0.024) | 0.571(±0.024) | **0.778**(±**0.014**) | 0.817(±0.027) |
| 6 → 19 | 0.879(±0.015) | 0.864(±0.052) | 0.864(±0.052) | 0.833(±0.052) | *0.889(±0.009)* | **0.879**(±**0.015**) | 0.662(±0.363) | 0.838(±0.049) | 0.879(±0.009) |
| 7 → 18 | 0.717(±0.05) | 0.572(±0.076) | 0.572(±0.076) | 0.547(±0.068) | *0.648(±0.054)* | **0.717**(±**0.05**) | 0.597(±0.076) | 0.384(±0.047) | 0.717(±0.05) |
| Avg. | 0.759(±0.03) | *0.769(±0.037)* | 0.769(±0.037) | 0.756(±0.035) | 0.757(±0.055) | **0.759**(±**0.03**) | 0.663(±0.149) | 0.722(±0.026) | 0.817(±0.019) |

