# OpenReview forum: "Addressing Parameter Choice Issues in Unsupervised Domain Adaptation by Aggregation"
_ICLR.cc/2023/Conference — ICLR 2023 notable top 5%_

### Official Review · Reviewer_AZPJ · 2022-10-15

**Confidence:** 3
**Correctness:** 4
**Technical Novelty And Significance:** 3
**Empirical Novelty And Significance:** 3
**Recommendation:** 8

**Clarity, Quality, Novelty And Reproducibility:**

The paper is written very clearly and easy to follow. Below I give a few suggestions though they are minor.

- In references, Sugiyamai et al. (2007) should be corrected to Sugiyama et al. (2007).
- When showing the main theoretical claim Theorem 1, I would emphasize that this error analysis is independent of the estimation error of density ratio.
- In Section 5.1, the meaning of Figure 2 is a little bit ambiguous. For example, "the approach proposed in You et al. (2019) and a heuristics baseline" (12nd line of Section 5.1) are compared in Figure 2, but I could not find which parts in Figure 2 indicate them. It is not clear which part in Figure 2 corresponds to "the given sequence of models" (14th line). In addition, I do not understand why three models (IWA, SOR, and DEV) are aggregated as shown at the bottom of Figure 2, given my understanding that IWA is already an aggregation of multiple models.

**Strength And Weaknesses:**

### Strengths

- **Simple method**: The proposed method is based on importance weighting, which is a well-understood yet pretty simple method. In spite of its popularity, importance weighting has not been well studied in the context of linear aggregation for unsupervised domain adaptation. Hence, it does not undermine the novelty or contribution of this paper.
- **Effective empirical performance**: The empirical performances through massive experiments demonstrate the practical effectiveness of the proposed method. It is remarkable that the proposed aggregation approach outperforms the existing methods to select a single best model.

### Weaknesses

- **Theory does not elucidate the effect of the model size.** Although the experiments demonstrate that the aggregation of multiple model candidates outperforms a single best model, it is not evident from the claim of Theorem 1. I tried to figure out from its proof but failed because the model size $\\ell$ is hidden in absolute constants of concentration bounds. Is it possible to comment on what goes on with the target error when we increase $\\ell$?

**Summary Of The Paper:**

This paper handles the model selection problem in the context of unsupervised domain adaptation. In classical domain adaptation literature, model selection is a hard problem because of the lack of labeled target data. In recent years, several linear aggregation methods have been proposed yet they lack theoretical guarantee on the target error. Under the covariate shift assumption, the authors propose a simple yet effective linear aggregation approach based on importance weighting and provide a target error guarantee with respect to the target error incurred by the best linear aggregation. The proposed adaptation algorithm outperforms not only heuristic aggregation methods but also existing single-model selection approaches with theoretical guarantees.

**Summary Of The Review:**

The authors provide a transparent algorithm with a clear presentation, hence I would suggest accepting this paper. Minor issues should be easily fixed.

---

> ### Author Response · Authors · 2022-11-14
> **Response to reviewer AZPJ**
>
> Thank you very much for your review which helps to improve our work! Our answers are as follows:
>
> - Please find the first part of our answer in the general comment above and a discussion on the dependence of $l$ in our new paragraph “On the dependence of the error bound on the number l of models”. Please note that, if instead of excess risk, we measure the performance of the aggregation in terms of the $L^{2}(q_ \mathcal{X})$-difference between constructed predictor and the ideal one (according to Eq. (15) this is the square root of the excess risk), then, with the same arguments as used in the proof of Theorem 1, we obtain an analog of Eq. (3), but with factor 1 instead of factor 2 before the approximation error (first term), and, with a sample error dependency $l^{3/2}$ instead of $l^3$. This is interesting because, (a) we operate in scenarios with $m,n$ being much larger than $l$, i.e. the approximation error dominates the sample error, and, (b) it shows that our method (asymptotically) optimally approximates $f^*$ in $L^{2}(q_ \mathcal{X})$-norm.
> - Thank you very much for the suggestions on minor points. We changed the corresponding places in our revision.
>
> Thank you again for your review! If you are satisfied with our rebuttal, please take this into account in your final decision.

---

> > ### Comment · Reviewer_AZPJ · 2022-11-23
> > **Reply to the authors**
> >
> > Thanks for addressing my comments dedicatedly. The newly added paragraph “relation to model selection” reveals the superiority of the aggregation in a fairly simple way. Anyway the paper contributes significantly and is interesting!

---

### Official Review · Reviewer_WzwU · 2022-10-24

**Confidence:** 4
**Correctness:** 3
**Technical Novelty And Significance:** 3
**Empirical Novelty And Significance:** 3
**Recommendation:** 6

**Clarity, Quality, Novelty And Reproducibility:**

Clarity: Good. This paper is well-written.

Quality: Needs improvements. More insightful discussions, more experiments on existing benchmarks should be provided.

Novelty: Good. The proposed method is somewhat novel.

**Details Of Ethics Concerns:**

No.

**Strength And Weaknesses:**

Strengths:
1. The targeted problem is important and valuable for domain adaptation (without labeled data).
2. This paper is well-written, it is enjoyable to read.
3. Experiments on language, image, text and time-series data show the effectiveness of the proposed method.

Weaknesses:
1. My main concern is the theories for bounding the target error. The conclusion is the target error of the computed aggregation is asymptotically at most twice the target error of the optimal aggregation. However, this bound is loose and cannot give a guarantee of superior performance.
2. Another concern is the selection of datasets. For example, the performance gains achieved by DA methods on Amazon Reviews and WISDM are marginal (The performance of So is close to TB).
3. The authors should conduct experiments on the datasets in the original paper of baselines. For example, experiments on digit, office and visda should be considered.
4. More insightful analyses should be provided. For example, the visualization of the importance weight of each model should be shown. It is interesting if the importance weight can tell us what is the best model in the sequence. Now the paper only tell us what the method is and if the proposed method can achieve better results without further discussions.
5. The scope of the selection model should also be discussed. The ensemble method works well given many effective models. However, if the candidate set of the model contains a bad model, the ensemble method may not work well.



**Summary Of The Paper:**

This paper studies how to address the issue of hyper-parameter selection in unsupervised domain adaptation. Specifically, the authors propose a method that subsequently computes a linear aggregation of the models. In addition, theories for bounding the target error are added to make the method convincing. Experiments show the effectiveness of the proposed method.

**Summary Of The Review:**

Overall, this paper is well-written and contributions are not very marginal. However, more improvements should be added to the paper.

Strengths:
1. The targeted problem is important and valuable for domain adaptation (without labeled data).
2. This paper is well-written, it is enjoyable to read.
3. Experiments on language, image, text and time-series data show the effectiveness of the proposed method.

Weaknesses:
1. My main concern is the theories for bounding the target error. The conclusion is the target error of the computed aggregation is asymptotically at most twice the target error of the optimal aggregation. However, this bound is loose and cannot give a guarantee of superior performance.
2. Another concern is the selection of datasets. For example, the performance gains achieved by DA methods on Amazon Reviews and WISDM are marginal (The performance of So is close to TB).
3. The authors should conduct experiments on the datasets in the original paper of baselines. For example, experiments on digit, office and visda should be considered.
4. More insightful analyses should be provided. For example, the visualization of the importance weight of each model should be shown. It is interesting if the importance weight can tell us what is the best model in the sequence. Now the paper only tell us what the method is and if the proposed method can achieve better results without further discussions.
5. The scope of the selection model should also be discussed. The ensemble method works well given many effective models. However, if the candidate set of the model contains a bad model, the ensemble method may not work well.


------Update------

I carefully read the author's response and most of my concerns are addressed. I'm happy to improve my rating.

---

> ### Author Response · Authors · 2022-11-14
> **Response to reviewer WzwU**
>
> Thank you very much for your review which helps to improve our work! Our answers are:
>
> 1. Please see our general comment (a).
> 2. For completeness, our results include datasets with small domain shifts (e.g., AmRev and WISDM). AmRev is particularly often used as a benchmark dataset. Also in practice, how should we know without labels that we don’t force such a small domain shift scenario?
> 3. Please see our general comment (b).
> 4. The correlation between the aggregation weights and the models are indeed very interesting and we thank you for your suggestion! Please find our results in Section D.5 and Figures 4–6. Indeed, a tendency to positive correlation between aggregation weights and target accuracy can be observed for our method (in contrast to other approaches). Note that another important analyzed property (besides performance) is convergence rate (Theorem 1), which goes beyond the state-of-the-art analysis.
> 5. In fact there are bad models included. For example consider Figure 2, where half of the models are bad, where a “bad model” is defined as a model with target accuracy lower than 80% of the target accuracy of the model computed without domain adaptation (SO). This trend is in general: On average over all datasets, 5% of the models are bad. Based on your helpful comment, we also performed a new experiment on this issue (see general comment (c) above). It turns out that the stability of our method (w.r.t. Adding bad models) even tends to be higher than the one of the baseline methods.
>
> If you are satisfied with our answers and new evaluations, we kindly ask you to take this into account in your final decision.

---

### Official Review · Reviewer_V758 · 2022-10-24

**Confidence:** 3
**Clarity, Quality, Novelty And Reproducibility:** The paper is well written and easy to…
**Correctness:** 3
**Technical Novelty And Significance:** 3
**Empirical Novelty And Significance:** 3
**Recommendation:** 6

**Strength And Weaknesses:**

Strenth:
1. The proposed aggregation approch is general without strong assumptions.
2. The paper gives detailed theoretical analysis.

Weaknesses:
1. For experiments, though the paper’s benchmarks are large and diverse, but still some popular UDA benchmarks, such as VisDA-2017, DomainNet etc are missing. Besides, the paper mainly compares with conventional methods, while certain more recent works are missing.
2. The novelty of the paper is limited besides the combination of IWV with linear aggregation.


**Summary Of The Paper:**

The work proposes a theoretical approach for choosing hyper-parameters in unsupervised domain adaptation. The main strategy is to compute an aggregation of models with target error bound, which theoretically relies on the extension of importance weighted least squares to linear aggregation of vector-valued functions. The paper conducts large scale comparative experiments on language, images, time-series classification tasks.The proposed method outperforms IWV and DEV and sets a new state-of-the-art performance.

**Summary Of The Review:**

The paper proposes a new aggregation models based method in unsupervised domain adaptation. Experiments on language, images, time-series classification tasks show effectiveness of the work.

---

> ### Author Response · Authors · 2022-11-14
> **Response to reviewer V758**
>
> Thank you very much for your review which helps to improve our work!
> For 1., please find our general comment (b) above. Concerning 2., please note the novelty beside our technical introduction of Algorithm 1: (a) First ensemble learning method for deep (vector-valued) unsupervised domain adaptation with theoretical guarantees, (b) First target-error-bound for methods for parameter choice issues in general (IWV et al. show only consistency to the best of our knowledge), (c) Novel state-of-the-art performance in a large-scale study, and (d) we extend the recent open-source benchmark suite Adatime (Ragab et al., 2022) by new methods and datasets.
> If you are satisfied with our answers/changes/rebuttal, we kindly ask you to take this into account in your final decision.

---

### Official Review · Reviewer_2X6X · 2022-10-25

**Confidence:** 3
**Correctness:** 3
**Technical Novelty And Significance:** 2
**Empirical Novelty And Significance:** 3
**Recommendation:** 6

**Clarity, Quality, Novelty And Reproducibility:**

Novelty: This proposed method is somewhat novel.

Quality: The experimental validation is extensively evaluated for different tasks.

Clarity: This paper is well written, and the theoretical analysis is easy to follow from the outline provided.

Reproducibility: The code needed to reproduce the experimental results is provided.

**Strength And Weaknesses:**

Strengths:

1: This paper proposes a simple but effective method that extends weighted least squares to deep neural networks for hyperparameter selection in unsupervised domain adaptation.

2: Experiments on different tasks are conducted to verify the superiority of the proposed algorithm.

3: This paper is well-written and easy to follow. The background knowledge is presented well. The readers can easily understand the paper.


Weaknesses:

1. The baselines are few. Do the baselines include important and all the SOTA UDA methods?

2. In theory, the authors show that the target error of the proposed algorithm is asymptotically not worse than twice the error of the unknown optimal aggregation. Such a bound cannot provide guarantees of better performance of the proposed method over others.




**Summary Of The Paper:**

This paper proposed a method that extends weighted least squares to deep neural networks for hyperparameter selection in unsupervised domain adaptation. In theory, the authors show that the target error of the proposed algorithm is asymptotically not worse than twice the error of the unknown optimal aggregation. Numerical validate the superiority of the proposed method over the DEV and the IWV method.

**Summary Of The Review:**

This paper proposes the first method that extends weighted least squares to deep neural networks for hyperparameter selection in unsupervised domain adaptation. The method is technically sound.

------Update------

I have read the author's response and most of my concerns are addressed. I will keep the score for acceptance.

---

> ### Author Response · Authors · 2022-11-14
> **Response to reviewer 2X6X**
>
> Thank you very much for your comments which help to improve our work! Our answers are as follows:
> - From our point of view, the baselines are chosen in accordance to recent knowledge in the field (5 of 12 domain adaptation baselines are 2-years old, 7 public datasets, all available theory-based model selection methods, representative set of ensemble baselines, 6-months old public benchmark suite Adatime). Please see our general comment (b) above for further details.
> - For clarification, we added a short paragraph to the paper. Please see also our general comment (a) above for details.
>
> If you are satisfied with our answers and extensions, we kindly ask you to take this into account in your final decision.

---

### Author Response · Authors · 2022-11-14
**General response to all reviewers**

We thank all reviewers for the invested time, your efforts and the constructive comments which help to improve our work! We especially appreciate:
- Consensus about empirical effectiveness of Algorithm 1, correctness of proofs, and clarity of the paper.
- That our approach is (to the best of the authors' and the reviewers' knowledge) the first ensemble learning method with theoretical guarantees in deep unsupervised domain adaptation.
- Explicitly emphasizing comments of 2X6X, V758, AZPJ on the large-scale of our experiments.

Reviewers’ concerns are mainly about (a) the message of Theorem 1 (2X6X, WzwU, AZPJ), (b) the choice of baselines (2X6X, V758, WzwU), and (c) the behavior of Algorithm 1 beyond the outperformance of baselines (WzwU). Our main answers are:

**a)** It is easy to show that the optimal aggregation $f^*$ is better than the best single model selection $\mathcal{R}_ q(f^*)\leq \min_ {f_1,\ldots,f_l} \mathcal{R}_ q(f_i)$, see new paragraph “relation to model selection”. We agree with the reviewers that this proves outperformance only for infinite sample sizes $m,n$. But, Theorem 1 proves consistency, i.e., convergence (for $m,n\to\infty$) to an optimal error (first term in Eq. (3)). The factor $2$ in front of the error even disappears when working with the $L^{2}(q_ \mathcal{X})$-norm instead of the excess risk (due to the square in Eq. (16)). To the best of our knowledge, Theorem 1 is the **first error rate result** $\mathcal{O}(n^{-1} +  m^{-1})$ for a parameter choice solution in UDA (IWV provides only consistency), although such a result is of major interest in learning theory.

**b)** 5 of our 12 domain adaptation baselines (AdvSKM, DSAN, HoMM, MMDA, CoDATS) were published in the last 2 years. For parameter choice baselines, there are mainly 4 possible options: (a) theory-based (consistent) aggregation, (b) heuristic aggregation, (c) theory-based selection (only one model), (d) heuristic selection. In group (a), our method is the first one (to the best of our knowledge). In (b), we select a representative set of 4 heuristics corresponding to references in the second paragraph of the Related Work Section 2.. In (c), we are only aware of IWV and DEV. In group (d), e.g., [A], it is clear that our method outperforms any other method on at least 4 of 7 datasets (TransfMoons, AmRev, SleepEDV, HHAR), as we outperform the best single model (TB in Table 12). In sum, we used 7 datasets (including a reduced version of VisDA/DomainNet-2019), 38 source-target pairs, 12 domain adaptation methods, 6 parameter choice baselines; in total, we trained 16000 models and our benchmark tables span 15 pages in the supplementary material. From our point of view a further extension of this large-scale setup **cannot substantially change the result** of the empirical effectiveness.

**c)** We performed two new evaluations. The results further generalize Figure 2: The aggregation **weights tend to favor accurate models** (see Sections D.5), and, our method is **stable w.r.t. many inaccurate models** in the given sequence (see Section D.6). It turns out that this trend generalizes beyond the already provided examples: Our method tends to be more stable (w.r.t. adding inaccurate models) than others. This might serve as another reason for its new state-of-the-art performance.

[A] Saito, K., Kim, D., Teterwak, P., Sclaroff, S., Darrell, T., & Saenko, K. (2021). Tune it the right way: Unsupervised validation of domain adaptation via soft neighborhood density. In Proceedings of the IEEE/CVF International Conference on Computer Vision (pp. 9184-9193).

---

### Decision · Program_Chairs · 2023-01-20

**Decision:**

Accept: notable-top-5%

**Justification For Why Not Higher Score:**

N/A

**Justification For Why Not Lower Score:**

A good paper with good theoretical foundation and extensive experimental evaluation.

**Metareview: Summary, Strengths And Weaknesses:**

Authors have developed a general and theoretical approach to choose good hyper-parameters in unsupervised domain adaptation. Its key idea is to compute an aggregation of models with target error bound. In addition, it performed extensive experiments on language, images, and time-series classification tasks, demonstrating it performs better than existing state-of-the-art methods. While authors can use more benchmark datasets to do more comparison, overall it has done quite a number of experiments. It is a well written paper and easy to follow.

**Note From Pc:**

if the above contains the word "oral" or "spotlight" please see: "oral" presentation means -> notable-top-5% and "spotlight" means -> notable-top-25%. As stated in our emails, we are disassociating presentation type from AC recommendations

**Summary Of Ac-Reviewer Meeting:**

As average rating is 6.5, we did not do the meeting